# Is Best-of-N the Best of Them?
# Coverage, Scaling, and Optimality in Inference-Time Alignment

**Audrey Huang** [1]  **Adam Block** [2]  **Qinghua Liu** [2]  **Nan Jiang** [1]  **Akshay Krishnamurthy** [2]  **Dylan J. Foster** [2]

## Abstract

Recent work on *inference-time alignment* has established the benefits of increasing inference-time computation in language models, but naively scaling compute through techniques like Best-of-N sampling can cause performance to degrade due to reward hacking. Toward a theoretical understanding of how to best leverage additional computation, we formalize inference-time alignment as improving a pre-trained policys responses for a prompt of interest, given access to an imperfect reward model. We analyze the performance of inference-time alignment algorithms in terms of (i) response quality, and (ii) compute, and provide new results that highlight the importance of the pre-trained policys *coverage* over high-quality responses for performance and compute scaling: (1) We show that Best-of-N alignment with an ideal $N$ can achieve optimal performance under stringent notions of coverage, but provably suffers from reward hacking when $N$ is large, and fails to achieve tight guarantees under more realistic coverage conditions; (2) We introduce `InferenceTimePessimism`, a new algorithm which mitigates reward hacking through deliberate use of inference-time compute, implementing *pessimism in the face of uncertainty*; we prove that its performance is *optimal* and *scaling-monotonic*, i.e., does not degrade as $N$ increases. We complement our theoretical results with experiments that demonstrate the practicality of our algorithm across a variety of tasks and models.

## 1. Introduction

*Inference-time computation* has emerged as a new axis for scaling in language models, which has led to dramatic

[1]Computer Science Department, University of Illinois at Urbana-Champaign, Champaign, Illinois, USA [2]Microsoft Research, New York, New York, USA. Correspondence to: Audrey Huang <audreyh5@illinois.edu>.

*Proceedings of the 42$^{nd}$ International Conference on Machine Learning*, Vancouver, Canada. PMLR 267, 2025. Copyright 2025 by the author(s).

improvements in their capabilities (Brown et al., 2024; Snell et al., 2024; Wu et al., 2024b; OpenAI, 2024b; DeepSeek-AI, 2025) and played a central role in recent AI breakthroughs (Chollet et al., 2024). While there are a multitude of ways in which additional computation can be utilized during inference—e.g., to generate long chains of thought (Wei et al., 2022; Li et al., 2024b), have the model rate or correct its own responses (Zheng et al., 2023; Wu et al., 2024a), or implement planning and search (Yao et al., 2024; Zhang et al., 2024)—even exceedingly simple methods like *Best-of-N (parallel) sampling* can provide significant performance gains (Lightman et al., 2023; Brown et al., 2024), and enjoy provable representational benefits (Huang et al., 2024a). Such algorithms are also widely used throughout post-training for frontier models (DeepSeek-AI, 2025; Yang et al., 2024a), and thereby serve as a cornerstone of fine-tuning and inference.

Toward developing a deeper understanding of the algorithmic landscape for inference-time computation, in this paper, we focus on the problem of *inference-time alignment*: given a task and a reward model that is a proxy for task performance, how can we best use inference-time computation to improve the quality of a response chosen from many candidates generated by the base model? The Best-of-N heuristic is one of the most widely used inference-time alignment methods, and proceeds by generating $N$ candidate responses for a given prompt, then returning the response with the highest reward under a reward model (Stiennon et al., 2020; Nakano et al., 2021; Touvron et al., 2023; Gao et al., 2023; Eisenstein et al., 2023; Mudgal et al., 2024). While attractive in its simplicity, Best-of-N and related heuristics are known to suffer from *reward overoptimization* or *reward hacking* when $N$ increases (Gao et al., 2023; Stroebl et al., 2024; Chow et al., 2024; Frick et al., 2024). While one might hope that increasing computation to generate a larger number of candidates will increase the likelihood of selecting a high-quality response, reward model errors at the tail of the response distribution can cause Best-of-N to return generations with high (modeled) reward, but poor task performance.

This overoptimization phenomenon raises two fundamental questions, which we investigate in this paper: (1) what is the extent to which the imperfect reward model limits

the performance of inference-time alignment methods; and (2) can more deliberate algorithm design lead to performance gains that scale monotonically with increased computation? Beyond guiding practical interventions for inference-time alignment, answers to these questions contribute to a foundational understanding of inference-time computation more broadly.

**Our framework.** To address the questions above, we pose *inference-time alignment*[1] as the task of extracting a high-quality response $\widehat{y}$ for a prompt $x$ from a pre-trained language model, or *base policy* $\pi_{\text{ref}} : \mathcal{X} \to \Delta(\mathcal{Y})$, that maps a prompt $x$ to a distribution over responses $y \in \mathcal{Y}$. The quality of the response $\widehat{y}$ is determined by an underlying *true reward function* $r^\star : \mathcal{X} \times \mathcal{Y} \to [0, R_{\text{max}}]$, that expresses, for example, the correctness of a math proof or the helpfulness of a chat response. The algorithm designer does not know $r^\star$ and instead uses an *imperfect reward model* $\widehat{r}$ (e.g., one learned from preference-based feedback (Christiano et al., 2017; Ouyang et al., 2022; Wang et al., 2024b)) to maximize performance as follows. Using black-box access to $\pi_{\text{ref}}$ and $\widehat{r}$, i.e., *sampling queries* $y \sim \pi_{\text{ref}}(\cdot \mid x)$ and *evaluation queries* $\widehat{r}(x, y)$ and $\pi_{\text{ref}}(y \mid x)$, we aim to produce a response $\widehat{y}$ that approximately maximizes the true reward (thereby minimizing *regret* to the optimal policy),

$$r^\star(x, \widehat{y}) \approx \max_{y \in \mathcal{Y}} r^\star(x, y). \tag{1}$$

This formulation presents two central questions:

**Q1**: **Regret.** *How close to optimal can we make the reward $r^\star(x, \widehat{y})$ in Eq. (1), as a function of the quality of the model $\widehat{r}$?*

**Q2**: **Compute.** *What is the computational cost—measured by the number of sampling queries $y \sim \pi_{\text{ref}}(\cdot \mid x)$, and evaluation queries $\widehat{r}(x, y)$ and $\log \pi_{\text{ref}}(y \mid x)$ used by the algorithm—required for optimal reward?*

Note that while our goal is to maximize the true reward $r^\star$, the quality of the reward model $\widehat{r}$ creates an inherent, information-theoretic barrier to optimizing $r^\star$, since the latter unobserved. Intuitively, we should not be able to maximize quality under $r^\star$ if $\widehat{r}$ is highly inaccurate.

## 1.1. Contributions

We show that Best-of-N alignment and related heuristics may fail to achieve optimal regret, but that more sophisticated use of inference-time computation—namely, to extract additional information from the reward model and quantify its uncertainty—can mitigate overoptimization, and achieve optimal regret and compute scaling.

**Statistical framework & necessity of coverage (Sec. 2).**

---

[1] Following prior work (e.g., Rafailov et al. (2023); Ye et al. (2024)), we adopt contextual bandit/reinforcement learning terminology, interpreting the language model as a policy.

Our formal framework, summarized above, reformulates inference-time alignment as a *statistical problem* via query complexity. This allows us to derive fundamental limits on the performance of any inference-time alignment algorithm. We show that the best possible reward one can achieve, irrespective of computational cost, is determined by the base policy's *coverage* over high-quality responses, along with the mean-squared error of $\widehat{r}$. This serves as a skyline for our investigation into improved algorithmic interventions.

**Tight analysis of `BoN-Alignment` (Sec. 3).** Within our framework, we offer the first theoretical analysis of the regret of Best-of-N alignment (`BoN-Alignment`). We show that `BoN-Alignment` can achieve optimal regret under a stringent notion of coverage ("uniform" or $L_\infty$-type) when $N$ is tuned appropriately, but:

1. provably suffers from overoptimization, degrading in performance once $N$ scales past a critical threshold; and

2. fails to achieve tight guarantees under weaker notions of coverage ("average-case" or $L_1$-type), thereby falling short of the skyline established in Section 2.

**Optimal algorithm: `InferenceTimePessimism` (Sec. 4).** Motivated by the shortcomings of Best-of-N, we introduce an improved algorithm, `InferenceTimePessimism`. We prove that `InferenceTimePessimism`:

1. is *regret-optimal*, in the sense that it can achieve the best possible reward in our framework, thereby matching the skyline in Section 2; and

2. is *scaling-monotonic*, in the sense that it is guaranteed to avoid overoptimization beyond a certain point (determined by the regularization parameter), even as $N \to \infty$.

To achieve this, our algorithm uses a novel rejection sampling scheme to implement $\chi^2$-regularization—which is known to mitigate overoptimization via the principle of *pessimism in the face of uncertainty* (Huang et al., 2024b)—purely at inference time. Beyond achieving optimal regret, we show that `InferenceTimePessimism` uses near-optimal compute under our framework.

**Empirical evaluation (Sec. 5).** To demonstrate the benefits of `InferenceTimePessimism`, we compare the algorithm to `BoN-Alignment` across several tasks, base policies, and reward models. As predicted by our theory, `BoN-Alignment` degrades in performance as computation increases (via $N$), while `InferenceTimePessimism` is scaling-monotonic and does not suffer from this characteristic overoptimization phenomenon (Figure 1). We also observe several instances where `InferenceTimePessimism` outperforms `BoN-Alignment` in terms of maximal reward achieved when the computational budget is untuned, demonstrating the robustness of our algorithm. Appendix B contains further empirical results, including

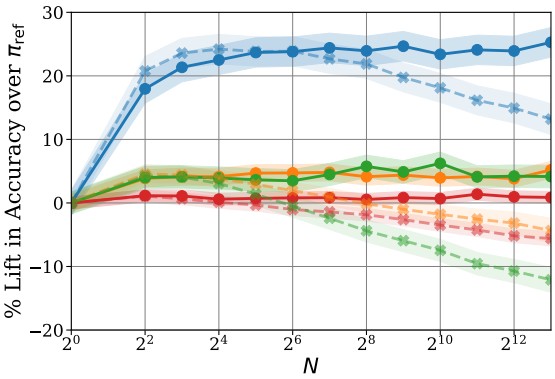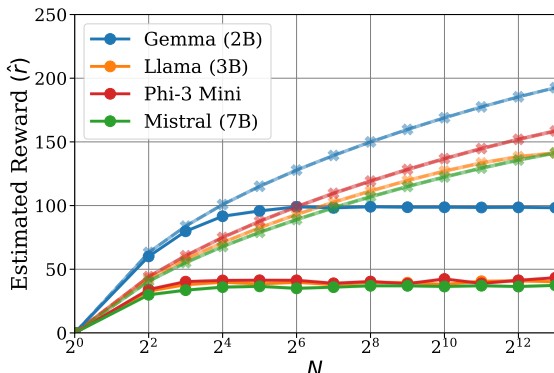

*Figure 1.* Comparison between performance of `BoN-Alignment` (dashed lines) and `InferenceTimePessimism` algorithm (solid lines) on GSM8K with reward model `OASST` as $\hat{r}$ and several different choices of $\pi_{\mathsf{ref}}$. **Left**: `BoN-Alignment` initially improves accuracy over $\pi_{\mathsf{ref}}$, but eventually degrades as $N$ increases, while `InferenceTimePessimism` is monotone, as predicted by our theory. **Right**: `BoN-Alignment` overoptimizes the reward model, with high $\hat{r}$ but lower accuracy as $N$ increases, whereas `InferenceTimePessimism` stops increasing $\hat{r}$ with $N$ beyond a certain threshold determined by a regularization parameter.

examinations of the reward model's tail behavior and demonstrations of `InferenceTimePessimism`'s robustness to choice of hyperparameter.

### 1.2. Related Work

To our knowledge, our work provides the first theoretical framework for understanding and mitigating overoptimization in inference-time alignment. Various works have analyzed specific properties of the Best-of-N alignment algorithm (Yang et al., 2024b; Beirami et al., 2024; Mroueh, 2024) such as tradeoffs between reward and KL-divergence, but do not ultimately provide guarantees on performance when the estimated reward model and true reward are mismatched. We focus on analyzing Best-of-N specifically because it is the most widely used and foundational inference-time alignment technique, but other algorithms (Welleck et al., 2024) including variants of rejection sampling (Khanov et al., 2024; Chen et al., 2024; Shi et al., 2024; Liu et al., 2024a; Jinnai et al., 2024) and Monte-Carlo Tree Search (Feng et al.; Yao et al., 2024; Zhang et al., 2024), are also used in practice.

Our work is also closely related to research on *offline alignment at training time* (Christiano, 2014; Ouyang et al., 2022; Rafailov et al., 2023), which involves fitting a reward model $\hat{r}$ (typically through pairwise feedback), and then maximizing it using RL. A growing body of theoretical works on offline alignment provide theoretical guarantees for mitigating reward overoptimization that—like our work—depend on notions of *coverage* for the base policy (Zhu et al., 2023; Zhan et al., 2023a; Li et al., 2023b; Xiong et al., 2024; Liu et al., 2024b; Cen et al., 2024; Fisch et al., 2024; Ji et al., 2024; Huang et al., 2024b). Our formulation for inference-time alignment can be viewed as a variant of this problem that abstracts away the reward model training; in particular, we take $\hat{r}$ as given and ask how to achieve the best possible performance on a *per-instance*

basis, with respect to the true reward $r^{\star}$ and the prompt $x \in \mathcal{X}$.

Our algorithm, `InferenceTimePessimism`, is closely related to $\chi$PO (Huang et al., 2024b), a training-time offline alignment algorithm that aims to mitigate overoptimization via regularization with the $\chi^2$-divergence. `InferenceTimePessimism` also leverages $\chi^2$-regularization, but implements this purely at inference-time via a novel rejection sampling scheme. **See Appendix A.1 for a detailed discussion of connections to offline alignment.** Finally, our analysis draws on the treatment of *approximate rejection sampling* in Block & Polyanskiy (2023), which tightly characterizes the performance of rejection sampling with unbounded likelihood ratios.

## 2. Inference-Time Alignment Framework

In this section, we formally introduce our inference-time alignment framework, then use it to prove lower bounds that highlight the necessity of coverage. In our framework, the algorithm designer begins with a *base policy* $\pi_{\mathsf{ref}} : \mathcal{X} \to \Delta(\mathcal{Y})$, which is a conditional distribution mapping *prompts* $x \in \mathcal{X}$ to distributions over *responses* $y \in \mathcal{Y}$, and $\pi_{\mathsf{ref}}(y \mid x)$ is the probability that the base policy generates a response $y$ given the prompt $x$.[2] The algorithm designer also has access to an *imperfect reward model* $\hat{r} : \mathcal{X} \times \mathcal{Y} \to [0, R_{\mathsf{max}}]$, where $R_{\mathsf{max}} \geq 1$, and the reward label $\hat{r}(x, y)$ estimates the quality of the response $y$ for the prompt $x$. This serves as a proxy for the *true reward model* $r^{\star} : \mathcal{X} \times \mathcal{Y} \to [0, R_{\mathsf{max}}]$, which is unknown.

Given the base policy $\pi_{\mathsf{ref}}$, the reward model $\hat{r}$, and a

---

[2]The motivating special case is autoregressive language modeling, where $\mathcal{Y} = \mathcal{V}^H$ for a vocabulary space $\mathcal{V}$ and sequence length $H$, and $\pi_{\mathsf{ref}}$ has the autoregressive structure $\pi_{\mathsf{ref}}(y_{1:H} \mid x) = \prod_{h=1}^{H} \pi_{\mathsf{ref}}(y_h \mid y_{1:h-1}, x)$ for $y = y_{1:H} \in \mathcal{Y}$.

prompt $x \in \mathcal{X}$, the goal is to produce a high quality response $\widehat{y} \in \mathcal{Y}$ as measured by the true reward $r^\star$, in the sense that the following *inference-time regret* is small.

$$\mathsf{Reg}_{\pi^\star}(\widehat{y}\,;x) := J(\pi^\star\,;x) - J(\widehat{y}\,;x) \leq \varepsilon, \qquad (2)$$

Here, $\pi^\star$ can be viewed as a comparator policy with high reward, and $J(\pi\,;x) := \mathbb{E}_{y\sim\pi(\cdot|x)}[r^\star(x,y)]$ denotes the expected reward under $r^\star$ for any policy $\pi$. When the response $\widehat{y}$ is chosen by a randomized algorithm $\mathcal{A}$, we use $\widehat{\pi}_{\mathcal{A}}(\cdot \mid x)$ to denote the conditional distribution over responses induced by $\mathcal{A}$, and write regret as $\mathsf{Reg}_{\pi^\star}(\widehat{\pi}\,;x) := J(\pi^\star\,;x) - J(\widehat{\pi}_{\mathcal{A}}\,;x)$.

In applications, the base policy $\pi_{\mathsf{ref}}$ represents a language model trained in some manner. The true reward $r^\star$ can represent the extent to which $y$ agrees with human preference, passes a proof checker, or passes a unit test suite. The reward model $\widehat{r}$ can be an open source reward model, a model trained by the algorithm designer themselves, or even derived directly from $\pi_{\mathsf{ref}}$ itself (Wang et al., 2022; Huang et al., 2024a; Song et al., 2024).

**Reward model quality versus regret.** There is no hope of making the regret in Eq. (2) small without requirements on the fidelity of the reward model $\widehat{r}$. For a prompt $x \in \mathcal{X}$, we measure the quality of $\widehat{r}$ via the expected squared error with respect to $r^\star$, where responses are drawn by the base policy $\pi_{\mathsf{ref}}$:

$$\varepsilon_{\mathsf{RM}}^2(x) := \mathbb{E}_{y\sim\pi_{\mathsf{ref}}(\cdot|x)}\left[(\widehat{r}(x,y) - r^\star(x,y))^2\right]. \quad (3)$$

Empirically and in theory, one can minimize Eq. (3) by fitting $\widehat{r}$ to rewards or human preference data for responses generated from $\pi_{\mathsf{ref}}$ (cf. Appendix A.1), which is already standard in post-training pipelines. As our focus is on *inference-time* interventions, we simply abstract the reward model training step away and assume we have a reward model $\widehat{r}$ with error $\varepsilon_{\mathsf{RM}}^2$, as defined in Eq. (3). We aim to understand how small we can make the regret in Eq. (2) as a function of the reward model quality $\varepsilon_{\mathsf{RM}}^2$, and what algorithmic interventions are required to achieve this (cf. **Q1**).

**Remark 2.1.** *In practice, the reward estimation error in Eq. (3) may not be the tightest notion of reward error, and thus our bounds may be conservative. Nevertheless, it is arguably the most common formulation of reward mismatch (Zhu et al., 2023; Zhan et al., 2023a; Xiong et al., 2024; Gao et al., 2024; Huang et al., 2024b), and serve as a foundation for future investigations into other notions of error.*

**Measuring computational efficiency via query complexity.** The next question we consider (cf. **Q2**) is how much computation is required to minimize the inference-time regret in Eq. (2) for a given prompt $x$. To allow for computational understanding that is agnostic to the architecture or description of $\pi_{\mathsf{ref}}$ and $\widehat{r}$, we draw inspiration from Huang

et al. (2024a) and abstract it as a *statistical problem*, where computational efficiency is quantified via the number of samples and labels queried from the base policy and reward model.

**Definition 2.2** (Sample-and-evaluate framework). *In the **sample-and-evaluate** framework, the algorithm designer does not have explicit access to the base policy $\pi_{\mathsf{ref}}$ or the reward model $\widehat{r}$. Instead, they access $\pi_{\mathsf{ref}}$ and $\widehat{r}$ through* sample-and-evaluate queries*: For a given prompt $x \in \mathcal{X}$, they can sample $N$ responses $y_1, y_2, \ldots y_N \sim \pi_{\mathsf{ref}}(\cdot \mid x)$ and observe the likelihood $\pi_{\mathsf{ref}}(y_i \mid x)$ and reward model value $\widehat{r}(x, y_i)$ for each such response. The efficiency of the algorithm is measured by the total number of queries $N$.*

This is a natural statistical abstraction for computation— analogous to oracle/query complexity in optimization (Nemirovski et al., 1983; Traub et al., 1988; Raginsky & Rakhlin, 2011; Agarwal et al., 2012), and learning theory (Blum et al., 1994; Kearns, 1998; Feldman, 2012; 2017)— and encompasses Best-of-N alignment and other schemes such as rejection sampling (Khanov et al., 2024; Chen et al., 2024; Shi et al., 2024; Liu et al., 2024a; Jinnai et al., 2024).

### 2.1. A Skyline: The Necessity of Coverage
To motivate our main algorithmic results, and as a step toward answering question **Q1**, we begin by proving a lower bound on the best possible regret one can hope to achieve, as determined by the reward model's error $\varepsilon_{\mathsf{RM}}^2(x)$. This lower bound highlights the importance of the base policy's *coverage*, defined formally for a comparator policy $\pi^\star$ via

$$\mathcal{C}^{\pi^\star}(x) := \mathbb{E}_{y\sim\pi^\star(\cdot|x)}\left[\frac{\pi^\star(y \mid x)}{\pi_{\mathsf{ref}}(y \mid x)}\right]. \qquad (4)$$

**Proposition 2.3** (Necessity of coverage). *Fix a prompt $x \in \mathcal{X}$, and $\pi_{\mathsf{ref}} : \mathcal{X} \to \Delta(\mathcal{Y})$. For any alignment algorithm $\mathcal{A}$ and any $16 \leq C^\star \leq \max_{\pi:\mathcal{X}\to\Delta(\mathcal{Y})} \mathcal{C}^\pi(x)$, there exists a reward function $r^\star$ and reward model $\widehat{r}$ satisfying Eq. (3) and comparator policy $\pi^\star$ with $\mathcal{C}^{\pi^\star}(x) \leq C^\star$ such that*

$$J(\pi^\star\,;x) - J(\widehat{\pi}_{\mathcal{A}}\,;x) \geq \frac{1}{4} \cdot \sqrt{C^\star \cdot \varepsilon_{\mathsf{RM}}^2(x)}.$$

Coverage in the sense of Eq. (4) plays a central role in the theoretical study of offline alignment (Zhu et al., 2023; Zhan et al., 2023a; Li et al., 2023b; Xiong et al., 2024; Liu et al., 2024b; Cen et al., 2024; Fisch et al., 2024; Ji et al., 2024; Huang et al., 2024a;b), and Proposition 2.3 shows that it plays a similar role for inference-time alignment. Recent works show that standard language models can indeed exhibit favorable coverage over desirable responses in standard tasks of interest, which translate to performance gains at inference time (Brown et al., 2024; Snell et al., 2024; Wu et al., 2024b). In the remainder of the paper, we will

**Algorithm 1** Best-of-N Alignment (BoN-Alignment)
___
**input:** Prompt $x$, queries $N$, reference policy $\pi_{\mathsf{ref}}$, reward model $\widehat{r}$
1: Draw $\widehat{\mathcal{Y}}_N = (y_1, \ldots, y_N) \sim \pi_{\mathsf{ref}}(\cdot \mid x)$ i.i.d.
2: Query $\widehat{r}$ for reward labels $(\widehat{r}(x, y_1), \ldots, \widehat{r}(x, y_N))$
3: **return** response $y = \arg\max_{y_i \in \widehat{\mathcal{Y}}_N} \widehat{r}(x, y_i)$
___

explore the extent to which this skyline can be achieved—through existing techniques, or through new interventions.

**Additional notation.** We adopt standard big-oh notation, and write $f \lesssim g$ as shorthand for $f = O(g)$ and $f = \widetilde{O}(g)$ to denote $f = O(g \cdot \max\{1, \mathrm{polylog}(g)\})$.

## 3. Understanding Best-of-N Alignment

As our first result, we give a sharp analysis of Best-of-N alignment, highlighting the role of coverage in determining its scaling properties and (sub-) optimality.

### 3.1. A Sharp Analysis of Best-of-N Alignment

In Algorithm 1 we display the Best-of-N algorithm for an input prompt $x \in \mathcal{X}$. BoN-Alignment draws $N$ candidate responses $y_1, \ldots, y_N \sim \pi_{\mathsf{ref}}(\cdot \mid x)$ i.i.d., and returns the response $y \in \widehat{\mathcal{Y}}_N$ that has the largest reward under the reward model $\widehat{r}$. Next, for a fixed prompt $x$ and sample size parameter $N$, let $\widehat{\pi}_{\mathsf{BoN}}(x) \in \Delta(\mathcal{Y})$ denote the policy corresponding to the distribution over responses output by BoN-Alignment, which is a random variable depending on draws of candidate $\widehat{\mathcal{Y}}_N \sim \pi_{\mathsf{ref}}$. which, given $x \in \mathcal{X}$, draws $N$ responses $y_1, \ldots, y_N \sim \pi_{\mathsf{ref}}(\cdot \mid x)$ i.i.d., and returns the response $\widehat{y} = \arg\max_{y_i} \widehat{r}(x, y_i)$. Our main guarantee for BoN-Alignment bounds the regret in terms of the sample size $N$, the coverage coefficient $\mathcal{C}^{\pi^\star}(x)$, and the reward model error $\varepsilon_{\mathsf{RM}}^2(x)$.

**Theorem 3.1** (Guarantee for BoN-Alignment). *For any prompt $x \in \mathcal{X}$, reward model error $\varepsilon_{\mathsf{RM}}^2(x) \in (0, 1]$, and comparator $\pi^\star$, whenever $N \geq c \cdot \mathcal{C}^{\pi^\star}(x) \cdot \log(R_{\mathsf{max}}/\varepsilon_{\mathsf{RM}}(x))$ for a sufficiently large constant $c$, the BoN-Alignment policy $\widehat{\pi}_{\mathsf{BoN}}$ satisfies*

$$J(\pi^\star; x) - J(\widehat{\pi}_{\mathsf{BoN}}; x) \tag{5}$$
$$\lesssim R_{\mathsf{max}} \cdot \frac{\mathcal{C}^{\pi^\star}(x) \log(R_{\mathsf{max}}/\varepsilon_{\mathsf{RM}}(x))}{N} + \sqrt{N \cdot \varepsilon_{\mathsf{RM}}^2(x)}.$$

*In particular, as long as $N \asymp \left( \frac{R_{\mathsf{max}} \cdot \mathcal{C}^{\pi^\star} \log(R_{\mathsf{max}}/\varepsilon_{\mathsf{RM}}(x))}{\varepsilon_{\mathsf{RM}}(x)} \right)^{\frac{2}{3}}$,[3]*

$$J(\pi^\star; x) - J(\widehat{\pi}_{\mathsf{BoN}}; x) \tag{6}$$
$$\lesssim \left( R_{\mathsf{max}} \cdot \mathcal{C}^{\pi^\star}(x) \cdot \varepsilon_{\mathsf{RM}}^2(x) \cdot \log(R_{\mathsf{max}}/\varepsilon_{\mathsf{RM}}(x)) \right)^{1/3}.$$

___
[3]We use $N \asymp \square$ to indicate that $C_1 \square \leq N \leq C_2 \cdot \square$ for any sufficiently large absolute constants $C_1 \leq C_2$.

The main regret bound in Eq. (5) has two terms of interest. The first, which scales as roughly $\frac{\mathcal{C}^{\pi^\star}(x)}{N}$, decreases as the sample size increases, and reflects the extent to which the set of responses $y_1, \ldots, y_N \sim \pi_{\mathsf{ref}}(\cdot \mid x)$ contains enough information to compete with $\pi^\star$. Intuitively, as $N$ grows, there is a higher probability of drawing a response that, under the true reward $r^\star$, is at least as good as one from the comparator $\pi^\star$. However, $r^\star$ is not available to the learner, who instead evaluates response quality using the imperfect proxy $\widehat{r}$. There becomes a greater risk of overfitting to errors in $\widehat{r}$ as $N$ increases, and candidates are drawn from the tail of the base distribution where $\widehat{r}$ is more error-prone. The cost of this overoptimization is expressed in the second term of Eq. (5), $\sqrt{N \cdot \varepsilon_{\mathsf{RM}}^2(x)}$, which increases with sample size and leads to arbitrarily large regret as $N$ is scaled.

Because sample size is the only parameter in BoN-Alignment, it plays dual, but opposing, roles in both performing regularization (smaller $N$ to stay on-support) and increasing response quality (larger $N$ to draw good responses). The optimal choice of $N$ must balance gains in quality with risk of overoptimization and be large but not *too* large, which leads to the second guarantee in Theorem 3.1. Eq. (6) resembles the idealized lower bound in Proposition 2.3 but has a slower rate, scaling with $\varepsilon_{\mathsf{RM}}^{2/3}(x)$ instead of $\varepsilon_{\mathsf{RM}}(x)$.[4] The following result shows that this dependence is in fact tight.

**Theorem 3.2** (Lower bound for BoN-Alignment). *For any $\varepsilon_{\mathsf{RM}} \in (0, \frac{1}{4}]$ and $N \gtrsim 1$, there exists a problem instance with $\varepsilon_{\mathsf{RM}}(x) \leq \varepsilon_{\mathsf{RM}}$ and comparator policy $\pi^\star$ with $\mathcal{C}^{\pi^\star}(x) = \widetilde{O}(1)$, such that, for any $x \in \mathcal{X}$, BoN-Alignment has regret*

$$J(\pi^\star; x) - J(\widehat{\pi}_{\mathsf{BoN}}; x) \geq \widetilde{\Omega}\left( \sqrt{N \cdot \varepsilon_{\mathsf{RM}}^2} \right). \tag{7}$$

*Further, for all $N \in \mathbb{N}$, there exists a problem instance satisfying the same conditions such that BoN-Alignment has*

$$J(\pi^\star; x) - J(\widehat{\pi}_{\mathsf{BoN}}; x) \geq \widetilde{\Omega}\left( \varepsilon_{\mathsf{RM}}^{2/3} \right). \tag{8}$$

Eq. (7) shows that the regret indeed grows as $N$ is scaled, and is an algorithm-dependent lower bound that reflects the consequences of overfitting in BoN-Alignment. This is a serious concern since, when scaling $N$ in practice, it may be impossible to know the sample size beyond which overoptimization occurs, especially on a per-prompt basis. Then, building on Eq. (7), the bound in Eq. (8) shows that $\varepsilon_{\mathsf{RM}}^{2/3}(x)$ is the best possible error achievable by BoN-Alignment, which matches the upper bound in

___
[4]The bound in Eq. (6) is larger than the bound in Proposition 2.3 in the non-trivial regime where $(\mathcal{C}^{\pi^\star}(x) \cdot \varepsilon_{\mathsf{RM}}^2(x))^{1/2} \leq R_{\mathsf{max}}$.

Eq. (6). In other words, *irrespective of the amount of computation*, BoN-Alignment *may fail to achieve the optimal skyline for regret in Proposition 2.3.*

The suboptimality stems from the dual role of $N$, which serves as the regularizer, but is only a weak one at best. In particular, $N$ cannot be safely increased to sample better candidates without disproportionately increasing the risk of overfitting. This insight motivates the algorithms we develop in Section 4, which mitigate overoptimization through a more refined form of regularization and use of inference-time computation.

**Remark 3.3** (Proof technique)**.** *The lower bound in Eq. (7) is algorithm-specific, and the construction exposes the cost of overfitting to errors in $\widehat{r}$, specifically, for responses with small probability under $\pi_{\mathsf{ref}}$ that are drawn when $N$ is relatively large. To prove Eq. (8), we first leverage an information-theoretic argument showing that if $N \ll \mathrm{poly}\left(\mathcal{C}^{\pi^\star}(x), \frac{1}{\varepsilon_{\mathsf{RM}}^2(x)}\right)$, no algorithm can extract enough information to compete with $\pi^\star$. Combining this regime with the one in Eq. (7) yields Eq. (8).*

### 3.2. Stronger Guarantees under Uniform Coverage

Per Theorem 3.2, the regret of BoN-Alignment must scale with $\varepsilon_{\mathsf{RM}}^{2/3}(x)$ in general. However, it does enjoy tighter guarantees when a stronger form of coverage—the *uniform coverage coefficient*—is bounded: $\mathcal{C}_\infty^{\pi^\star}(x) := \sup_{y \in \mathcal{Y}} \frac{\pi^\star(y|x)}{\pi_{\mathsf{ref}}(y|x)}$.

**Theorem 3.4** (BoN-Alignment under uniform coverage)**.** *For any $x \in \mathcal{X}$ and comparator policy $\pi^\star$, if $N \geq \mathcal{C}_\infty^{\pi^\star}(x)$, the BoN-Alignment policy $\widehat{\pi}_{\mathsf{BoN}}$ satisfies*

$$J(\pi^\star; x) - J(\widehat{\pi}_{\mathsf{BoN}}; x)$$
$$\lesssim R_{\mathsf{max}} \cdot \exp(-N/\mathcal{C}_\infty^{\pi^\star}(x)) + \sqrt{N \cdot \varepsilon_{\mathsf{RM}}^2(x)}.$$

*In particular, as long as $N \asymp \mathcal{C}_\infty^{\pi^\star} \log(R_{\mathsf{max}}/\varepsilon_{\mathsf{RM}}(x))$,*

$$J(\pi^\star; x) - J(\widehat{\pi}_{\mathsf{BoN}}; x) \lesssim \sqrt{\mathcal{C}_\infty^{\pi^\star}(x) \cdot \varepsilon_{\mathsf{RM}}^2(x) \cdot \log(R_{\mathsf{max}}/\varepsilon_{\mathsf{RM}}(x))}.$$

When we are willing to pay for uniform coverage, the regret of BoN-Alignment scales as $\sqrt{\varepsilon_{\mathsf{RM}}^2(x)}$, which matches the statistical rate in the skyline of Proposition 2.3. This suggests that BoN may already be sufficient in some cases, at least when it is possible to tune $N$ optimally; we revisit this point empirically in Section 5. Generally, however, we expect that $\mathcal{C}_\infty^{\pi^\star}(x) \gg \mathcal{C}^{\pi^\star}(x)$, making Theorem 3.4 highly suboptimal relative to the skyline. For example, softmax policies, which are normalized exponentials of the logits, can have exponentially large $\mathcal{C}_\infty^{\pi^\star}$, while $\mathcal{C}^{\pi^\star}$ is $\widetilde{O}(1)$. This is (loosely) the case in the lower bound construction of Theorem 3.2, where $\pi_{\mathsf{ref}}$ is an exponential policy, and is upweighted by an exponential multiplier to form a $\pi^\star$ that is more sharply peaked on $r^\star$.

---

**Algorithm 2** InferenceTimePessimism

> **input:** Prompt $x$, reference policy $\pi_{\mathsf{ref}}$, reward model $\widehat{r}$, query size $N$, regularization coeff. $\beta > 0$.

1: Draw $\widehat{\mathcal{Y}}_N := (y_1, \ldots, y_N) \overset{\text{i.i.d.}}{\sim} \pi_{\mathsf{ref}}(\cdot \mid x)$.
2: Using ComputeNormConstant (Algorithm 3), compute normalization constant $\widehat{\lambda}(x)$ such that

$$\frac{1}{N} \sum_{y \in \widehat{\mathcal{Y}}_N} \mathsf{relu}\left(\beta^{-1}\left(\widehat{r}(x, y) - \widehat{\lambda}(x)\right)\right) = 1. \quad (9)$$

3: Sample resp. $y \sim \mathsf{RejectionSampling}_{N,M}(w; \pi_{\mathsf{ref}}, x)$ (Algorithm 5), where $M := \frac{R_{\mathsf{max}} - \widehat{\lambda}(x)}{\beta}$ and

$$w(y \mid x) := \mathsf{relu}\left(\beta^{-1}\left(\widehat{r}(x, y) - \widehat{\lambda}(x)\right)\right).$$

4: **return:** response $y$.

---

## 4. Inference-Time Pessimism

We now present InferenceTimePessimism, an optimal inference-time alignment method that implements a statistically sound regularizer purely at inference time. In contrast to BoN-Alignment, InferenceTimePessimism separates the scaling parameter ($N$) from the regularization parameter, and is thereby able to achieve the performance skyline in Proposition 2.3. It is, as a result, also *scaling-monotone*—as $N$ is increased, it does not overfit or lose performance. We first give an algorithm overview, then state the main theoretical guarantee.

### 4.1. The Inference-Time Pessimism Algorithm

The InferenceTimePessimism algorithm is displayed in Algorithm 2. For a regularization parameter $\beta > 0$, the algorithm is designed to sample from the distribution

$$\pi_\beta^\chi(y \mid x) := \pi_{\mathsf{ref}}(y \mid x) \cdot \mathsf{relu}(\beta^{-1}(\widehat{r}(x, y) - \lambda(x))), \quad (10)$$

where $\mathsf{relu}(z) := \max\{z, 0\}$, and $\lambda(x)$ is a normalization constant chosen such that

$$\sum_{y \in \mathcal{Y}} \pi_{\mathsf{ref}}(y \mid x) \cdot \mathsf{relu}(\beta^{-1}(\widehat{r}(x, y) - \lambda(x))) = 1. \quad (11)$$

The distribution in Eq. (10) is the exact solution to the $\chi^2$-*regularized reinforcement learning* objective: defining $D_{\chi^2}(\pi(x) \parallel \pi_{\mathsf{ref}}(x)) = \frac{1}{2} \mathbb{E}_{y \sim \pi_{\mathsf{ref}}(\cdot|x)}\left[\left(\frac{\pi(y|x)}{\pi_{\mathsf{ref}}(y|x)} - 1\right)^2\right] = \frac{1}{2}(\mathcal{C}^\pi(x) - 1)$ as the $\chi^2$-divergence, we have

$$\pi_\beta^\chi(x) = \underset{p \in \Delta(\mathcal{Y})}{\arg\max}\left\{\mathbb{E}_{y \sim p}[\widehat{r}(x, y)] - \beta \cdot D_{\chi^2}(p \parallel \pi_{\mathsf{ref}}(x))\right\}.$$

As we discuss in the sequel, the $\chi^2$-regularizer adapts to the uncertainty in the reward model $\widehat{r}(x, y)$, so that the regularized policy in Eq. (10) implements *pessimism in the face of uncertainty*, a principle from offline reinforcement learning

that carries strong guarantees (Jin et al., 2021; Rashidine-jad et al., 2021; Huang et al., 2024b). In particular, Huang et al. (2024b) showed that $\chi^2$-regularization at training time can achieve the skyline in Proposition 2.3; our method implements similar regularization but at inference-time, and attains those same guarantees.

The pessimistic $\chi^2$-regularization prevents the algorithm from overfitting to potentially misleading responses, e.g., those in the tail of the base distribution for which $\pi_{\mathsf{ref}}(y \mid x)$ is small and $\widehat{r}(x, y)$ is erroneously estimated to be large. The parameter $\beta > 0$ controls the degree of pessimism, with small values inducing a greedier policy, and large values inducing a conservative, heavy-tailed distribution over responses. For any choice of $\beta$, however, there is a set performance level the algorithm will never drop below, even as $N \to \infty$.

To (approximately) sample from the optimal $\chi^2$-regularized policy in Eq. (10), Algorithm 2 proceeds in two steps. First, since the normalization constant $\lambda(x)$ in Eq. (11) is unknown, the algorithm computes an approximate normalizer $\widehat{\lambda}(x)$. It draws $N$ responses $\widehat{\mathcal{Y}}_N := (y_1, \ldots, y_N) \overset{\text{i.i.d.}}{\sim} \pi_{\mathsf{ref}}(\cdot \mid x)$ and uses them to solve Eq. (9), an empirical approximation to Eq. (11). This is accomplished via the dynamic programming subroutine in ComputeNormConstant (Algorithm 3), in $O(N \log N)$ time. See Appendix C for details.

Next, InferenceTimePessimism generates samples from (approximately) the policy in Eq. (10) using classical *rejection sampling* (Von Neumann, 1963; Block & Polyanskiy, 2023); see Algorithm 5 in Appendix D. Given a rejection sampling threshold $M > 0$, the algorithm draws another set of responses $\widehat{\mathcal{Y}}_N = y_1, \ldots, y_N$. For each response $i$, it samples a Bernoulli random variable $\xi_i \sim \mathrm{Ber}\left( \frac{\mathsf{relu}(\beta^{-1}(\widehat{r}(x,y_i) - \widehat{\lambda}(x)))}{M} \wedge 1 \right)$, and returns $y_i$ as the final response if $\xi_i = 1$. By the heavy-tailed nature of the idealized $\chi^2$-regularized distribution in Eq. (10), a rejection threshold of $M \approx \frac{R_{\max}}{\beta}$ is sufficient. Accounting for both the normalization constant computation and rejection sampling phase, a total of $N = \widetilde{O}\left( \frac{R_{\max}}{\beta} \right)$ samples ensures that the response distribution of Algorithm 2 is a good approximation of Eq. (10).

## 4.2. Theoretical Guarantees

We now bound the regret for InferenceTimePessimism, which achieves the skyline rate of Proposition 2.3 with coverage coefficient $\mathcal{C}^{\pi^\star}(x)$.

**Theorem 4.1** (Guarantee for InferenceTimePessimism). *For any* $\beta > 0$ *and* $\varepsilon_{\mathsf{RM}}^2(x) \in (0, 1]$, *if* $N \geq c \cdot \frac{R_{\max}}{\beta} \log\left( \frac{R_{\max}}{\beta \cdot \varepsilon_{\mathsf{RM}}(x)} \right)$ *for a sufficiently large constant* $c$,

InferenceTimePessimism *satisfies*

$$J(\pi^\star; x) - J(\widehat{\pi}_{\mathsf{Pes}}; x) \tag{12}$$
$$\lesssim \sqrt{\mathcal{C}^{\pi^\star}(x) \cdot \varepsilon_{\mathsf{RM}}^2(x)} + \beta \cdot \mathcal{C}^{\pi^\star}(x) + \beta^{-1} \cdot \varepsilon_{\mathsf{RM}}^2(x).$$

*Setting* $\beta \asymp \sqrt{\frac{\varepsilon_{\mathsf{RM}}^2(x)}{\mathcal{C}^{\pi^\star}(x)}}$, *as long as* $N \gtrsim \widetilde{\Omega}\left( \sqrt{\frac{R_{\max}^2 \cdot \mathcal{C}^{\pi^\star}(x)}{\varepsilon_{\mathsf{RM}}^2(x)}} \right)$, *we have*

$$J(\pi^\star; x) - J(\widehat{\pi}_{\mathsf{Pes}}; x) \lesssim \sqrt{\mathcal{C}^{\pi^\star}(x) \cdot \varepsilon_{\mathsf{RM}}^2(x)}. \tag{13}$$

On the statistical side, for any choice of the regularization parameter $\beta > 0$, the regret bound in Eq. (12) balances overoptimization (reflected in the term $\beta^{-1} \cdot \varepsilon_{\mathsf{RM}}^2(x)$) with bias (reflected in the term $\beta \cdot \mathcal{C}^{\pi^\star}(x)$). Choosing $\beta$ to balance these terms leads to the regret bound in Eq. (13), which matches the lower bound in Proposition 2.3 up to absolute constants, showing that InferenceTimePessimism is *regret-optimal*.

Computationally, achieving the regret bound in Eq. (12) requires $N \geq \widetilde{\Omega}\left( \frac{R_{\max}}{\beta} \right)$. In contrast to Best-of-N (Theorem 3.1), InferenceTimePessimism is robust to overoptimization, in the sense that for any fixed $\beta > 0$, the guarantee in Eq. (12) holds *for all $N$ sufficiently large*, and there is no risk of dropping below the bound on the right-hand side of Eq. (12) as we scale computation; we refer to this property as *scaling-monotonicity*.

**Lower bounds and compute-optimality.** InferenceTimePessimism requires $N \geq \widetilde{\Omega}\left( \sqrt{\mathcal{C}^{\pi^\star}(x)} \cdot \frac{R_{\max}}{\varepsilon_{\mathsf{RM}}(x)} \right)$ samples to achieve the optimal regret bound Eq. (13), where $\beta \asymp \sqrt{\frac{\varepsilon_{\mathsf{RM}}^2(x)}{\mathcal{C}^{\pi^\star}(x)}}$. The following result shows that the $\varepsilon_{\mathsf{RM}}(x)^{-1}$ dependence is necessary for any algorithm in the sample-and-evaluate framework.

**Theorem 4.2** (Query complexity lower bound). *For any* $\varepsilon_{\mathsf{RM}} \in (0, 1/4]$ *and* $N \lesssim \frac{1}{\varepsilon_{\mathsf{RM}}}$, *there exists a problem instance with* $R_{\max} = 1$, $\varepsilon_{\mathsf{RM}}(x) \leq \varepsilon_{\mathsf{RM}}$, *and a comparator policy* $\pi^\star$ *with* $\mathcal{C}^{\pi^\star}(x) = \mathcal{C}^{\pi^\star} = \widetilde{O}(1)$ *such that any algorithm* $\mathcal{A}$ *using at most $N$ sample-and-evaluate queries must have*

$$J(\pi^\star) - J(\widehat{\pi}_{\mathcal{A}}) = \widetilde{\Omega}(1/N).$$

This result implies that *any* algorithm achieving the lower bound in Proposition 2.3 requires $N \gtrsim \frac{1}{\varepsilon_{\mathsf{RM}}(x)}$, which is matched by the guarantee for InferenceTimePessimism (Theorem 4.1). Moreover, when $N \lesssim \frac{1}{\varepsilon_{\mathsf{RM}}(x)}$, the regret can be even larger than the reward estimation error $\varepsilon_{\mathsf{RM}}$, since $\frac{1}{N} \geq \varepsilon_{\mathsf{RM}}$. We remark that the computational cost here is comparable (though slightly larger) than the cost of achieving the sub-optimal bound in Eq. (6) using Best-of-N (roughly $N \gtrsim \varepsilon_{\mathsf{RM}}^{-1}(x)$ versus $N \gtrsim \varepsilon_{\mathsf{RM}}^{-2/3}(x)$).

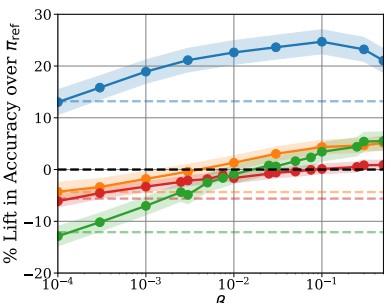 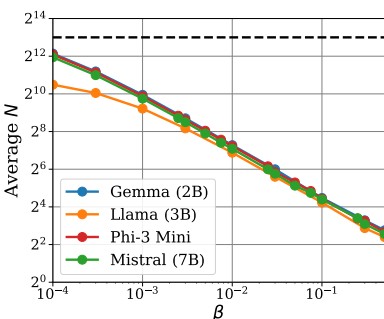 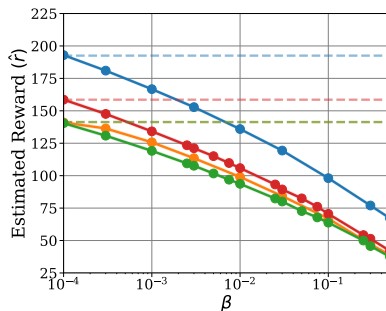

*Figure 2.* Compute-normalized comparison of `InferenceTimePessimism` (solid lines) and `BoN-Alignment` (dashed lines) on GSM8K with OASST as $\widehat{r}$, as a function of regularization parameter $\beta$. We run `BoN-Alignment` with $N = 2^{13}$ and run `InferenceTimePessimism` until rejection sampling accepts (capped to $N = 2^{13}$). **Left**: `InferenceTimePessimism` can improve significantly over `BoN-Alignment` for large $N$ due to the reward overoptimization. **Center**: Number of responses required for `InferenceTimePessimism` to accept an answer decreases as $\beta$ increases, as predicted by our theory. **Right**: Estimated reward $\widehat{r}$ for `InferenceTimePessimism` decreases as $\beta$ increases.

**Parameter tuning and practicality.** As presented in Algorithm 2, `InferenceTimePessimism` has two parameters, $\beta$ and $N$, compared to the single parameter $N$ used by `BoN-Alignment`. This separation enables the optimal and scale-monotonic guarantees for `InferenceTimePessimism`, and we also view it as a beneficial feature from a practical perspective: by using two parameters, we achieve a clear separation between tuning the computational budget (through $N$) and tuning the statistical performance (through $\beta$). While to achieve Eq. (13) $\beta$ must be chosen based on potentially unknown parameters ($\mathcal{C}^{\pi^\star}(x)$ and $\varepsilon_{\mathsf{RM}}^2(x)$), this represents a meaningful improvement over `BoN-Alignment`, which cannot be tuned to achieve Eq. (13) even when these parameters are known (Theorem 3.2). This is because `BoN-Alignment` conflates computational and statistical considerations in its single parameter $N$.

Empirically, we find that for any fixed choice of $\beta$, `InferenceTimePessimism` is robust to overfitting when computation and sample size is increased, with performance essentially monotonic as $N$ grows (Figure 10), as predicted by the guarantee in Eq. (12). Because of this robustness, it is easier to tune the parameter $\beta$, which we also find is important to achieve strong performance; further discussion and practical guidance is provided in Section 5. Altogether, we believe it is most natural to interpret `InferenceTimePessimism` as a "single-parameter" algorithm where only $\beta$ needs to be tuned, and $N$ is as large as the computational budget allows.

**Overview of analysis.** The proof of Theorem 4.1, and has three parts. First, we show that the idealized $\chi^2$-regularized distribution in Eq. (10) achieves the regret bound in Eq. (12). This follows the same reasoning as the analysis of training-time interventions based on $\chi^2$-regularization in Huang et al. (2024b), and uses the property that for any function $\Delta(x, y)$ (we use $\Delta(x, y) = |\widehat{r}(x, y) - r^\star(x, y)|$)

and policy $\pi$, $\mathbb{E}_\pi[\Delta(x, y)] \lesssim$

$$\sqrt{(1 + D_{\chi^2}(\pi(x) \,\|\, \pi_{\mathsf{ref}}(x))) \cdot \mathbb{E}_{y \sim \pi_{\mathsf{ref}}(x)}[\Delta^2(x, y)]}.$$

Of course, we cannot sample from $\pi_\beta^\chi$ itself because the distribution depends on the "true" normalization constant $\lambda(x)$ in Eq. (11). To address this, we prove a robustness result showing that, given a $\widehat{\lambda}$ that approximately normalizes the distribution in Eq. (10), the policy $\widetilde{\pi}_\beta^\chi(y \mid x) = \frac{\pi_{\mathsf{ref}}(y|x) \cdot \mathsf{relu}\left(\beta^{-1}(\widehat{r}(x,y) - \widehat{\lambda})\right)}{\sum_{y' \in \mathcal{Y}} \pi_{\mathsf{ref}}(y'|x) \cdot \mathsf{relu}\left(\beta^{-1}(\widehat{r}(x,y') - \widehat{\lambda})\right)}$ achieves a regret bound that matches Eq. (12) up to absolute constants. From here, a concentration argument implies that the $\widehat{\lambda}(x)$ computed in Eq. (9) is an approximate normalizer whenever $N \gtrsim \frac{R_{\max}}{\beta}$, with high probability.

Finally, we leverage analysis for rejection sampling to show that, as long as $N \gtrsim \max_{y \in \mathcal{Y}} \frac{\widetilde{\pi}_\beta^\chi(y|x)}{\pi_{\mathsf{ref}}(y|x)} \log(1/\delta)$, the rejection sampling procedure will terminate and return $y \sim \widetilde{\pi}_\beta^\chi(\cdot \mid x)$ with probability at least $1 - \delta$. Critically, due to the heavy-tailed nature of the $\chi^2$-regularizer, this density ratio is bounded as $\frac{\widetilde{\pi}_\beta^\chi(y|x)}{\pi_{\mathsf{ref}}(y|x)} \lesssim \frac{R_{\max}}{\beta}$, which yields the claimed query complexity bound. This observation highlights an important computational benefit of $\chi^2$-regularization that goes beyond its statistical benefits, illustrated below.

**Remark 4.3** (Comparison to KL-regularization). *Liu et al. (2023); Li et al. (2024a) use rejection sampling to simulate samples from the KL-regularized distribution:*

$$\pi_\beta^{\mathsf{KL}}(y \mid x) := \arg\max_{p \in \Delta(\mathcal{Y})} \{\mathbb{E}_{y \sim p}[\widehat{r}(x, y)] - \beta \cdot D_{\mathsf{KL}}(p \,\|\, \pi_{\mathsf{ref}}(x))\},$$

*which satisfies This has two issues. First, as shown by Huang et al. (2024b), this distribution can fail to achieve the guarantee in Eq. (13), no matter how $\beta$ is chosen. Second, the density ratio is exponential in general, i.e. $\frac{\pi_\beta^{\mathsf{KL}}(y|x)}{\pi_{\mathsf{ref}}(y|x)} \geq \exp\left(\frac{R_{\max}}{\beta}\right)$, which means that $N \gtrsim \exp\left(\frac{R_{\max}}{\beta}\right)$ sample-and-evaluate queries are required to simulate it with rejection sampling.*

*Table 1.* Performance of $\pi_{\text{ref}} =$ Phi-3-Mini (% Lift in Accuracy over $\pi_{\text{ref}}$).

| Task | OASST | GEMMA-RM | LLAMA-RM | ARMO-RM |
|------|-------|----------|----------|---------|
| GSM8K (Pessimism) | $0.87 \pm 0.94$ | $5.61 \pm 0.92$ | $12.03 \pm 0.90$ | $12.44 \pm 0.88$ |
| GSM8K (BoN) | $-5.61 \pm 1.13$ | $4.10 \pm 1.08$ | $12.12 \pm 0.95$ | $13.12 \pm 0.96$ |
| MMLU (Pessimism) | $-0.71 \pm 4.76$ | $14.29 \pm 5.24$ | $24.48 \pm 5.55$ | $21.12 \pm 5.58$ |
| MMLU (BoN) | $-5.61 \pm 5.50$ | $7.57 \pm 6.05$ | $25.41 \pm 6.10$ | $16.20 \pm 6.42$ |
| MATH (Pessimism) | $3.41 \pm 3.84$ | $18.36 \pm 4.02$ | $41.47 \pm 4.30$ | $26.72 \pm 3.98$ |
| MATH (BoN) | $3.32 \pm 3.98$ | $15.36 \pm 4.27$ | $41.74 \pm 4.35$ | $21.72 \pm 4.19$ |

## 5. Experiments

In this section, we complement our theoretical results with a suite of experiments that investigate the practicality of InferenceTimePessimism, and compare its performance to that of BoN-Alignment. We consider three standard tasks: the test split of the elementary school math dataset GSM8K (Cobbe et al., 2021); math and chemistry splits of MMLU (Hendrycks et al., 2020), and the test split of the advanced math problems dataset MATH (Hendrycks et al., 2021). We also present a preliminary study with AlpacaEval-2.0 (Li et al., 2023a) in Appendix B. We consider four reward models in increasing order of size: OASST (1.4B) (Köpf et al., 2024), GEMMA-RM (2B) (Dong et al., 2023), LLAMA-RM (3B) (Yang et al., 2024c), and ARMO-RM (7B) (Wang et al., 2024a). Finally, for each task we consider a subset of four policies for the base model: GEMMA-2-2B (Team et al., 2024), LLAMA-3-3B (Dubey et al., 2024), Mistral-7B (Jiang et al., 2023), and Phi-3-Mini (Abdin et al., 2024). For each task-policy-reward triplet, and each prompt, we generate a large number of responses (20K) and conduct experiments by bootstrapping subsamples of this large set. In all cases, we reuse the same samples for normalization constant estimation as for the rejection sampling step.

To produce Figure 1, we compare the performance of InferenceTimePessimism (with tuned $\beta$) and BoN-Alignment for a range of $N$ in terms of both true reward (accuracy) and estimated reward ($\hat{r}$ is OASST) on GSM8K, defaulting to BoN-Alignment if InferenceTimePessimism does not terminate. As we scale $N$, we observe the characteristic dip (e.g., Gao et al. (2023)) in accuracy of BoN-Alignment in the left panel; from the right panel, we can infer that this is caused by reward overoptimization. On the other hand, we find that the accuracy of InferenceTimePessimism monotonically increases with $N$, as predicted by Theorem 4.1. To further investigate the effect that regularization has on InferenceTimePessimism, in Figure 2, we fix a compute budget of $N = 2^{13}$ and compare the performance of BoN-Alignment to InferenceTimePessimism as we vary the regularization parameter $\beta$. As we increase $\beta$, we see that InferenceTimePessimism leads to improved true reward (Left), a smaller computational budget (Center) and significantly less reward overoptimization (Right).

Table 1 collects similar results across all tasks GSM8K, MMLU, and MATH and reward models OASST, GEMMA-RM, LLAMA-RM, and ARMO-RM, with Phi-3-Mini as the base policy $\pi_{\text{ref}}$. We use a fixed computational budget, and compare the naïve BoN-Alignment for $N = 2^{13}$ with InferenceTimePessimism for the best $\beta$ (see Appendix B for further details). Here, we find that InferenceTimePessimism tends to have higher average performance than BoN-Alignment, although in many instances this difference is not statistically significant; we suspect this is because, when $r^\star$ is binary (as it is in all tasks we use), there is no separation between $\mathcal{C}^{\pi^\star}$ and $\mathcal{C}^{\pi^\star}_\infty$ when $\pi^\star$ is the optimal policy (uniform over the set of correct answers); thus, we are in the regime where Theorem 3.4 predicts near-optimal performance for BoN. However, a different story may emerge under more refined evaluation metrics, such as the correctness of proofs in addition to the final answer. Here we should expect $\mathcal{C}^{\pi^\star}_\infty \gg \mathcal{C}^{\pi^\star}$, and we leave evaluation in more realistic environments to future work. For the sake of space, we defer further empirical results to Appendix B, including plots and tables analogous to Figures 1 and 2 and Table 1 for other policies, tasks, and rewards (Appendix B.2) as well as additional experiments (Appendix B.3).

## 6. Conclusion

Our results reveal the interplay between coverage, scaling, and optimality in inference-time alignment, and highlight the benefits of deliberate compute scaling. Beyond providing optimal algorithms (InferenceTimePessimism) and insights into the performance of BoN-Alignment, our framework can serve as starting point toward a foundational understanding of inference-time computation more broadly. In particular, our work raises a number of interesting directions for future research, including moving beyond the worst-case assumption on $\hat{r}$ and designing inference-aware training procedures that optimize for InferenceTimePessimism at generation time.

## Impact Statement

This paper presents work whose goal is to advance the field of Machine Learning. There are many potential societal consequences of our work, none which we feel must be specifically highlighted here.

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

# Contents of Appendix

# Part I

# Additional Discussion and Results

## A. Additional Related Work

In this section, we discuss additional related work in greater detail.

**Theoretical analysis of inference-time alignment.** Inference-time alignment has received limited theoretical investigation so far. Most notably, various works have analyzed specific properties of the Best-of-N alignment algorithm (Yang et al., 2024b; Beirami et al., 2024; Mroueh, 2024) such as tradeoffs between reward and KL-divergence, but do not ultimately provide guarantees on downstream performance in the presence of mismatch between the estimated reward model and true reward. Our theoretical framework—which abstracts the role of the base policy through sample-and-evaluate access—is inspired by Huang et al. (2024a), who used a similar framework to give guarantees for the complementary problem of language model self-improvement, but our specific problem formulation and techniques are quite different.

**Empirical algorithms for inference-time alignment.** Empirically, the Best-of-N alignment heuristic (Stiennon et al., 2020; Nakano et al., 2021; Touvron et al., 2023; Gao et al., 2023; Eisenstein et al., 2023; Mudgal et al., 2024) is perhaps the most widely used inference-time alignment heuristic; specific works that have observed the overoptimization phenomenon for Best-of-N include Gao et al. (2023, Figure 1), Chow et al. (2024, Figure 3), Frick et al. (2024, Figure 7), and Stroebl et al. (2024). There are also other algorithms based on more sophisticated variants of Best-of-N or other techniques such as rejection sampling (Liu et al., 2023; Chen et al., 2024; Chakraborty et al., 2024; Xu et al., 2024; Shi et al., 2024; Qiu et al., 2024; Jinnai et al., 2024; Zhao et al., 2024), though few of these works are explicitly designed to address these issue of over-optimization. For example, most algorithms based on inference-time search, such as Monte-Carlo Tree Search (MCTS) and relatives (Feng et al.; Yao et al., 2024; Zhang et al., 2024), are designed with the complementary goal of maximizing a fixed reward function of interest given exact access, and do not account for overoptimization or mismatch between the reward function and task performance. Specific algorithms that make use of rejection sampling include Liu et al. (2023); Li et al. (2024a); Xiong et al. (2024); Zhao et al. (2024); Khaki et al. (2024), though the specific distributions these works aim to sample from are quite different from that used in `InferenceTimePessimism`. See also Welleck et al. (2024) for a survey of inference-time algorithms.

Other related but complementary line of work include (1) distilling inference-time procedures into policies, thereby giving training time procedures (Amini et al., 2024; Sessa et al., 2024; Gui et al., 2024; Pace et al., 2024), and (2) designing "inference-aware" training procedures which change the training process to optimize performance of downstream inference-time such as Best-of-N (Balashankar et al., 2024; Chow et al., 2024).

### A.1. Connection to Offline (Training-Time) Alignment

As discussed in Section 1.2, our problem formulation and algorithms are closed related to a growing body of research on theoretical algorithms for *offline alignment* (Zhu et al., 2023; Zhan et al., 2023a; Li et al., 2023b; Xiong et al., 2024; Liu et al., 2024b; Cen et al., 2024; Fisch et al., 2024; Ji et al., 2024; Huang et al., 2024b; Rashidinejad & Tian, 2024), which give training-time interventions that enjoy robustness to reward model overoptimization under various notions of coverage. In particular, our `InferenceTimePessimism` algorithm can be viewed as implementing an "idealized" version of the $\chi^2$-regularized RLHF algorithms introduced by Huang et al. (2024b) at inference-time.

Our formulation of inference-time alignment can be viewed as a variant of the offline alignment problem that abstracts away the process of training the reward model. Rather than concerning ourselves with the details of training $\widehat{r}$ from $\mathcal{D}_{\mathsf{pref}}$ to minimize Eq. (3), we take $\widehat{r}$ as a given (in the process, abstracting away the dataset $\mathcal{D}_{\mathsf{pref}}$), and ask how to achieve the best possible regret on a *per-instance* basis, both with respect to the reward model $\widehat{r}$ itself, and with respect to the prompt $x \in \mathcal{X}$ (which is arbitrary and fixed, rather than assumed to be i.i.d. as $x \sim \rho$). Naturally, our algorithms and analyses can be combined with any reward estimation procedure that minimizes $\mathbb{E}_{x \sim \rho}\left[\varepsilon_{\mathsf{RM}}^2(x)\right]$ from $\mathcal{D}_{\mathsf{pref}}$ to derive end-to-end sample complexity guarantees for offline alignment.

**Online alignment.** A complementary line of theoretical research which is somewhat less related to our work studies alignment with *online feedback* (Xu et al., 2020; Novoseller et al., 2020; Pacchiano et al., 2021; Wu & Sun, 2023; Zhan et al., 2023b; Chen et al., 2022; Wang et al., 2023; Du et al., 2024; Das et al., 2024; Ye et al., 2024; Xie et al., 2024; Cen

et al., 2024; Xiong et al., 2024; Gao et al., 2024; Chang et al., 2024; Song et al., 2024), where feedback from the true reward model $r^\star(x, y)$ is available.

## B. Further Empirical Results

In this section, we expand on the experiments discussed in Section 5. We begin by providing a complete description of our expeimical setup in Appendix B.1, before proceeding to expand the breadth of empirical results reported in the main body to more policies, tasks, and reward models in Appendix B.2. We continue by conducting a further investigation into the robustness of InferenceTimePessimism to the regularization parameter $\beta$ as well as a distributional study of the estimated rewards $\widehat{r}$ sampled from $\pi_{\text{ref}}$ in Appendix B.3. Finally, we present preliminary results for the AlpacaEval-2.0 task in Appendix B.4.

### B.1. Further Experimental Details

As we summarized in the main text, we conduct an extensive empirical suite by considering many tasks, reference policies, and estimated reward models. We now detail each of these in turn. The four tasks we consider are the following:

1. GSM8K: We consider the test split of the popular grade-school math dataset introduced in Cobbe et al. (2021). This dataset consists of about 1K short math word problems. We prompt all of our policies with Chain of Thought (CoT) prompting (Wei et al., 2022) but do not include any example demonstrations, i.e., we are zero-shot. We measure correctness in the sense that we assign $r^\star(x, y) = 1$ if the resulting policy gets the correct mathematical answer and $r^\star(x, y) = 0$ otherwise.

2. MMLU: We consider the college math and college chemistry splits of the MMLU dataset introduced in Hendrycks et al. (2020). This dataset consists about 100 questions each of math and chemistry at the college level with multiple choice answers. Again, we use CoT with zero-shot prompting for all policies. Correctness is measured in the sense that the resulting policy gets the correct multiple-choice answer.

3. MATH: We consider the randomly sampled set of 512 questions from the test split of the MATH dataset introduced in Hendrycks et al. (2021). This dataset consists of hard mathematics problems. As above, we use CoT and zero-shot prompting, with correctness measured in the sense that the resulting policy gets the correct mathematical answer.

4. AlpacaEval-2.0: For a small subset of our policies and rewards, we consider the AlpacaEval-2.0 task introduced in Li et al. (2023a), with 128 randomly sampled questions. This task is a challenging LM benchmark where we compare a policy's generation to that of a benchmark LM, and define $r^\star$ according to *win rate* against an evaluator LM. In order to collect a denser signal, we compare win rate against generations sampled from $\pi_{\text{ref}}$ for each $\pi_{\text{ref}}$ we evaluate. We use GPT-4o-mini (OpenAI, 2024a) as our evaluator.

For each of our tasks, we consider a subset of the following four reward models for use as the estimated reward $\widehat{r}$:

1. OASST: the OpenAssistant reward model based on Pythia-1.4b (Köpf et al., 2024).

2. GEMMA-RM: a reward model based on Gemma-2-2b (Dong et al., 2023).

3. LLAMA-RM: a reward model based on Llama-3-3b (Yang et al., 2024c).

4. ARMO-RM: a reward model based on Llama-3-8b (Wang et al., 2024a).

Finally, we consider the following policies for $\pi_{\text{ref}}$:

1. GEMMA-2-2B: the Gemma-2-2b model introduced in Team et al. (2024).

2. LLAMA-3-3B: the Llama-3-3b model introduced in Dubey et al. (2024).

3. Mistral-7B: the Mistral-7b model introduced in Jiang et al. (2023).

4. Phi-3-Mini: the Phi-3-mini model (3.8b parameters) introduced in Abdin et al. (2024).

5. Phi-3-Small: the Phi-3-small model (7b parameters) introduced in Abdin et al. (2024).

In all of our experiments, for each prompt in each task and each policy, we generate about 20K responses sampled with temperature 1 from the chosen $\pi_{\text{ref}}$. For a given number $M$ of replicates ($M = 50$ in all tasks except for AlpacaEval-2.0, where $M = 5$ due to resource constraints), we then bootstrap $M$ subsets of $N$ samples each from this large set and run our

algorithm on these subsampled responses. We define the *accuracy* of a given algorithm on a prompt as the average number of correct answers produced over the number of replicates; an exception to this is AlpacaEval-2.0, where we measure the *win rate* according to the evaluator LM. The reported accuracy of a policy is given by the average accuracy over all prompts in the task, and the standard error is estimated by marginalizing over the prompts in the task. As stated in the main body, in all cases for fixed $N$, we use the same $N$ samples to estimate the normalization constant (Algorithm 3) as we do to run the rejection sampling.

In what follows, we first present additional plots for the experiments described in Section 5, omitted from the main body for the sake of space (Appendix B.2), then describe results of additional experiments (Appendices B.3 and B.4).

### B.2. Results for Further Policies and Tasks

We complement Figures 1 and 2 as well as Table 1 with analogous figures and tables for the remaining tasks, reward models, and policies described above. First, we investigate how the compute budget $N$ affects performance and estimated reward of the response in GSM8K (Figure 3), MMLU (Figure 4), and MATH (Figure 5). In all cases, we see that InferenceTimePessimism is essentially monotonic in compute budget, as predicted by our theory. In some cases, we see *monotonicity* in the performance of BoN-Alignment, which is consistent with the observation (e.g., Figure 9) that some (task, policy) pairs appear to be more in-distribution for some rewards (such as ARMO-RM) than others; for such cases, we expect BoN-Alignment to perform well.

We also display the effect of the regularization parameter $\beta$ on performance, average compute, and estimated $\widehat{r}$ for each of GSM8K (Figure 6), MMLU (Figure 7), and MATH (Figure 8); we again compare our performance to BoN-Alignment with the same compute budget of $N = 2^{13}$. To be precise, we fix $N$ and compare the naïve BoN-Alignment approach with this large $N$ to InferenceTimePessimism where we use all $N$ samples to estimate the normalization constant and then run rejection sampling until acceptance for each fixed $\beta$ and prompt; in this case all prompts across all tasks, policies, reward models, and $\beta$'s terminate within the $N$ samples. As discussed in Section 5, increasing $\beta$ leads to a smaller average required responses before rejection sampling terminates as well as a smaller estimated reward $\widehat{r}$.

Finally, we display analogues of Table 1 for the remaining policies we consider: For each of Phi-3-Small (Table 2), Mistral-7B (Table 3), and LLAMA-3-3B (Table 4), we compare the performance of compute-normalized BoN-Alignment with $N = 2^{13}$ to InferenceTimePessimism on GSM8K, MMLU, and MATH (with the exception of LLAMA-3-3B, where we only consider the first two tasks) for our four reward models. We continue to find that the performance of BoN-Alignment with properly tuned $N$ and InferenceTimePessimism is similar, which we believe is caused by the fact that $\mathcal{C}^{\pi^\star} = \mathcal{C}_\infty^{\pi^\star}$ when $r^\star$ is binary and $\pi^\star$ is uniform over the set of correct answers; by Theorem 3.4, BoN-Alignment will perform near-optimally in this regime.

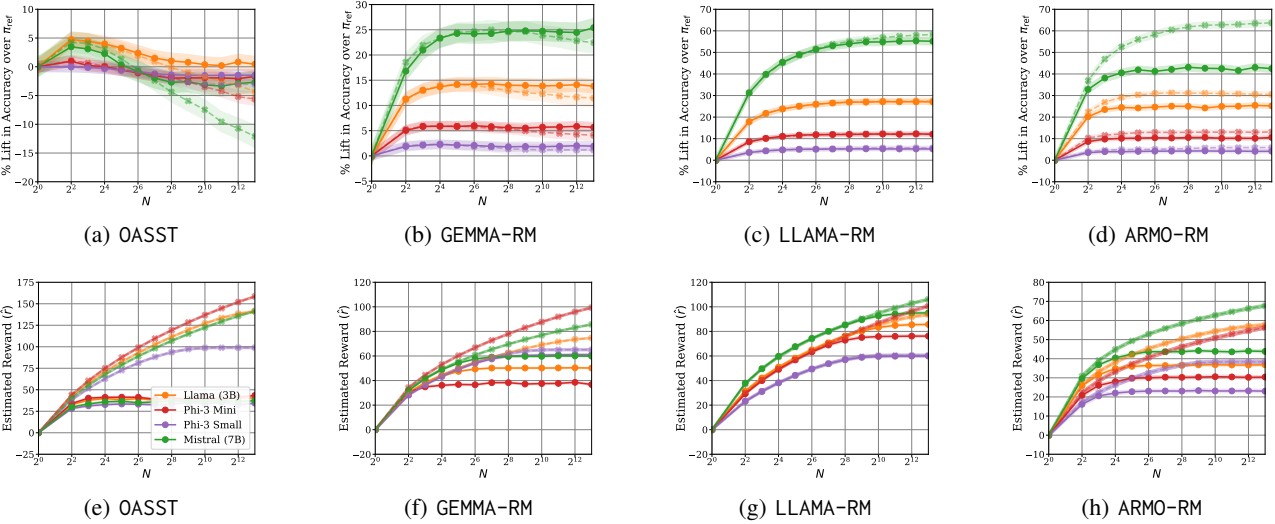

(a) OASST     (b) GEMMA-RM     (c) LLAMA-RM     (d) ARMO-RM

(e) OASST     (f) GEMMA-RM     (g) LLAMA-RM     (h) ARMO-RM

*Figure 3.* Comparison of InferenceTimePessimism (solid lines) and BoN-Alignment (dashed lines) in accuracy and estimated reward $\widehat{r}$ for GSM8K for four reward models and choices of $\pi_{\text{ref}}$.

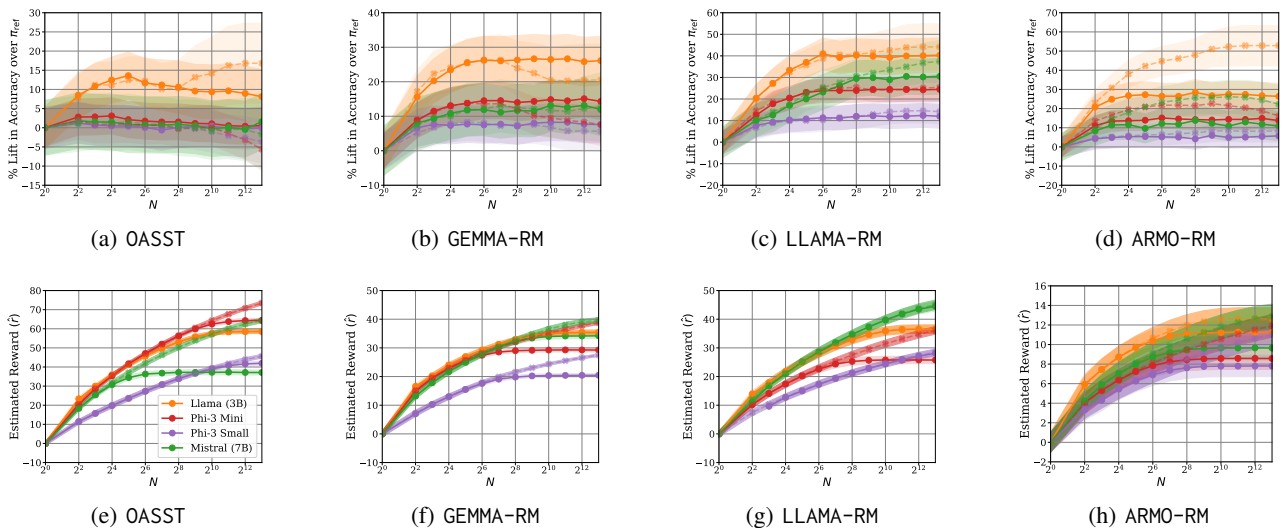

*Figure 4.* Comparison of `InferenceTimePessimism` (solid lines) and `BoN-Alignment` (dashed lines) in accuracy and estimated reward $\hat{r}$ for MMLU for four reward models and choices of $\pi_{\mathsf{ref}}$.

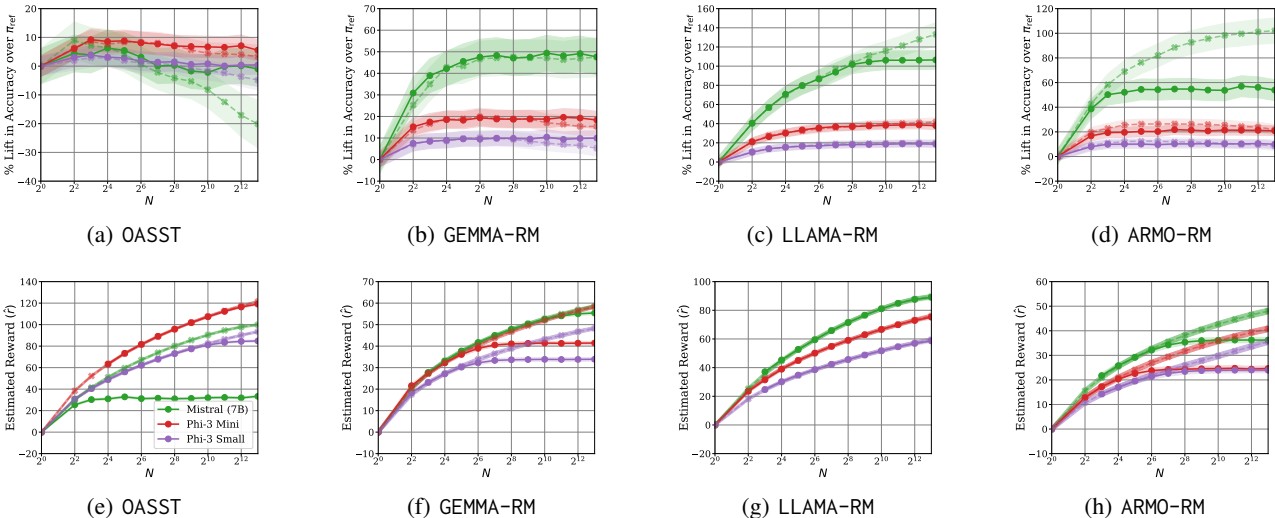

*Figure 5.* Comparison of `InferenceTimePessimism` (solid lines) and `BoN-Alignment` (dashed lines) in accuracy and estimated reward $\hat{r}$ for MATH for four reward models and choices of $\pi_{\mathsf{ref}}$.

*Table 2.* Performance of $\pi_{\mathsf{ref}} = $ `Phi-3-Small` (% Lift in Accuracy over $\pi_{\mathsf{ref}}$).

| Task | OASST | GEMMA-RM | LLAMA-RM | ARMO-RM |
|------|-------|----------|----------|---------|
| GSM8K (Pessimism) | $0.25 \pm 0.88$ | $1.29 \pm 0.97$ | $5.43 \pm 0.93$ | $4.79 \pm 0.84$ |
| GSM8K (BoN) | $-3.06 \pm 1.21$ | $1.19 \pm 1.08$ | $5.71 \pm 0.95$ | $5.87 \pm 0.93$ |
| MMLU (Pessimism) | $-2.18 \pm 5.37$ | $7.61 \pm 5.31$ | $14.47 \pm 5.60$ | $6.74 \pm 5.52$ |
| MMLU (BoN) | $-3.65 \pm 5.66$ | $5.67 \pm 5.76$ | $14.12 \pm 5.64$ | $8.27 \pm 5.84$ |
| MATH (Pessimism) | $-1.93 \pm 3.22$ | $9.94 \pm 3.40$ | $20.23 \pm 3.54$ | $12.71 \pm 3.37$ |
| MATH (BoN) | $-4.85 \pm 3.45$ | $5.27 \pm 3.66$ | $19.99 \pm 3.61$ | $8.39 \pm 3.56$ |

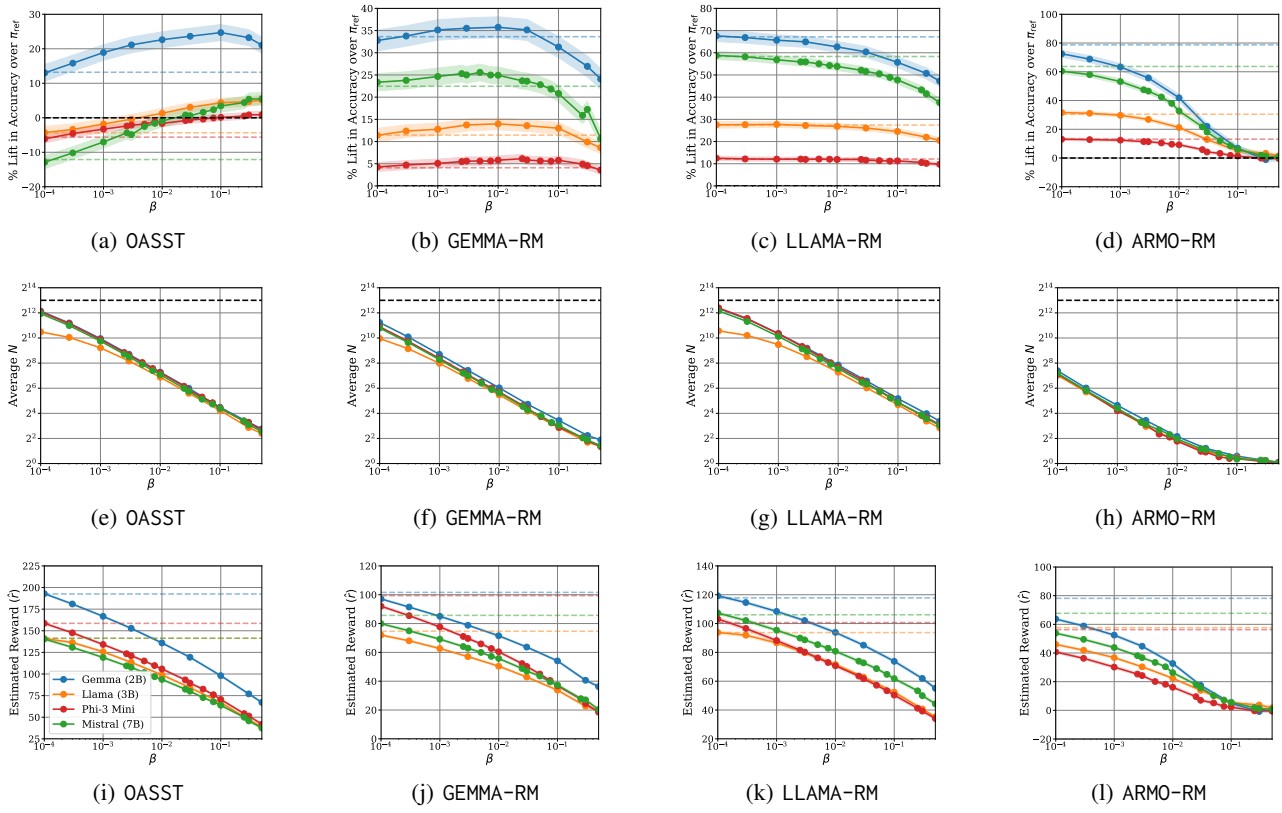

*Figure 6.* Compute-normalized comparison for $N = 2^{13}$ between `BoN-Alignment` and `InferenceTimePessimism` on GSM8K for four reward models and choices of $\pi_{\mathsf{ref}}$, as a function of regularization $\beta$.

*Table 3.* Performance of $\pi_{\mathsf{ref}} = $ `Mistral-7B` (% Lift in Accuracy over $\pi_{\mathsf{ref}}$).

| Task | OASST | GEMMA-RM | LLAMA-RM | ARMO-RM |
|---|---|---|---|---|
| GSM8K (Pessimism) | $4.15 \pm 1.77$ | $25.41 \pm 1.87$ | $57.30 \pm 1.77$ | $53.19 \pm 1.77$ |
| GSM8K (BoN) | $-12.10 \pm 1.96$ | $\mathbf{22.46 \pm 2.08}$ | $\mathbf{58.31 \pm 1.86}$ | $\mathbf{63.69 \pm 1.76}$ |
| MMLU (Pessimism) | $1.28 \pm 7.26$ | $14.56 \pm 7.66$ | $35.01 \pm 9.16$ | $24.63 \pm 8.21$ |
| MMLU (BoN) | $-1.70 \pm 8.84$ | $12.71 \pm 9.92$ | $37.43 \pm 9.70$ | $22.23 \pm 9.89$ |
| MATH (Pessimism) | $10.32 \pm 6.51$ | $46.15 \pm 8.43$ | $129.58 \pm 11.55$ | $91.41 \pm 9.42$ |
| MATH (BoN) | $-20.13 \pm 8.26$ | $47.49 \pm 9.12$ | $133.13 \pm 12.15$ | $102.10 \pm 10.32$ |

*Table 4.* Performance of $\pi_{\mathsf{ref}} = $ `LLAMA-3-3B` (% Lift in Accuracy over $\pi_{\mathsf{ref}}$).

| Task | OASST | GEMMA-RM | LLAMA-RM | ARMO-RM |
|---|---|---|---|---|
| GSM8K (Pessimism) | $5.20 \pm 1.33$ | $14.38 \pm 1.32$ | $27.54 \pm 1.28$ | $29.66 \pm 1.18$ |
| GSM8K (BoN) | $-4.35 \pm 1.94$ | $11.45 \pm 1.78$ | $27.43 \pm 1.52$ | $30.49 \pm 1.44$ |
| MMLU (Pessimism) | $16.82 \pm 10.21$ | $21.77 \pm 9.89$ | $46.55 \pm 10.49$ | $46.87 \pm 7.94$ |
| MMLU (BoN) | $16.82 \pm 10.44$ | $20.48 \pm 10.42$ | $44.13 \pm 10.71$ | $52.86 \pm 10.54$ |

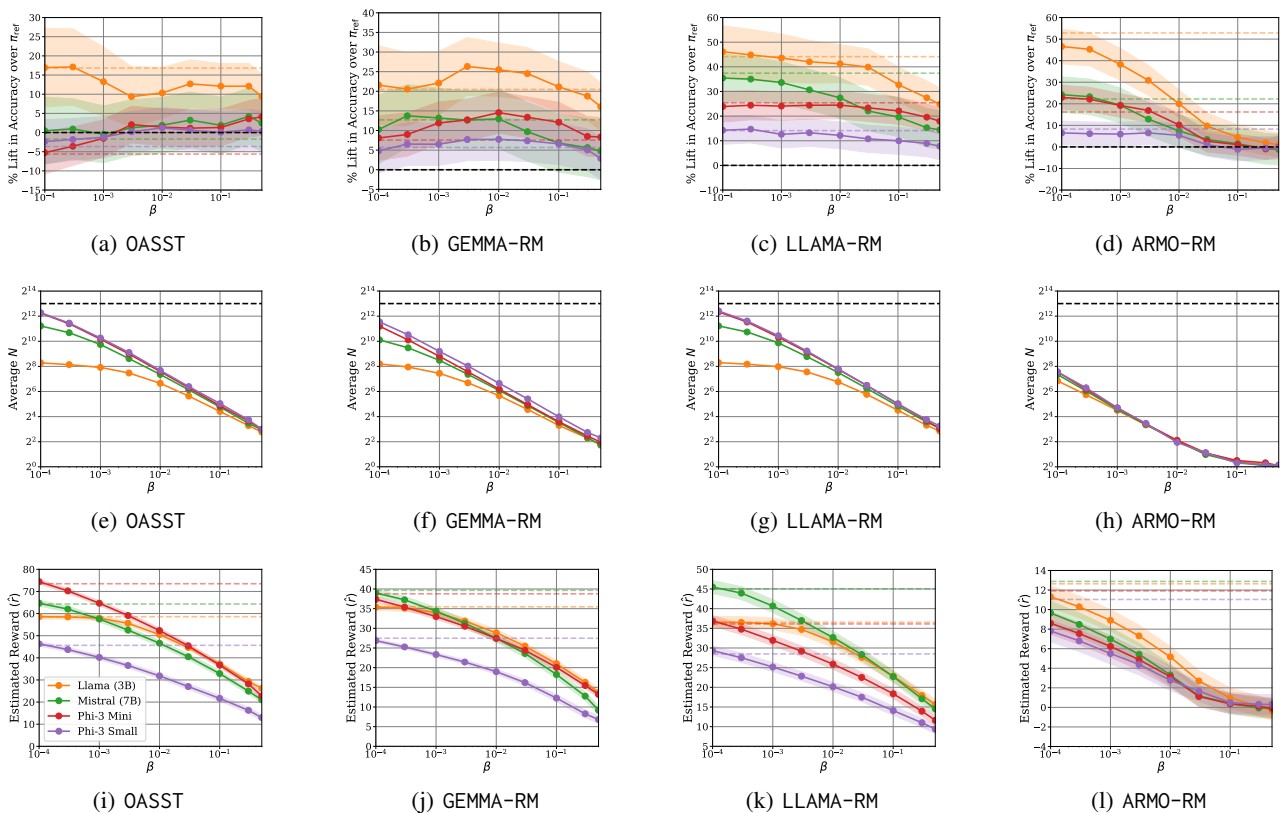

*Figure 7.* Compute-normalized comparison for $N = 2^{13}$ between `BoN-Alignment` and `InferenceTimePessimism` on `MMLU` for four reward models and choices of $\pi_{\mathsf{ref}}$, as a function of regularization $\beta$.

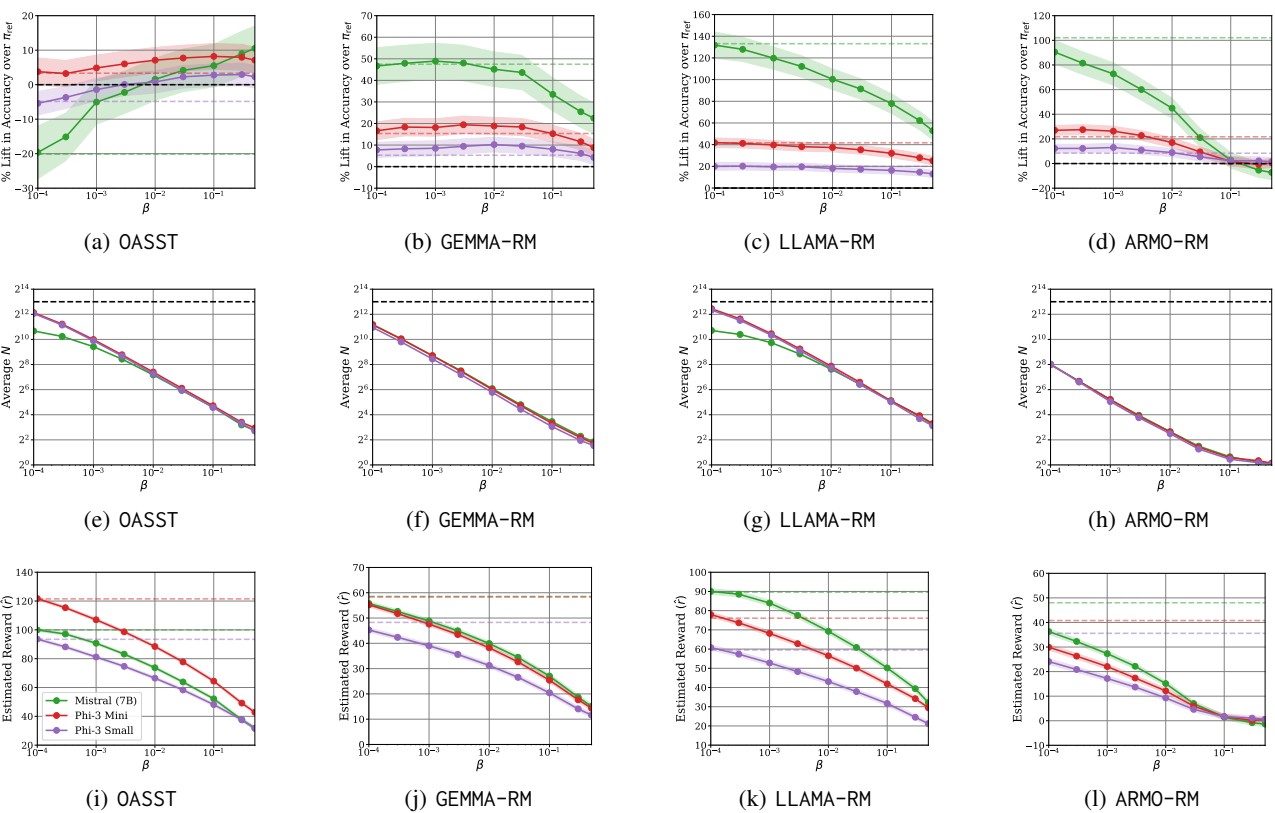

*Figure 8.* Compute-normalized comparison for $N = 2^{13}$ between `BoN-Alignment` and `InferenceTimePessimism` on `MATH` for four reward models and choices of $\pi_{\mathsf{ref}}$, as a function of regularization $\beta$.

### B.3. Further Experiments

We performed several additional experiments to (i) validate the basic modeling assumptions in our inference-time alignment framework—particularly the assumed reward model accuracy bound in Eq. (3);, and (ii) probe the behavior and robustness of InferenceTimePessimism, and. We begin by examining the distribution of the reward model scores $\widehat{r}(x, y)$ under $\pi_{\text{ref}}$, then explore the robustness of InferenceTimePessimism to the choice of regularization parameter $\beta$. we also present preliminary results on the AlpacaEval-2.0 task in Appendix B.4.

**Reward distribution under $\pi_{\text{ref}}$.** In order to get a more fine-grained sense for the extent to which reward overoptimization is a problem, in Figure 9 we plot the distribution of the reward model value $\widehat{r}(x, y)$ for a single representative prompt from GSM8K for all GEMMA-2-2B-generated responses, according to each of our four reward models, and conditioned on whether or not the response is correct. The more separated the distributions are, and the further to the right the correct (blue) distribution is, the better the reward model is at estimating the true reward. As we see, ARMO-RM is by far the best reward model in this respect. In particular, one reason to expect that would not observe reward overoptimization in BoN-Alignment for ARMO-RM with this task is the fact that the maximal value in the support of the incorrect distribution is, empirically, strictly smaller than that of the correct distribution; thus for sufficiently large $N$, BoN-Alignment will always choose the correct answer, at least for the prompt we visualize. This observation is consistent with our theoretical results, but suggests that in some cases our assumptions may be too pessimistic; indeed, InferenceTimePessimism may be overly conservative in situations where $\widehat{r}$ already underestimates $r^\star$ by a large margin (since pessimism, or under-estimating the true reward value, is precisely what the regularization in InferenceTimePessimism is designed to enforce).

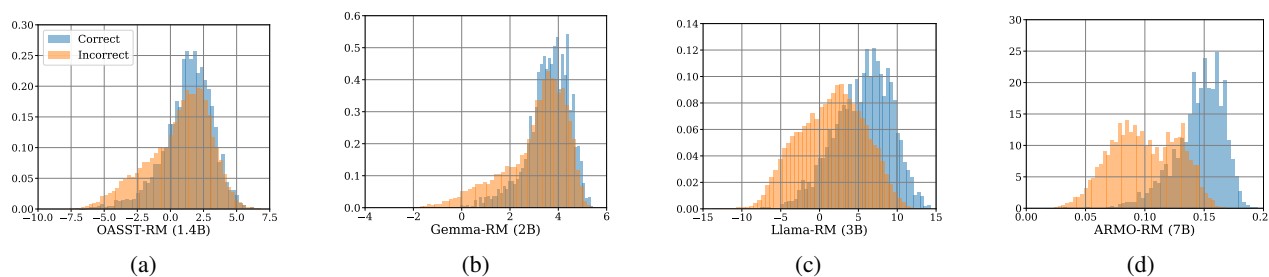

*Figure 9.* Distribution of estimated rewards of responses generated by GEMMA-2-2B on GSM8K prompt number 100 conditioned on whether or not the response is correct for (a) OASST, (b) GEMMA-RM, (c) LLAMA-RM, and (d) ARMO-RM. Greater separation of the distributions with the correct (blue) further to the right than incorrect (orange) indicates the reward model is more informative.

**Robustness of InferenceTimePessimism to $\beta$.** In addition to the theoretical suboptimality under general notions of coverage, the central drawback to BoN-Alignment is the lack of monotonicity, requiring careful tuning of the computational budget $N$ in order to avoid over-optimization. In Figure 10, we plot the accuracy of InferenceTimePessimism on Mistral-7B generations for a representative subset of the tasks and reward models $\widehat{r}$ we consider, evaluating the effect of the regularization parameter $\beta$ on performance. We find (Figures 10(a) and 10(c)) that InferenceTimePessimism experiences less over-optimization than BoN-Alignment fairly robustly across a range of $\beta$ values, though tuning $\beta$ is typically required to avoid over-optimization entirely. We also observe that in some cases, where the reward model remains in-distribution for all task responses, BoN-Alignment is monotonic without further interventions (Figures 10(b) and 10(d)).

Note that for small values of the regularization $\beta$, we do observe a small dip in performance for InferenceTimePessimism (cf. Figures 10(a) and 10(c)). This is caused by our heuristic of defaulting to BoN-Alignment when no responses are accepted by rejection sampling in InferenceTimePessimism, which is more likely to occur when the computational budget $N$ is small relative to the inverse of regularization $1/\beta$ according to our theory (Lemma D.4). We should like to remark that this is a reasonable heuristic, as it is precisely in the small $N$ regime that BoN-Alignment is expected to still perform well according to our theoretical results.

### B.4. Results for AlpacaEval-2.0

In addition to the experiments with GSM8K, MATH, and MMLU, we conduct a preliminary investigation on the performance of InferenceTimePessimism on AlpacaEval-2.0 (cf. Appendix B.1 for an explanation of this task). Because evaluation requires many queries to the proprietary OpenAI models, we consider a significantly smaller scale of experiments for this task, and use only 5 replicates for each prompt as opposed to 50. The results are displayed in Figure 11. Due to the

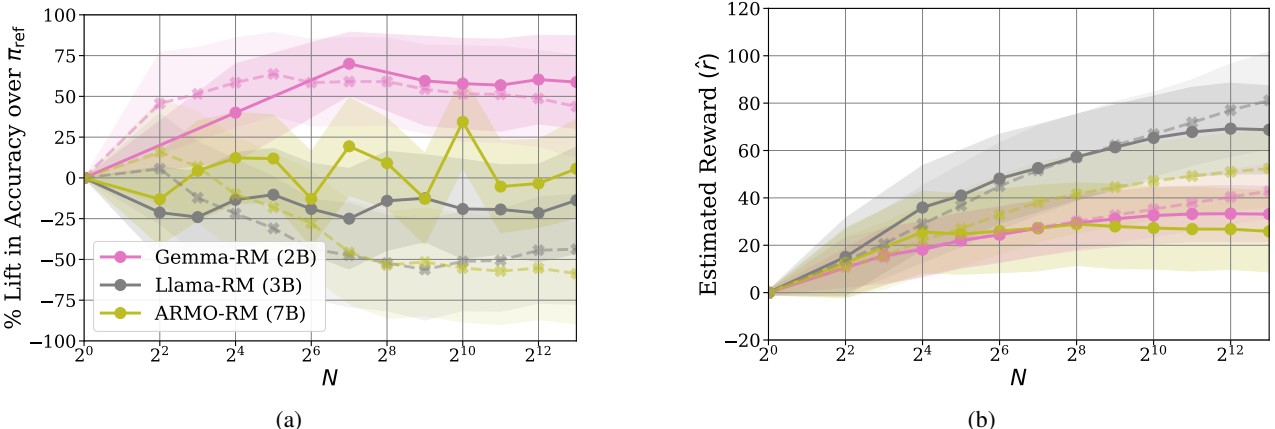

*Figure 10.* Demonstration that monotonicity is robust to the choice of regularization parameter $\beta$ for `Mistral-7B` generations with a representative sample of tasks and estimated rewards $\widehat{r}$.

noise of the evaluation, coupled with the significantly smaller number of replicates and prompts, it is difficult to separate the performance of `BoN-Alignment` and `InferenceTimePessimism` in a statistically significant way, although the broader trends agree with those found in our other tasks.

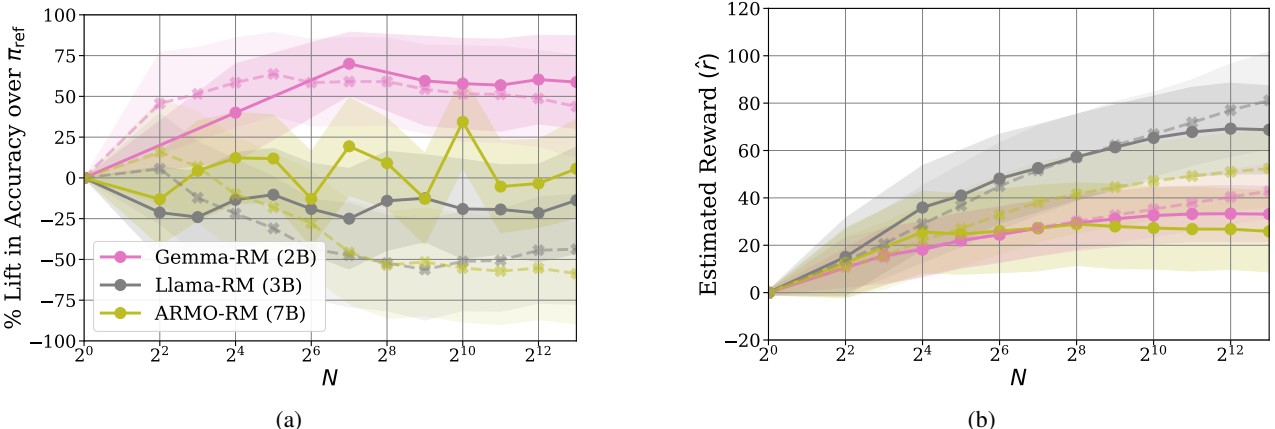

*Figure 11.* Performance of `BoN-Alignment` and `InferenceTimePessimism` on `AlpacaEval-2.0` with `GEMMA-2-2B` as $\pi_{\text{ref}}$ for several reward models.

---

**Algorithm 3** `ComputeNormConstant`

---

    **input:** Prompt $x$, reward model $\widehat{r}$, regularization coefficient $\beta > 0$, set of responses $\widehat{\mathcal{Y}}_N = (y_1, \ldots, y_N)$.

1: Sort and bucket $\widehat{\mathcal{Y}}_N$ into bins $\mathcal{Y}_1, \ldots, \mathcal{Y}_M$ according to the value of $\widehat{r}(x, y)$, in ascending order, such that

$$\widehat{r}(x, y) = \widehat{r}_i, \quad \forall y \in \mathcal{Y}_i, \forall i \in [M], \quad \text{and} \quad \widehat{r}_i < \widehat{r}_{i+1}, \qquad \forall i \in [M]$$

2: Initialize $\widehat{r}_0 = -\infty$, $J \leftarrow \sum_{i=1}^{M} \widehat{r}_i \cdot \frac{|\mathcal{Y}_i|}{N}$ and $Z \leftarrow 1$.

3: **for** $i = 1 \ldots M$ **do**

4:      Set $\lambda \leftarrow \frac{J - \beta}{Z}$.

5:      **if** $\widehat{r}_{i-1} \leq \lambda < \widehat{r}_i$ or $i = M$ **then**

6:          **return** $\lambda$.

7:      **else**

8:          Update $J \leftarrow J - \widehat{r}_i \cdot \frac{|\mathcal{Y}_i|}{N}$ and $Z \leftarrow Z - \frac{|\mathcal{Y}_i|}{N}$.

---

**Algorithm 4** `ComputeNormConstant` for general base measures

---

    **input:** Prompt $x$, reward model $\widehat{r}$, reference policy $\pi_{\mathsf{ref}}$, regularization coefficient $\beta > 0$, set of responses $\widehat{\mathcal{Y}}_N = (y_1, \ldots, y_N)$.

1: Sort and bucket $\widehat{\mathcal{Y}}_N$ into bins $\mathcal{Y}_1, \ldots, \mathcal{Y}_M$ according to the value of $\widehat{r}(x, y)$ in ascending order, such that

$$\widehat{r}(x, y) = \widehat{r}_i, \quad \forall y \in \mathcal{Y}_i, \forall i \in [M]$$
$$\widehat{r}_i < \widehat{r}_{i+1}, \qquad \forall i \in [M],$$

    and denote $\pi_{\mathsf{ref}}(\mathcal{Y}_i \mid x) = \sum_{y \in \mathcal{Y}_i} \pi_{\mathsf{ref}}(y \mid x)$

2: Initialize $\widehat{r}_0 = -\infty$, $J \leftarrow \sum_{i=1}^{M} \pi_{\mathsf{ref}}(\mathcal{Y}_i \mid x) \cdot \widehat{r}_i$ and $Z \leftarrow \sum_{i=1}^{M} \pi_{\mathsf{ref}}(\mathcal{Y}_i \mid x)$.

3: **for** $i = 1 \ldots M$ **do**

4:      Set $\lambda \leftarrow \frac{J - \beta}{Z}$.

5:      **if** $\widehat{r}_{i-1} \leq \lambda < \widehat{r}_i$ or $i = M$ **then**

6:          **return** $\lambda$.

7:      **else**

8:          Update $J \leftarrow J - \widehat{r}_i \cdot \pi_{\mathsf{ref}}(\mathcal{Y}_i \mid x)$ and $Z \leftarrow Z - \pi_{\mathsf{ref}}(\mathcal{Y}_i \mid x)$.

---

# Part II

# Proofs

## C. Normalization Constant Computation

### C.1. Background

This section gives guarantees for the `ComputeNormConstant` subroutine (Algorithm 3) used within `InferenceTimePessimism`. Given a set of response $\widehat{\mathcal{Y}}_N$ and $x \in \mathcal{X}$, the algorithm computes a normalization constant $\lambda$ such that

$$\widehat{\Phi}(\lambda) := \frac{1}{N} \sum_{y \in \widehat{\mathcal{Y}}_N} \mathsf{relu}\big(\beta^{-1}(\widehat{r}(x, y) - \lambda)\big) = 1$$

in time $O(N \log N)$ using a dynamic programming-like procedure. Note that such a $\lambda$ always exists because $\widehat{\Phi}(\lambda)$ is a continuous, piecewise linear function that is decreasing in $\lambda$, with $\widehat{\Phi}(-\infty) = \infty$ and $\widehat{\Phi}(\infty) = 0$. In what follows, we state and prove the main guarantee for the algorithm. En route, we will also prove a guarantee for a more general version of `ComputeNormConstant` (Algorithm 4), which computes a normalization constant such that $\Phi(\lambda) := \sum_{y \in \mathcal{Y}} \pi_{\mathsf{ref}}(y \mid x) \mathsf{relu}\big(\beta^{-1}(\widehat{r}(x, y) - \lambda)\big) = 1$ in time $O(N \log N)$.

### C.2. Guarantee for `ComputeNormConstant`

**Lemma C.1** (Main guarantee for `ComputeNormConstant`). *For any $\widehat{r}$, $\beta$, $\widehat{\mathcal{Y}}_N$, and $x \in \mathcal{X}$, Algorithm 3 finds $\lambda$ such that $\widehat{\Phi}(\lambda) := \frac{1}{N} \sum_{y \in \widehat{\mathcal{Y}}_N} \mathsf{relu}\big(\beta^{-1}(\widehat{r}(x, y) - \lambda)\big) = 1$ in $O(N \log N)$ time.*

**Proof of Lemma C.1.** Algorithm 3 is equivalent to running Algorithm 4 with $\widehat{r}$, $\beta$, $\mathcal{Y} = \widehat{\mathcal{Y}}_N$ and $\pi'_{\mathsf{ref}}(y \mid x) = \frac{1}{N} \cdot \mathbb{I}\Big[y \in \widehat{\mathcal{Y}}_N\Big]$ and so we can directly apply Lemma C.2. $\qquad\square$

**Lemma C.2** (Guarantee for generalized `ComputeNormConstant`). *For any $\widehat{r}$, $\beta$, $\widehat{\mathcal{Y}}_N$, and $x \in \mathcal{X}$, Algorithm 4 finds $\lambda$ such that $\widehat{\Phi}(\lambda) := \sum_{y \in \widehat{\mathcal{Y}}_N} \pi_{\mathsf{ref}}(y \mid x)\mathsf{relu}\big(\beta^{-1}(\widehat{r}(x, y) - \lambda)\big) = 1$ in $O(N \log N)$ time.*

**Proof of Lemma C.2.** Let $x \in \mathcal{X}$ be fixed; we omit dependence on $x$ going forward to keep notation compact. We begin by defining a surrogate problem with response space $\widehat{\mathcal{Y}}_M := \{y_1, \ldots, y_M\}$, where $y_i$ is any response from $\mathcal{Y}_i$, surrogate reward function $\widehat{r}'$ with $\widehat{r}'(y_i) =: \widehat{r}'_i = \widehat{r}_i$, and surrogate reference policy $\pi'_{\mathsf{ref}}$, where for $i \in [M]$ we set

$$\pi'_{\mathsf{ref}}(y_i) = \sum_{y \in \mathcal{Y}_i} \pi_{\mathsf{ref}}(y).$$

Note that all responses in this collection have unique values under a surrogate reward function $\widehat{r}'$. Since Algorithm 4 sorts rewards in ascending order, we also have that the surrogate rewards are indexed in ascending order, i.e., $\widehat{r}'_i < \widehat{r}'_{i+1}$ for all $i$. We also define the following function, which is the analog of $\widehat{\Phi}(\lambda)$ defined over the $M$ surrogate responses,

$$\widehat{\Phi}'(\lambda) = \sum_{i=1}^{M} \pi'_{\mathsf{ref}}(y_i)\mathsf{relu}\big(\beta^{-1}(\widehat{r}'_i - \lambda)\big).$$

For any $\lambda$, it can be seen that $\widehat{\Phi}'(\lambda) = \widehat{\Phi}(\lambda)$ since

$$\widehat{\Phi}'(\lambda) = \sum_{i=1}^{M} \left(\sum_{y \in \mathcal{Y}_i} \pi_{\mathsf{ref}}(y)\right)\mathsf{relu}\big(\beta^{-1}(\widehat{r}_i - \lambda)\big) = \sum_{y \in \widehat{\mathcal{Y}}_N} \pi_{\mathsf{ref}}(y)\mathsf{relu}\big(\beta^{-1}(\widehat{r}(y) - \lambda)\big) = \widehat{\Phi}(\lambda),$$

because each $y \in \widehat{\mathcal{Y}}_N$ is binned into one of $\{\mathcal{Y}_i\}_{i=1}^{M}$, and within each $\mathcal{Y}_i$ all responses share the reward label $\widehat{r}'_i$.

Next, define $\lambda^\star$ to be such that $\widehat{\Phi}'(\lambda^\star) = 1$, that is, the true constant that normalizes the distribution over the $M$ surrogate responses. Our goal is to show that Algorithm 4 computes $\lambda^\star$, which, given the previously shown equivalence, then proves the lemma statement since

$$1 = \widehat{\Phi}'(\lambda^\star) = \widehat{\Phi}(\lambda^\star).$$

Note that such a $\lambda^\star$ is guaranteed to exist because $\widehat{\Phi}(\lambda)$ is continuous and decreasing in $\lambda$. Moreover, from the form of $\widehat{\Phi}'$, it can be seen that there exists some index $j$ such that $\lambda^\star \geq \widehat{r}'_j$ and

$$1 = \widehat{\Phi}'(\lambda^\star) = \beta^{-1} \sum_{i>j} \pi'_{\mathsf{ref}}(y_i)(\widehat{r}'_i - \lambda^\star).$$

In other words, there exists an index $j \in [M]$ such that $\lambda^\star \in [\widehat{r}'_j, \widehat{r}'_{j+1}]$, which also shows that $\lambda^\star \in [J(\pi_{\mathsf{ref}}) - \beta, \widehat{r}'_M]$. Then to find $\lambda^\star$, that is, a $\lambda$ such that $\widehat{\Phi}'(\lambda) = 1$, it is sufficient to find an index $j$ such that for

$$\lambda = \frac{\sum_{i>j} \pi'_{\mathsf{ref}}(y_i)\widehat{r}'_i - \beta}{\sum_{i>j} \pi'_{\mathsf{ref}}(y_i)},$$

we have $\widehat{r}'_j \leq \lambda < \widehat{r}'_{j+1}$. This is exactly the output of Algorithm 4 run on $\widehat{\mathcal{Y}}_M, \pi'_{\mathsf{ref}}, \widehat{r}', \beta$, which breaks at an index that satisfies the above conditions, and outputs the corresponding $\lambda$. By inspection, such a $\lambda$ satisfies $\widehat{\Phi}'(\lambda) = 1$, and therefore also satisfies $\widehat{\Phi}(\lambda) = 1$.

Lastly, we discuss the computational complexity of Algorithm 4. Sorting $\widehat{\mathcal{Y}}_N$ and $\widehat{r}$ require $O(N \log N)$ time, while binning is $O(N)$. Finally, we make one forward pass through the responses, which requires at most $N$ iterations, each with $O(1)$ computations, before termination. $\qquad\square$

# D. Rejection Sampling

This section includes background on and guarantees for approximate rejection sampling, which forms the basis for our analysis for BoN and InferenceTimePessimism.

The organization is as follows. First, in Appendix D.1, we describe the approximate rejection sampling algorithm (Algorithm 5)—which is used within InferenceTimePessimism, as well as within the analysis of BoN-Alignment—and introduce the fundamental concepts that we will use to analyze the sample complexity of RejectionSampling. Then, in Appendix D.2 we provide upper bounds on the sample complexity of RejectionSampling, with matching lower bounds in Appendix D.3, in terms of the aforementioned measure, demonstrating its fundamental nature and the tightness of our results.

---

**Algorithm 5** Rejection Sampling ($\texttt{RejectionSampling}_{N,M}(w\,;\pi_{\mathsf{ref}},x)$)

---

    **input:** Prompt $x$, base policy $\pi_{\mathsf{ref}}$, importance weight $w$, truncation level $M$.
1: Draw $\widehat{\mathcal{Y}}_N = (y_1, \ldots, y_N, y_{N+1}) \sim \pi_{\mathsf{ref}}(\cdot \mid x)$ i.i.d.
2: **for** $i = 1 \ldots N$ **do**
3:     Sample Bernoulli random variable $\xi_i$ such that $\mathbb{P}(\xi_i = 1 \mid y_i) = \min\left\{\frac{w(y_i|x)}{M}, 1\right\}$
4:     **if** $\xi_i = 1$ **then**
5:         **return** response $y = y_i$
6: **return** response $y = y_{N+1}$.

---

## D.1. Background

The rejection sampling algorithm RejectionSampling is shown in Algorithm 5. The input parameters are the sample size $N$ and the rejection threshold $M$, a prompt $x$, and an importance sampling weight $w$.

The algorithm first draws $N$ samples from the conditional distribution of $\pi_{\mathsf{ref}}$ for the fixed prompt $x$, i.e., it draws $\widehat{\mathcal{Y}}_N = \{y_1, \ldots, y_N\}$ where $y_i \sim \pi_{\mathsf{ref}}(\cdot \mid x)$ for each $i \in [N]$. Then, for each $y_i \in \widehat{\mathcal{Y}}_N$, it samples a Bernoulli random variable $\xi_i$ where the probability of observing $\xi_i = 1$ is given by the importance weight $w(y_i \mid x)$ divided by the rejection threshold $M$, truncated to be at most 1. The algorithm returns any $y_i$ for which $\xi_i = 1$, and if no such event is observed, returns the first response (which is equivalent to randomly sampling from the base policy).

If $\pi(\cdot \mid x) = w(\cdot \mid x) \cdot \pi_{\mathsf{ref}}(\cdot \mid x)$ is a valid distribution, and the rejection threshold $M$ upper bounds the importance weight (or now, likelihood ratio) uniformly, that is, $M \geq \frac{\pi(y|x)}{\pi_{\mathsf{ref}}(y|x)}$ for all $x$ and $y$, then Algorithm 5 is identical to classical rejection sampling, where it is known that the law of accepted samples matches the target distribution $\pi$, i.e. $\mathbb{P}(y \mid \xi = 1, x) = \pi(y \mid x)$. If $M$ is not a uniform upper bound on the likelihood, the law of the accepted samples is not identical to $\pi$. This is because Algorithm 5 effectively truncates the distribution of $\pi$ to be at most $M \cdot \pi_{\mathsf{ref}}$, and any mass above this threshold is effectively lost. Nonetheless, the law of accepted responses is a close approximation to $\pi$ when $M$ is sufficiently large. *Approximate* rejection sampling under this regime was first analyzed in Block & Polyanskiy (2023), and our results in this section below borrow from their analysis.

For the remainder of this section, we will consider the problem of sampling from a target distribution or policy $\pi : \mathcal{X} \to \Delta(\mathcal{Y})$, by calling $\texttt{RejectionSampling}_{N,M}(\frac{\pi}{\pi_{\mathsf{ref}}}\,;\pi_{\mathsf{ref}}, x)$, where $\frac{\pi}{\pi_{\mathsf{ref}}}$ is its importance weight (or here, likelihood ratio) over the base policy. Concretely, we are concerned with analyzing how close the RejectionSampling response distribution is to $\pi$ as a function of the truncation level $M$. For example, if there exists $y$ such that $w(y \mid x) > 0$ but $\pi_{\mathsf{ref}}(y \mid x) = 0$ then it is not possible information-theoretically to sample from this portion of the target distribution, and similar reasoning applies to $y$ with poor coverage under $\pi_{\mathsf{ref}}$.

**Preliminaries: $\mathcal{E}_M$-divergence.** Based on the intuition above, a central object in our analysis will be the $\mathcal{E}_M$-divergence in Definition D.1 (Polyanskiy, 2010; Block & Polyanskiy, 2023), which, for an input rejection threshold $M$, quantifies the mass of the target distribution $\pi$ that is lost by truncating the importance weight $\frac{\pi}{\pi_{\mathsf{ref}}}$ to $M$.

**Definition D.1** ($\mathcal{E}_M$-divergence)**.** *For a prompt $x$, base policy $\pi_{\mathsf{ref}} : \mathcal{X} \to \Delta(\mathcal{Y})$, and target policy $\pi : \mathcal{X} \to \Delta(\mathcal{Y})$, let $\pi(x) := \pi(\cdot \mid x)$ refer to the distribution conditioned on $x$, and define $\pi_{\mathsf{ref}}(x)$ similarly. Then for any rejection threshold*

$M \geq 1$, the $\mathcal{E}_M$-divergence is defined as

$$\mathcal{E}_M(\pi(x), \pi_{\mathsf{ref}}(x)) := \mathbb{E}_{y \sim \pi_{\mathsf{ref}}(x)} \left[ \left( \frac{\pi(y \mid x)}{\pi_{\mathsf{ref}}(y \mid x)} - M \right)_+ \right] = \sum_{y \in \mathcal{Y}_M(x)} \pi(y \mid x) - M \cdot \pi_{\mathsf{ref}}(y \mid x),$$

where $\mathcal{Y}_M(x) = \{y \in \mathcal{Y} : \pi(y \mid x) > M \cdot \pi_{\mathsf{ref}}(y \mid x)\}$. In addition, the expected $\mathcal{E}_M$-divergence over the prompt distribution $\rho$ is defined as

$$\mathcal{E}_M(\pi, \pi_{\mathsf{ref}}) := \mathbb{E}_{x \sim \rho}[\mathcal{E}_M(\pi(x), \pi_{\mathsf{ref}}(x))].$$

Note that $\mathcal{E}_M(\pi(x), \pi_{\mathsf{ref}}(x))$ and $\mathcal{E}_M(\pi, \pi_{\mathsf{ref}})$ are non-increasing in $M$ in the sense that they do not increase as $M$ gets larger, and we will show in the sequel that they control the approximation error for RejectionSampling. In particular, the parameter $M$ strikes a bias-variance tradeoff in the RejectionSampling procedure. When $M$ is large, the algorithm requires a large number of samples in order to terminate; but for $M$ small, incurs larger bias from failing to sample from regions of the target density.

For the results that follow, it will be useful to define the smallest parameter $M$ that guarantees $\mathcal{E}_M(\pi(x), \pi_{\mathsf{ref}}(x)) \leq \varepsilon$ for some error tolerance of interest $\varepsilon \in [0, 1]$.

**Definition D.2.** *Given the base policy $\pi_{\mathsf{ref}}$, for a target policy $\pi$, prompt $x$, and any $\varepsilon \in [0, 1]$, define the smallest rejection threshold $M$ that ensures $\mathcal{E}_M(\pi(x), \pi_{\mathsf{ref}}(x)) \leq \varepsilon$ to be*

$$\mathcal{M}_{x,\varepsilon}^\pi := \min\{M \mid \mathcal{E}_M(\pi(x), \pi_{\mathsf{ref}}(x)) \leq \varepsilon\}.$$

*Similarly, define the smallest rejection threshold $M$ that ensures $\mathcal{E}_M(\pi, \pi_{\mathsf{ref}}) \leq \varepsilon$ to be*

$$\mathcal{M}_\varepsilon^\pi := \min\{M \mid \mathcal{E}_M(\pi, \pi_{\mathsf{ref}}) \leq \varepsilon\}.$$

Though the $\mathcal{E}_M$-divergence is perhaps the most natural object by which to quantiy the error of aproximate rejection sampling, it can also be upper bounded by other information-theoretic divergences, such as $\mathcal{C}_\infty^\pi$ and $\mathcal{C}^\pi$, which will be useful in the later analysis for BoN-Alignment and InferenceTimePessimism. We state the result below for $\mathcal{M}_{x,\varepsilon}^\pi$ for a fixed $x$, which can be stated for $\mathcal{M}_\varepsilon^\pi$ in a similar manner.

**Proposition D.3.** *Given the base policy $\pi_{\mathsf{ref}}$, for any target policy $\pi$ and prompt $x$, recall that $\mathcal{C}_\infty^\pi(x) = \sup_{y \in \mathcal{Y}} \frac{\pi(y|x)}{\pi_{\mathsf{ref}}(y|x)}$, and define $\mathcal{C}_\alpha^\pi(x) := \frac{1}{\alpha} \mathbb{E}_{y \sim \pi} \left[ \left( \frac{\pi(y|x)}{\pi_{\mathsf{ref}}(y|x)} \right)^{\alpha-1} \right]$ for any $\alpha > 1$, so that $\mathcal{C}_2^\pi(x) = \mathcal{C}^\pi(x)$. Then for any $\varepsilon \in [0, 1]$ and $\alpha \in (1, \infty)$, we have*

$$\mathcal{M}_{x,\varepsilon}^\pi \leq \min\left( \mathcal{C}_\infty^\pi(x), \ \left( \frac{\mathcal{C}_\alpha^\pi(x)}{\varepsilon} \right)^{\frac{1}{\alpha-1}} \right).$$

In particular, for the coverage coefficient $\mathcal{C}^\pi(x) = \mathcal{C}_2^\pi(x)$, the above result shows that $\mathcal{M}_{x,\varepsilon}^\pi \leq \frac{\mathcal{C}^\pi(x)}{\varepsilon}$. The proof below utilizes Block & Polyanskiy (2023, Example 7) and the fact that $\mathcal{C}_\alpha^\pi$ controls the Renyi divergence of order $\alpha$ between $\pi$ and $\pi_{\mathsf{ref}}$.

**Proof of Proposition D.3.** As the prompt is fixed, we omit $x$ dependencies for notational compactness. Recall that for any $M$, $\mathcal{E}_M(\pi, \pi_{\mathsf{ref}}) = \pi(\mathcal{Y}_M) - M \cdot \pi_{\mathsf{ref}}(\mathcal{Y}_M)$. The statement for $M = \mathcal{C}_\infty^\pi$ follows directly from the definition of $\mathcal{E}_M(\pi, \pi_{\mathsf{ref}})$ since $\mathcal{E}_{\mathcal{C}_\infty^\pi}(\pi, \pi_{\mathsf{ref}}) = 0$. For $\mathcal{C}_\alpha^\pi$, it can be seen that $\pi(\mathcal{Y}_M) \leq \frac{\mathcal{C}_\alpha^\pi}{M^{\alpha-1}}$ since

$$\mathcal{C}_\alpha^\pi \geq \sum_y \frac{\pi(y)^\alpha}{\pi_{\mathsf{ref}}(y)^{\alpha-1}} \cdot \mathbb{I}[y \in \mathcal{Y}_M] > M^{\alpha-1} \sum_y \pi(y) \cdot \mathbb{I}[y \in \mathcal{Y}_M] = M^{\alpha-1} \cdot \pi(\mathcal{Y}_M).$$

Then for $M$ to satisfy

$$\varepsilon \leq \mathcal{E}_M(\pi, \pi_{\mathsf{ref}}) \leq \frac{\mathcal{C}_\alpha^\pi}{M^{\alpha-1}} - M \cdot \pi_{\mathsf{ref}}(\mathcal{Y}_M) \leq \frac{\mathcal{C}_\alpha^\pi}{M^{\alpha-1}},$$

it suffices to have $M = \left( \frac{\mathcal{C}_\alpha^\pi}{\varepsilon} \right)^{\frac{1}{\alpha-1}}$. □

### D.2. Guarantee for `RejectionSampling`

This section contains sample complexity upper bounds for Algorithm 5, which express the size of $N$ required to ensure that the total variation distance between the law of responses drawn from Algorithm 5 and the target distribution $\pi$ is small. The main result, Lemma D.4, that expresses sample complexity in terms of $M$ and $\mathcal{E}_M(\pi(x), \pi_{\text{ref}}(x))$.

For a fixed $N$, Lemma D.4 shows that the total variation distance is bounded by the $\mathcal{E}_M$-divergence corresponding to the input rejection threshold $M$, as well as an exponential in $\frac{1}{N}$ term that bounds the error when the algorithm fails to terminate. As expected, the former term decreases as $M$ increases, while the lattter increases. In addition, while increasing $N$ can reduce error from the latter term, the former term is irreducible even as $N$ tends too infinity, and this too we expect given that $\mathcal{E}_M(\pi, \pi_{\text{ref}})$ is an information-theoretic measure of error that is a property of the distributions and the choice of $M$.

**Lemma D.4** (Per-prompt rejection sampling upper bound; adapted from Theorem 3 in Block & Polyanskiy (2023))**.** *For any valid policy $\pi : \mathcal{X} \to \Delta(\mathcal{Y})$, $N \in \mathbb{Z}$, and $M \in \mathbb{R}_+$, for a given prompt $x \in \mathcal{X}$, let $\pi_{\text{R}}(x) \in \Delta(\mathcal{Y})$ be the law of responses induced by running* `RejectionSampling`$_{N,M}(\frac{\pi}{\pi_{\text{ref}}}; \pi_{\text{ref}}, x)$ *(Algorithm 5). We have*

$$D_{\text{TV}}(\pi(x), \pi_{\text{R}}(x)) \leq \mathcal{E}_M(\pi(x), \pi_{\text{ref}}(x)) + \frac{1}{2}\exp\left(-\frac{N \cdot (1 - \mathcal{E}_M(\pi(x), \pi_{\text{ref}}(x)))}{M}\right).$$

*In particular, if $M = \mathcal{M}^{\pi}_{x,\varepsilon}$ and $N \gtrsim \mathcal{M}^{\pi}_{x,\varepsilon} \cdot \log\left(\frac{1}{\varepsilon}\right)$ for some $\varepsilon \in \left(0, \frac{1}{2}\right)$, we have that $D_{\text{TV}}(\pi(x), \pi_{\text{ref}}(x)) \lesssim \varepsilon$.*

**Proof of Lemma D.4.** As the prompt is fixed, we omit $x$ dependencies for notational compactness. Define the truncated pseudo-distribution $\widetilde{\pi} = \min\{\pi, M \cdot \pi_{\text{ref}}\}$. Define $A_M := \sum_y \widetilde{\pi}(y)$ to be the total mass of the truncated policy. Recalling that $\mathcal{Y}_M(x) = \{y \in \mathcal{Y} : \pi(y \mid x) > M \cdot \pi_{\text{ref}}(y \mid x)\}$, $A_M$ can be equivalent expressed as

$$A_M = \sum_y \min\{\pi(y), M \cdot \pi_{\text{ref}}(y)\}$$
$$= \sum_{y \in \mathcal{Y}_M} M \cdot \pi_{\text{ref}}(y) + \sum_{y \notin \mathcal{Y}_M} \pi(y)$$
$$= 1 - \mathcal{E}_M(\pi, \pi_{\text{ref}}).$$

Recall that, for a single sample $y \sim \pi_{\text{ref}}$, Algorithm 5 samples $\xi \sim \text{Ber}(p_y)$, where $p_y := \min\left\{\frac{\pi(y)}{\pi_{\text{ref}}(y) \cdot M}, 1\right\}$ is a Bernoulli random variable such that $\mathbb{P}_{\xi \sim \text{Ber}(p_y)}(\xi = 1 \mid y) = p_y$. Then the expected probability of acceptance for a single sample is

$$\mathbb{P}_{y \sim \pi_{\text{ref}}, \xi \sim \text{Ber}(p_y)}(\xi = 1) = \mathbb{E}_{y \sim \pi_{\text{ref}}}[\mathbb{P}(\xi = 1 \mid y)]$$
$$= \sum_y \pi_{\text{ref}}(y) \cdot \min\left\{\frac{\pi(y)}{M \cdot \pi_{\text{ref}}(y)}, 1\right\}$$
$$= \frac{1}{M}\sum_y \min\{\pi(y), M \cdot \pi_{\text{ref}}(y)\}$$
$$= \frac{A_M}{M},$$

and it can be seen that the law of the response conditioned on acceptance, $\sum_{y \in \mathcal{Y}_M} M \cdot \pi_{\text{ref}}(y) + \sum_{y \notin \mathcal{Y}_M} \pi(y)$, is equivalent to the normalized version of $\widetilde{\pi}$ since

$$\mathbb{P}_{y' \sim \pi_{\text{ref}}}(y' = y \mid \xi = 1) = \frac{\mathbb{P}_{y' \sim \pi_{\text{ref}}, \xi \sim \text{Ber}(p_y)}(y' = y, \xi = 1)}{\mathbb{P}_{y' \sim \pi_{\text{ref}}, \xi \sim \text{Ber}(p_y)}(\xi = 1)} = \frac{\pi_{\text{ref}}(y) \cdot \min\left\{\frac{\pi(y)}{M \cdot \pi_{\text{ref}}(y)}, 1\right\}}{A_M/M} = \frac{\widetilde{\pi}(y)}{A_M}.$$

Next, given $\widehat{\mathcal{Y}}_N = \{y_1, \ldots, y_N\}$, let $\widehat{\xi}_N = \{\xi_1, \ldots, \xi_N\}$ be the random draw of Bernoulli random variables, where $\xi_i \sim \text{Ber}(p_{y_i})$ for all $i \in [N]$. For short, we write $\mathbb{P}_{\widehat{\mathcal{Y}}_N, \widehat{\xi}_N}(\cdot) \equiv \mathbb{P}_{\widehat{\mathcal{Y}}_N \sim \pi_{\text{ref}}, \xi_1 \sim \text{Ber}(y_1), \ldots, \xi_N \sim \text{Ber}(y_N)}(\cdot)$. Define also the following event, which is a random variable over the draw of $\widehat{\mathcal{Y}}_N$ and $\widehat{\xi}_N$, under which rejection sampling accepts one of the $N$ responses and Algorithm 5 outputs the $i^\star$th response in Line 5,

$$\text{stop} := \{\exists i^\star \in [N] \text{ s.t. } \xi_{i^\star} = 1\}. \tag{14}$$

Let $\widehat{y} = \texttt{RejectionSampling}_{N,M}(\frac{\pi}{\pi_{\text{ref}}} ; \pi_{\text{ref}}, x)$ be the response output by Algorithm 5, which is a random variable over draws of $\widehat{\mathcal{Y}}_N$ and $\widehat{\xi}_N$. Due to independence, we have that the law of responses is given by

$$\mathbb{P}_{\widehat{\mathcal{Y}}_N, \widehat{\xi}_N}(\widehat{y} = y \mid \text{stop}) = \mathbb{P}_{\widehat{\mathcal{Y}}_N, \widehat{\xi}_N}(y_{i^*} = y \mid \exists i^* \in [N] \text{ s.t. } \xi_{i^*} = 1) = \frac{\widetilde{\pi}(y)}{A_M}. \qquad (15)$$

It can be observed that via union bound,

$$\begin{aligned}
\mathbb{P}_{\widehat{\mathcal{Y}}_N, \widehat{\xi}_N}(\neg\text{stop}) &= \mathbb{P}_{\widehat{\mathcal{Y}}_N, \widehat{\xi}_N}(\xi_i = 0, \ \forall i \in [N]) \\
&= \left(1 - \mathbb{P}_{y \sim \pi_{\text{ref}}, \xi \sim \text{Ber}(p_y)}(\xi = 1)\right)^N \\
&= \left(1 - \frac{A_M}{M}\right)^N \\
&\le e^{-\frac{N \cdot A_M}{M}}. \qquad (16)
\end{aligned}$$

Recall that $\pi_{\text{R}}(y) = \mathbb{P}_{\widehat{\mathcal{Y}}_N, \widehat{\xi}_N}(y)$. Now for the upper bound, we first decompose the total variation distance using $\widetilde{\pi}$,

$$D_{\text{TV}}(\pi, \pi_{\text{R}}) = \frac{1}{2} \sum_y |\pi(y) - \pi_{\text{R}}(y)| \le \underbrace{\frac{1}{2} \sum_y |\pi(y) - \widetilde{\pi}(y)|}_{(\text{T1})} + \underbrace{\frac{1}{2} \sum_y |\pi_{\text{R}}(y) - \widetilde{\pi}(y)|}_{(\text{T2})},$$

where, via Definition D.1, the first term is equivalent to

$$(\text{T1}) = \sum_y \pi(y) - \min\{\pi(y), M \cdot \pi_{\text{ref}}(y)\} = \sum_y \pi(y) \cdot \mathbb{I}[\pi(y) > M \cdot \pi_{\text{ref}}(y)] = \mathcal{E}_M(\pi, \pi_{\text{ref}}).$$

and we further bound the second term as

$$\begin{aligned}
(\text{T2}) &= \sum_y \left| \mathbb{P}_{\widehat{\mathcal{Y}}_N, \widehat{\xi}_N}(\widehat{y} = y \mid \text{stop}) + \mathbb{P}_{\widehat{\mathcal{Y}}_N, \widehat{\xi}_N}(\widehat{y} = y \mid \neg\text{stop}) - \widetilde{\pi}(y) \right| \\
&\le \underbrace{\sum_y \left| \mathbb{P}_{\widehat{\mathcal{Y}}_N, \widehat{\xi}_N}(\widehat{y} = y \mid \text{stop}) - \widetilde{\pi}(y) \right|}_{(\text{T3})} + \mathbb{P}_{\widehat{\mathcal{Y}}_N, \widehat{\xi}_N}(\neg\text{stop})
\end{aligned}$$

From Eq. (16), have $\mathbb{P}_{\widehat{\mathcal{Y}}_N, \widehat{\xi}_N}(\neg\text{stop}) \le e^{-\frac{N \cdot A_M}{M}}$. In addition, using Eq. (15) and the fact that $A_M \le 1$, we have that

$$\begin{aligned}
(\text{T3}) &= \sum_y \left| \mathbb{P}_{\widehat{\mathcal{Y}}_N, \widehat{\xi}_N}(\widehat{y} = y \mid \text{stop}) - \widetilde{\pi}(y) \right| \\
&= \sum_y \left| \frac{\widetilde{\pi}(y)}{A_M} - \widetilde{\pi}(y) \right| = \sum_y \frac{\widetilde{\pi}(y)}{A_M} - \widetilde{\pi}(y) \\
&= 1 - A_M = \mathcal{E}_M(\pi, \pi_{\text{ref}}),
\end{aligned}$$

so that

$$(\text{T2}) \le \mathcal{E}_M(\pi, \pi_{\text{ref}}) + \exp^{-\frac{N \cdot A_M}{M}}.$$

Combining (T1) and (T2),

$$\begin{aligned}
D_{\text{TV}}(\pi, \pi_{\text{ref}}) &\le \frac{1}{2}((\text{T1}) + (\text{T2})) \\
&\le \mathcal{E}_M(\pi, \pi_{\text{ref}}) + \frac{1}{2} \exp^{-\frac{N \cdot A_M}{M}},
\end{aligned}$$

and substituting the expression for $A_M = 1 - \mathcal{E}_M(\pi, \pi_{\text{ref}})$ gives the result. $\qquad \square$

### D.3. Lower Bounds

For a fixed prompt $x$, Lemma D.6 shows $N \gtrsim \mathcal{M}_{x,\varepsilon}^\pi \log(\varepsilon^{-1})$ samples are sufficient to guarantee that $D_{\mathsf{TV}}(\pi(x), \pi_{\mathsf{ref}}(x)) \leq \varepsilon$. The information-theoretic lower bound in Lemma D.6 shows that this dependence is tight for any selection strategy that selects a response from $\widehat{\mathcal{Y}}_N$, defined more formally below.

**Definition D.5** (Selection strategy $\mathcal{A}$). *A selection strategy $\mathcal{A}$ is any method that, given a prompt $x$ and $N$ responses $\widehat{\mathcal{Y}}_N = (y_1, \ldots, y_N)$ sampled i.i.d. from $\pi_{\mathsf{ref}}$, returns some (possibly random) $y \in \widehat{\mathcal{Y}}_N$.*

Lemma D.6 states that selection strategy (Definition D.5) must obtain at least $\mathcal{M}_{x,\varepsilon}^\pi$ samples in order to induce a response distribution that is $\varepsilon$-close to $\pi$. We will later use this result as a component within our main regret lower bounds (Theorems 3.2 and 4.2).

**Lemma D.6** (TV lower bound, adapted from the proof of Theorem 5 in Block & Polyanskiy (2023)). *Fix the base policy $\pi_{\mathsf{ref}}$, target policy $\pi$, prompt $x$, and sample size $N$. Let $\mathcal{A}$ be any selection algorithm (Definition F.2) that, given $\widehat{\mathcal{Y}}_N \sim \pi_{\mathsf{ref}}(\cdot \mid x)$ in the sample-and-evaluate framework (Definition 2.2), outputs a response $y \in \widehat{\mathcal{Y}}_N$. Let $\pi_\mathcal{A} : \mathcal{X} \to \Delta(\mathcal{Y})$ denote the distribution of responses induced by $\mathcal{A}$. Then if $N < \mathcal{M}_{x,\varepsilon}^\pi$, we have*

$$D_{\mathsf{TV}}(\pi_\mathcal{A}(x), \pi(x)) > \varepsilon.$$

**Proof of Lemma D.6.** In the proof below we omit $x$ dependencies, including in $\mathcal{M}_{x,\varepsilon}^\pi \equiv \mathcal{M}_\varepsilon^\pi$, and $\mathcal{E}_M(\pi(x), \pi_{\mathsf{ref}}(x)) \equiv \mathcal{E}_M(\pi, \pi_{\mathsf{ref}})$. First fix $M$. Suppose $N < M$. Then

$$
\begin{aligned}
2 \cdot D_{\mathsf{TV}}(\pi_\mathcal{A}, \pi) &\geq \sum_{y \in \mathcal{Y}_M} |\pi(y) - \pi_\mathcal{A}(y)| \\
&\geq \sum_{y \in \mathcal{Y}_M} \pi(y) - \mathbb{P}\left(y \in \widehat{\mathcal{Y}}_N\right) \\
&\geq \sum_{y \in \mathcal{Y}_M} \pi(y) - N \cdot \mu(y) \\
&\geq \sum_{y \in \mathcal{Y}_M} \pi(y) - M \cdot \mu(y) \\
&= 2 \cdot \mathcal{E}_M(\pi, \pi_{\mathsf{ref}})
\end{aligned}
$$

It follows that if $N < M \leq \mathcal{M}_\varepsilon^\pi$, we have

$$D_{\mathsf{TV}}(\pi_\mathcal{A}, \pi) > \varepsilon$$

from the definition of $\mathcal{M}_\varepsilon^\pi$, since $\mathcal{E}_M(\pi, \pi_{\mathsf{ref}})$ is non-decreasing as $M$ decreases. $\qquad\square$

# E. Proofs from Section 2

**Proof of Proposition 2.3.** Let us write $J_r(\pi, x) := \mathbb{E}_{y \sim \pi(\cdot|x)}[r(x, y)]$ to denote the expected reward under a function $r(x, y)$. We first state an elementary technical lemma.

**Lemma E.1.** *For any prompt $x \in \mathcal{X}$, reward model $\widehat{r} : \mathcal{X} \times \mathcal{Y} \to \mathbb{R}$, and policies $\pi^\star, \widehat{\pi} : \mathcal{X} \to \Delta(\mathcal{Y})$ and $\varepsilon > 0$, there exists $r^\star : \mathcal{X} \times \mathcal{Y} \to \mathbb{R}$ such that*

$$J_{r^\star}(\pi^\star; x) - J_{r^\star}(\widehat{\pi}; x) \geq J_{\widehat{r}}(\pi^\star; x) - J_{\widehat{r}}(\widehat{\pi}; x) + \varepsilon \cdot \sqrt{\sum_{y \in \mathcal{Y}} \frac{(\pi^\star(y \mid x) - \widehat{\pi}(y \mid x))^2}{\pi_{\text{ref}}(y \mid x)}}$$

*and $\mathbb{E}_{y \sim \pi_{\text{ref}}(\cdot|x)}\left[(\widehat{r}(x, y) - r^\star(x, y))^2\right] \leq \varepsilon^2$.*

**Proof of Lemma E.1.** We use the choice

$$r^\star(x, y) = \widehat{r}(x, y) + \varepsilon \cdot \frac{\pi^\star(y \mid x) - \widehat{\pi}(y \mid x)}{\pi_{\text{ref}}(y \mid x)} \cdot \left( \sum_{y \in \mathcal{Y}} \frac{(\pi^\star(y \mid x) - \widehat{\pi}(y \mid x))^2}{\pi_{\text{ref}}(y \mid x)} \right)^{-1/2},$$

from which the result is immediate. $\qquad\square$

Going forward, we omit dependence on $x \in \mathcal{X}$ to keep notation compact. Define $\mathcal{C}^{\text{max}} = \max_{\pi : \mathcal{X} \to \Delta(\mathcal{Y})} \mathcal{C}^\pi$ and $\pi_{\text{max}} = \arg\max_{\pi : \mathcal{X} \to \Delta(\mathcal{Y})} \mathcal{C}^\pi$, and let $\mathcal{C}^{\text{max}} \geq C^\star \geq 8$ be given. Throughout the proof, we will use the fact that $\mathcal{C}^\pi \geq 1$ for all $\pi$. We set $\widehat{r}(x, y) = 0$ and consider two cases for the analysis.

**Case 1: $\mathcal{C}^{\widehat{\pi}} > \frac{1}{8}C^\star$.** In this case, we invoke Lemma E.1 with $\pi^\star = \pi_{\text{ref}}$ and $\varepsilon = \varepsilon_{\text{RM}}$, which shows that there exists some $r^\star$ for which

$$J_{r^\star}(\pi^\star) - J_{r^\star}(\widehat{\pi}) \geq \varepsilon_{\text{RM}} \cdot \sqrt{\sum_{y \in \mathcal{Y}} \frac{(\pi_{\text{ref}}(y) - \widehat{\pi}(y))^2}{\pi_{\text{ref}}(y)}}.$$

The result now follows by noting that

$$\sum_{y \in \mathcal{Y}} \frac{(\pi_{\text{ref}}(y) - \widehat{\pi}(y))^2}{\pi_{\text{ref}}(y)} = \mathcal{C}^{\widehat{\pi}} - 1 \geq \frac{1}{8}C^\star - 1 \geq \frac{1}{16}C^\star.$$

**Case 2: $\mathcal{C}^{\widehat{\pi}} \leq \frac{1}{8}C^\star$.** In this case, we set

$$\pi^\star = \lambda \pi_{\text{max}} + (1 - \lambda)\pi_{\text{ref}}$$

for $\lambda^2 := \frac{C^\star}{2\mathcal{C}^{\text{max}}} \in (0, 1)$. We compute directly that

$$\mathcal{C}^{\pi^\star} = \lambda^2 \mathcal{C}^{\text{max}} + (1 - \lambda^2),$$

so that $\mathcal{C}^{\pi^\star} \in \left[\frac{1}{2}C^\star, C^\star\right]$. We invoke Lemma E.1 with $\pi^\star$ and $\varepsilon = \varepsilon_{\text{RM}}$, which gives that there exists some $r^\star$ for which

$$J_{r^\star}(\pi^\star) - J_{r^\star}(\widehat{\pi}) \geq \varepsilon_{\text{RM}} \cdot \sqrt{\sum_{y \in \mathcal{Y}} \frac{(\pi^\star(y) - \widehat{\pi}(y))^2}{\pi_{\text{ref}}(y)}}.$$

We further compute that

$$\sum_{y \in \mathcal{Y}} \frac{(\pi^\star(y) - \widehat{\pi}(y))^2}{\pi_{\text{ref}}(y)} = \mathcal{C}^{\pi^\star} + \mathcal{C}^{\widehat{\pi}} - 2 \sum_{y \in \mathcal{Y}} \frac{\pi^\star(y)\widehat{\pi}(y)}{\pi_{\text{ref}}(y)} \geq \frac{1}{2}\mathcal{C}^{\pi^\star} - \mathcal{C}^{\widehat{\pi}},$$

where the last step follows by the AM-GM inequality, i.e.

$$\sum_{y \in \mathcal{Y}} \frac{\pi^\star(y)\widehat{\pi}(y)}{\pi_{\text{ref}}(y)} \leq \frac{1}{4} \sum_{y \in \mathcal{Y}} \frac{(\pi^\star(y))^2}{\pi_{\text{ref}}(y)} + \sum_{y \in \mathcal{Y}} \frac{(\widehat{\pi}(y))^2}{\pi_{\text{ref}}(y)}.$$

By the assumption for this case, we have $\mathcal{C}^{\widehat{\pi}} \leq \frac{1}{8}C^\star$, so that

$$\frac{1}{2}\mathcal{C}^{\pi^\star} - \mathcal{C}^{\widehat{\pi}} \geq \frac{1}{2}\mathcal{C}^{\pi^\star} - \frac{1}{8}C^\star \geq \frac{1}{4}C^\star - \frac{1}{8}C^\star = \frac{1}{8}C^\star,$$

completing the result.

$\square$

# F. Proofs from Section 3

This section gives proofs for the guarantees for BoN-Alignment in Section 3. These results in this section build on the guarantees for rejection sampling in Appendix D, and are organized as follows:

- Appendices F.1 and F.2 provide general tools to analyze BoN-Alignment. In Appendix F.1, we provide a general upper bound on the regret of BoN-Alignment in terms of the $\mathcal{E}_M$-*divergence* introduced in Appendix D, and in Appendix F.2 we give general lower bounds on the regret.

- In Appendices F.3 to F.5, we instantiate these results to prove the main theorems in Section 3. Namely, these results leverage Proposition D.3 to translate the $\mathcal{E}_M$-divergence to the $\mathcal{C}^\pi$ and $\mathcal{C}^\pi_\infty$ coverage coefficients, which are standard measures of distribution shift in offline alignment (cf. Proposition 2.3).

## F.1. General Regret Decomposition for BoN-Alignment

Recall that the $\mathcal{E}_M$-divergence (Definition D.1) is defined for a parameter $M > 0$ via

$$
\mathcal{E}_M(\pi(x), \pi_{\mathsf{ref}}(x)) := \sum_{y \in \mathcal{Y}_M(x)} \pi(y \mid x) - M \cdot \pi_{\mathsf{ref}}(y \mid x) = \mathbb{E}_{y \sim \pi_{\mathsf{ref}}(x)}\left[\left(\frac{\pi(y \mid x)}{\pi_{\mathsf{ref}}(y \mid x)} - M\right)_+\right].
$$

Our central technical result for the analysis of BoN-Alignment is Lemma F.1 below, which quantifies the regret of the BoN policy given $N$ samples. This result will later be instantiated to prove Theorem 3.1 and Theorem 3.4. Even though Lemma F.1 concerns BoN-Alignment, its analysis makes use of the rejection sampling algorithm (Algorithm 5) as a tool to analyze certain intermediate quantities. As a result, the lemma statement contains an extra parameter $M$, which corresponds to a rejection sampling threshold (cf. Algorithm 5), and the regret upper bound is expressed in terms of the information-theoretic $\mathcal{E}_M$-divergence (Definition D.1) which appears in the analysis of rejection sampling in Appendix D.

**Lemma F.1** ($\mathcal{E}_M$-divergence regret bound for BoN-Alignment). *Fix a prompt $x$. For any comparator policy $\pi^\star$ and $N \in \mathbb{Z}$ and $M \in \mathbb{R}_+$ such that $\mathcal{E}_M(\pi^\star(x), \pi_{\mathsf{ref}}(x)) \leq \frac{1}{2}$, the* BoN *policy $\widehat{\pi}_{\mathsf{BoN}}$ satisfies*

$$
J(\pi^\star; x) - J(\widehat{\pi}_{\mathsf{BoN}}; x) \leq R_{\max} \cdot \left(\mathcal{E}_M(\pi^\star(x), \pi_{\mathsf{ref}}(x)) + \exp\left(-\frac{N}{M} \cdot (1 - \mathcal{E}_M(\pi^\star(x), \pi_{\mathsf{ref}}(x)))\right)\right)
$$
$$
+ 2 \cdot \sqrt{\mathcal{C}^{\pi^\star}(x) \cdot \varepsilon_{\mathsf{RM}}^2(x)} + \frac{\varepsilon_{\mathsf{RM}}(x)}{2} + \sqrt{N \cdot \varepsilon_{\mathsf{RM}}^2(x)}
$$

Because Lemma F.1 holds for *any choice* of $M \in \mathbb{R}_+$, $M$ can be viewed as a parameter (within the analysis) that trades off between the best regret achievable —which is upper bounded by $\mathcal{E}_M(\pi^\star(x), \pi_{\mathsf{ref}}(x))$, which decreases with $M$— and the sample complexity required to achieve it, which increases with $M$. Indeed, when we later prove Theorem 3.1 and Theorem 3.4 later, we will choose $M$ to make the RHS of Lemma F.1 tight when $\mathcal{E}_M(\pi(x), \pi_{\mathsf{ref}}(x))$ is translated to our coverage coefficients of interest.

**Proof sketch.** The high-level idea of the proof is to use as an intermediate comparator the response distribution $\pi_{\mathsf{R}}$ induced simulating sampling from $\pi^\star$ via RejectionSampling$_{N,M}(\frac{\pi^\star}{\pi_{\mathsf{ref}}}; \pi_{\mathsf{ref}}, x)$, with the parameter $M$ from the lemma statement. The utility of using such a comparator is that, because the BoN procedure always chooses the response with largest reward label, $\widehat{\pi}_{\mathsf{BoN}}$ is always able to compete with $\pi_{\mathsf{R}}$ under $\widehat{r}$ (in the sense of having larger expected estimated reward).

We can then translate this observation to the desired bound by recalling that $\pi_{\mathsf{R}}$ approximates $\pi^\star$ in total variation distance (from Lemma D.4), which means that $\widehat{\pi}_{\mathsf{BoN}}$ is also approximately as good as $\pi^\star$ under $\widehat{r}$. Lastly, we translate the performance under $\widehat{r}$ to the performance under the true reward $r^\star$, which is penalized by the reward estimation error $\varepsilon_{\mathsf{RM}}(x)$ on the BoN response distribution, and the gap between the two is quantified by the reward estimation error under the distribution of responses drawn from $\widehat{\pi}_{\mathsf{BoN}}$, which is the source of the $\sqrt{N}$ term.

**Proof of Lemma F.1.** For simplicity in this proof, we will assume WLOG that $\widehat{r}(y)$ is unique for all $y \in \mathcal{Y}$, otherwise we can perturb $\widehat{r}(y)$ with a miniscule $\varepsilon(y) \ll \varepsilon_{\mathsf{RM}}$, as long as it is within floating-point precision to break ties.

Let $\pi_{\mathsf{R}}^\star(x)$ be the distribution of responses induced by running RejectionSampling$_{N,M}(\frac{\pi^\star}{\pi_{\mathsf{ref}}}; \pi_{\mathsf{ref}}, x)$ which we use to decompose the regret as follows:

$$
J(\pi^\star; x) - J(\widehat{\pi}_{\mathsf{BoN}}; x) = J(\pi^\star; x) - J(\pi_{\mathsf{R}}^\star; x) + J(\pi_{\mathsf{R}}^\star; x) - J(\widehat{\pi}_{\mathsf{BoN}}; x)
$$
$$
\leq R_{\max} \cdot D_{\mathsf{TV}}(\pi^\star(x), \pi_{\mathsf{R}}^\star(x)) + J(\pi_{\mathsf{R}}^\star; x) - J(\widehat{\pi}_{\mathsf{BoN}}; x).
$$

We next bound the last pair of terms as a function of the reward estimation error. Below, we use $\mathbb{E}_{\widehat{\mathcal{Y}}_N \sim \pi_{\mathsf{ref}}(x)}[\cdot]$ to refer to expectations over $N$ samples $\widehat{\mathcal{Y}}_N = (y_1, \ldots, y_N)$ drawn i.i.d. from $\pi_{\mathsf{ref}}(x)$, and, given $\widehat{\mathcal{Y}}_N$, we use $\mathbb{E}_{y \sim \pi_{\mathsf{R}}^\star(x)|\widehat{\mathcal{Y}}_N}[\cdot]$ to refer to the expectation over responses induced by running
$\texttt{RejectionSampling}_{N,M}(\pi^\star/\pi_{\mathsf{ref}}; \pi_{\mathsf{ref}}, x)$ conditioned on the realization of the set $\widehat{\mathcal{Y}}_N = (y_1, \ldots, y_N)$ drawn by the algorithm. We define $\mathbb{E}_{y \sim \widehat{\pi}_{\mathsf{BoN}}(x)|\widehat{\mathcal{Y}}_N}[\cdot]$ analogously.

We begin by decomposing the second term above as follows:

$$
\begin{aligned}
J(\pi_{\mathsf{R}}^\star; x) - J(\widehat{\pi}_{\mathsf{BoN}}; x) &= \mathbb{E}_{y \sim \pi_{\mathsf{R}}^\star(x)}[r^\star(x, y) - \widehat{r}(x, y)] + \mathbb{E}_{\widehat{\mathcal{Y}}_N \sim \pi_{\mathsf{ref}}(x)}\left[\mathbb{E}_{y \sim \pi_{\mathsf{R}}^\star(x)|\widehat{\mathcal{Y}}_N}[\widehat{r}(x, y)] - \mathbb{E}_{y \sim \widehat{\pi}_{\mathsf{BoN}}(x)|\widehat{\mathcal{Y}}_N}[\widehat{r}(x, y)]\right] \\
&\quad + \mathbb{E}_{y \sim \widehat{\pi}_{\mathsf{BoN}}(x)}[\widehat{r}(x, y) - r^\star(x, y)] \\
&\leq \mathbb{E}_{y \sim \pi_{\mathsf{R}}^\star(x)}[r^\star(x, y) - \widehat{r}(x, y)] + \mathbb{E}_{y \sim \widehat{\pi}_{\mathsf{BoN}}(x)}[\widehat{r}(x, y) - r^\star(x, y)] \\
&\leq \underbrace{\mathbb{E}_{y \sim \pi_{\mathsf{R}}^\star(x)}[|r^\star(x, y) - \widehat{r}(x, y)|]}_{(\mathrm{T1})} + \underbrace{\mathbb{E}_{y \sim \widehat{\pi}_{\mathsf{BoN}}(x)}[|\widehat{r}(x, y) - r^\star(x, y)|]}_{(\mathrm{T2})}.
\end{aligned}
$$

Note that above we use the linearity of expectation in the first inequality, so that

$$
\mathbb{E}_{\widehat{\mathcal{Y}}_N \sim \pi_{\mathsf{ref}}(x)}\left[\mathbb{E}_{y \sim \pi_{\mathsf{R}}^\star(x)|\widehat{\mathcal{Y}}_N}[\widehat{r}(x, y)] - \mathbb{E}_{y \sim \widehat{\pi}_{\mathsf{BoN}}(x)|\widehat{\mathcal{Y}}_N}[\widehat{r}(x, y)]\right]
$$

couples the set $\widehat{\mathcal{Y}}_N$ drawn by the two algorithms, and compares the performance of $\widehat{\pi}_{\mathsf{BoN}}$ and $\pi_{\mathsf{R}}^\star$ for a fixed set of $N$ responses. Then, because the BoN policy always chooses the response with the largest value under $\widehat{r}$ for any fixed $\widehat{\mathcal{Y}}_N$, i.e., $\mathbb{I}[y \in \mathrm{supp}(\widehat{\pi}_{\mathsf{BoN}}(\cdot \mid x))] = \mathbb{I}\left[\widehat{r}(x, y) \geq \widehat{r}(x, y'), \ \forall y' \in \widehat{\mathcal{Y}}_N\right]$, it can be seen that

$$
\mathbb{E}_{y \sim \pi_{\mathsf{R}}(x)|\widehat{\mathcal{Y}}_N}[\widehat{r}(x, y)] - \mathbb{E}_{y \sim \widehat{\pi}_{\mathsf{BoN}}(x)|\widehat{\mathcal{Y}}_N}[\widehat{r}(x, y)] \leq 0.
$$

For (T2), we first show that $\mathcal{C}^{\widehat{\pi}_{\mathsf{BoN}}}(x) \leq \mathcal{C}_\infty^{\widehat{\pi}_{\mathsf{BoN}}}(x) \leq N$. Letting $\sum_{\widehat{\mathcal{Y}}_N \sim \pi_{\mathsf{ref}}(x)}$ refer to the sum over all possible sequences of $N$ responses $\widehat{\mathcal{Y}}_N = (y_1, \ldots, y_N) \in \mathcal{Y}^N$, and $\pi_{\mathsf{ref}}(y_1, \ldots, y_N \mid x) = \prod_{i=1}^N \pi_{\mathsf{ref}}(y_i \mid x)$ to be its probability, we can express the BoN policy in closed form as

$$
\widehat{\pi}_{\mathsf{BoN}}(y \mid x) = \sum_{\widehat{\mathcal{Y}}_N \sim \pi_{\mathsf{ref}}(x)} \pi_{\mathsf{ref}}(y_1, \ldots, y_N \mid x) \cdot \mathbb{I}\left[y \in \widehat{\mathcal{Y}}_N\right] \cdot \mathbb{I}\left[\widehat{r}(x, y) \geq \widehat{r}(x, y'), \ \forall y' \in \widehat{\mathcal{Y}}_N\right],
$$

since, conditioned on a set of samples $\widehat{\mathcal{Y}}_N$, the BoN algorithm deterministically outputs the one with the largest $\widehat{r}$ value. The base policy $\pi_{\mathsf{ref}}$ can be written in a similar form, by marginalizing over the process through which we sample $\widehat{\mathcal{Y}}_N$, then sample $y$ uniformly from this set:

$$
\pi_{\mathsf{ref}}(y \mid x) = \sum_{\widehat{\mathcal{Y}}_N \sim \pi_{\mathsf{ref}}(x)} \pi_{\mathsf{ref}}(y_1, \ldots, y_N \mid x) \cdot \sum_{y' \in \widehat{\mathcal{Y}}_N} \frac{\mathbb{I}[y' = y]}{N}.
$$

Then for any $y$, we can upper bound the likelihood ratio between the BoN policy and $\pi_{\mathsf{ref}}$ by $N$,

$$
\begin{aligned}
\frac{\widehat{\pi}_{\mathsf{BoN}}(y \mid x)}{\pi_{\mathsf{ref}}(y \mid x)} &= N \cdot \frac{\sum_{\widehat{\mathcal{Y}}_N \sim \pi_{\mathsf{ref}}(x)} \pi_{\mathsf{ref}}(y_1, \ldots, y_N \mid x) \cdot \mathbb{I}\left[y \in \widehat{\mathcal{Y}}_N\right] \cdot \mathbb{I}\left[\widehat{r}(x, y) \geq \widehat{r}(x, y'), \ \forall y' \in \widehat{\mathcal{Y}}_N\right]}{\sum_{\widehat{\mathcal{Y}}_N \sim \pi_{\mathsf{ref}}(x)} \pi_{\mathsf{ref}}(y_1, \ldots, y_N \mid x) \cdot \sum_{y' \in \widehat{\mathcal{Y}}_N} \mathbb{I}[y' = y]} \\
&\leq N \cdot \frac{\sum_{\widehat{\mathcal{Y}}_N \sim \pi_{\mathsf{ref}}(x)} \pi_{\mathsf{ref}}(y_1, \ldots, y_N \mid x) \cdot \mathbb{I}\left[y \in \widehat{\mathcal{Y}}_N\right] \cdot \mathbb{I}\left[\widehat{r}(x, y) \geq \widehat{r}(x, y'), \ \forall y' \in \widehat{\mathcal{Y}}_N\right]}{\sum_{\widehat{\mathcal{Y}}_N \sim \pi_{\mathsf{ref}}(x)} \pi_{\mathsf{ref}}(y_1, \ldots, y_N \mid x) \cdot \mathbb{I}\left[y \in \widehat{\mathcal{Y}}_N\right]} \\
&\leq N.
\end{aligned}
$$

Now, to bound the reward estimation error in (T2), we combine this result with the Cauchy-Schwarz inequality, giving

$$
(\mathrm{T2}) \leq \sqrt{\mathcal{C}^{\widehat{\pi}_{\mathsf{BoN}}}(x) \cdot \mathbb{E}_{\pi_{\mathsf{ref}}(x)}\left[(\widehat{r}(x, y) - r^\star(x, y))^2\right]} \leq \sqrt{N \cdot \varepsilon_{\mathsf{RM}}^2(x)}.
$$

For (T1), we leverage results from Appendix D. Recall the random event from Eq. (14) random draws of $\widehat{\mathcal{Y}}_N$ and $\widehat{\xi}_N = (\xi_1, \dots, \xi_N)$, under which Algorithm 5 returns a response in Line 5,

$$\mathrm{stop} := \{\exists i^\star \in [N] \text{ s.t. } \xi_{i^\star} = 1\}.$$

From the proof of Lemma D.4 (Appendix D.2), recall that we can write the induced policy as

$$\pi_{\mathsf{R}}^\star(y \mid x) = \mathbb{P}_{\widehat{\mathcal{Y}}_N, \widehat{\xi}_N}(y \mid x, \mathrm{stop}) \cdot \mathbb{P}_{\widehat{\mathcal{Y}}_N, \widehat{\xi}_N}(\mathrm{stop} \mid x) + \mathbb{P}_{\widehat{\mathcal{Y}}_N, \widehat{\xi}_N}(y \mid x, \neg\mathrm{stop}) \cdot \mathbb{P}_{\widehat{\mathcal{Y}}_N, \widehat{\xi}_N}(\neg\mathrm{stop} \mid x).$$

On the event of $\neg\mathrm{stop}$, Algorithm 5 returns a randomly drawn response $y_{N+1} \sim \pi_{\mathsf{ref}}(\cdot \mid x)$ in Line 6, thus

$$\mathbb{P}_{\widehat{\mathcal{Y}}_N, \widehat{\xi}_N}(y \mid x, \neg\mathrm{stop}) = \pi_{\mathsf{ref}}(y \mid x)$$

As a result,

$$\begin{aligned}
\pi_{\mathsf{R}}^\star(y \mid x) &= \mathbb{P}_{\widehat{\mathcal{Y}}_N, \widehat{\xi}_N}(\mathrm{stop} \mid x) \cdot \mathbb{P}_{\widehat{\mathcal{Y}}_N, \widehat{\xi}_N}(y \mid x, \mathrm{stop}) + \mathbb{P}_{\widehat{\mathcal{Y}}_N, \widehat{\xi}_N}(\neg\mathrm{stop} \mid x) \cdot \pi_{\mathsf{ref}}(y \mid x) \\
&\leq \mathbb{P}_{\widehat{\mathcal{Y}}_N, \widehat{\xi}_N}(y \mid x, \mathrm{stop}) + \frac{1}{2} \cdot \pi_{\mathsf{ref}}(y \mid x) \\
&= \frac{\min\{\pi^\star(y \mid x), M \cdot \pi_{\mathsf{ref}}(y \mid x)\}}{1 - \mathcal{E}_M(\pi^\star(x), \pi_{\mathsf{ref}}(x))} + \frac{1}{2} \cdot \pi_{\mathsf{ref}}(y \mid x) \\
&\leq 2 \cdot \min\{\pi^\star(y \mid x), M \cdot \pi_{\mathsf{ref}}(y \mid x)\} + \frac{1}{2} \cdot \pi_{\mathsf{ref}}(y \mid x)
\end{aligned}$$

where in the last inequality we have used the assumption that $\mathcal{E}_M(\pi^\star(x), \pi_{\mathsf{ref}}(x)) \leq \frac{1}{2}$, and in the first we use the observation that $\mathbb{P}_{\widehat{\mathcal{Y}}_N, \widehat{\xi}_N}(\neg\mathrm{stop} \mid x) \leq \frac{1}{2}$ since $\mathbb{P}_{\widehat{\mathcal{Y}}_N, \widehat{\xi}_N}(\mathrm{stop} \mid x) = \frac{1 - \mathcal{E}_M(\pi^\star(x), \pi_{\mathsf{ref}}(x))}{M}$ and $M \geq 1$. We can then use Cauchy-Schwarz to bound

$$\begin{aligned}
(\mathrm{T1}) &= \mathbb{E}_{y \sim \pi_{\mathsf{R}}^\star(x)}[|r^\star(x, y) - \widehat{r}(x, y)|] \\
&\leq 2 \cdot \mathbb{E}_{y \sim \pi^\star(x)}[|r^\star(x, y) - \widehat{r}(x, y)|] + \frac{1}{2} \cdot \mathbb{E}_{y \sim \pi_{\mathsf{ref}}(x)}[|r^\star(x, y) - \widehat{r}(x, y)|] \\
&\leq 2 \cdot \sqrt{\mathcal{C}^{\pi^\star}(x) \cdot \varepsilon_{\mathsf{RM}}^2(x)} + \frac{\varepsilon_{\mathsf{RM}}(x)}{2}
\end{aligned}$$

Combining all the preceding bounds, we obtain

$$J(\pi_{\mathsf{R}}^\star; x) - J(\widehat{\pi}_{\mathsf{BoN}}; x) \leq 2 \cdot \sqrt{\mathcal{C}^{\pi^\star}(x) \cdot \varepsilon_{\mathsf{RM}}^2(x)} + \frac{\varepsilon_{\mathsf{RM}}(x)}{2} + \sqrt{N \cdot \varepsilon_{\mathsf{RM}}^2(x)}$$

thus the regret is bounded as

$$J(\pi^\star; x) - J(\widehat{\pi}_{\mathsf{BoN}}; x) \leq R_{\max} \cdot D_{\mathsf{TV}}(\pi^\star(x), \pi_{\mathsf{R}}^\star(x)) + 2 \cdot \sqrt{\mathcal{C}^{\pi^\star}(x) \cdot \varepsilon_{\mathsf{RM}}^2(x)} + \frac{\varepsilon_{\mathsf{RM}}(x)}{2} + \sqrt{N \cdot \varepsilon_{\mathsf{RM}}^2(x)}.$$

Finally, we apply Lemma D.4, which bounds

$$D_{\mathsf{TV}}(\pi^\star(x), \pi_{\mathsf{R}}^\star(x)) \leq \mathcal{E}_M(\pi^\star(x), \pi_{\mathsf{ref}}(x)) + \exp\left(-\frac{N}{2M} \cdot (1 - \mathcal{E}_M(\pi^\star(x), \pi_{\mathsf{ref}}(x)))\right)$$

to give the lemma statement. $\qquad\square$

## F.2. General Lower Bounds on Regret

This section contains two regret lower bounds that apply to both BoN and InferenceTimePessimism across a range of parameter values, for a single prompt $x$. Each bound contains an information-theoretic component that applies to *any selection algorithm*, which is defined formally in Definition F.2, and takes the general form that if $N <$ (threshold), the regret of any selection algorithm will be at least $\mathrm{poly}(\mathcal{C}^{\pi^\star}(x), \varepsilon_{\mathsf{RM}}(x))$. The results also have a component that is specific to BoN, that when $N \geq$ (threshold), BoN has at least $\sqrt{N \cdot \varepsilon_{\mathsf{RM}}^2(x)}$ regret.

- Theorem F.3 in Appendix F.2.1 shows that $(\text{threshold}) \propto \mathcal{C}_\infty^\pi(x)$, and for smaller $N$ any algorithm pays $\sqrt{\mathcal{C}_\infty^{\pi^\star}(x) \cdot \varepsilon_{\mathsf{RM}}^2(x)}$ regret, and for larger $N$ BoN incurs $\sqrt{N \cdot \varepsilon_{\mathsf{RM}}^2(x)} \geq \sqrt{\mathcal{C}_\infty^{\pi^\star}(x) \cdot \varepsilon_{\mathsf{RM}}^2(x)}$ regret.

- Theorem F.4 in Appendix F.2.2 utilizes a construction where $\mathcal{C}_\infty^{\pi^\star}(x)$ is exponentially larger than $\mathcal{C}^{\pi^\star}(x)$, and any algorithm has regret at least $\left(\mathcal{C}^{\pi^\star}(x) \cdot \varepsilon\right)^p$ for a range of $\varepsilon \geq \varepsilon_{\mathsf{RM}}(x)$, unless $N \geq (\text{threshold}) \propto \left(\varepsilon_{\mathsf{RM}}^2(x)\right)^{-p}$. BoN again pays $\sqrt{N \cdot \varepsilon_{\mathsf{RM}}^2(x)}$ in this regime.

The latter result is later used to prove Theorem 3.2, where $p = \frac{1}{3}$ to balance the terms, and Theorem 4.2, where $p = \frac{1}{2}$.

**Proof techniques.** The information-theoretic component of the results reflects the difficulty of simulating a sample from the target policy's distribution, which is required for any selection algorithm $\mathcal{A}$ to be able to compete with the target policy. Recall that the lower bound for rejection sampling (Lemma D.6) states that any selection algorithm requires at least $\mathcal{M}_{x,\varepsilon}^{\pi^\star}$ samples to be $\varepsilon$-close to the target distribution in TV distance. To convert this result to regret lower bounds, we a) construct a pair of distributions for which the conversion from $\mathcal{M}_{x,\varepsilon}^{\pi^\star}$ to coverage coefficient $\mathcal{C}^\pi(x)$ in Proposition D.3 is tight, and b) specify reward functions $r^\star$ and $\widehat{r}$ so that the regret maximally witnesses the reward estimation error $\varepsilon_{\mathsf{RM}}(x)$ where rejection sampling fails to approximate $\pi^\star$, and where $\widehat{\pi}_{\mathsf{BoN}}$ overfits to $\widehat{r}$.

While the construction used for the $\mathcal{C}_\infty^{\pi^\star}(x)$ result is relatively simple and has $|\mathcal{Y}| = O(1)$, the construction for the $\mathcal{C}^{\pi^\star}(x)$ utilizes a countable infinite response space $\mathcal{Y}$, which is necessary to create the exponential separation between $\mathcal{C}^{\pi^\star}(x)$ and $\mathcal{C}_\infty^{\pi^\star}(x)$. If we label the responses $i = 1, 2, 3, \ldots$, the ratio $\frac{\pi^\star}{\pi_{\mathsf{ref}}}$ increases exponentially in $i$, and $r^\star$ increases in $i$ while the reward model $\widehat{r}$ decreasess. Because $\pi^\star$ here is an exponential tilting of $\pi_{\mathsf{ref}}$ with respect to the true reward, the lower bound construction reflects the structure of language modeling, where policies are parameterized as softmax functions and the response space is exponentially large, which we believe may be of independent interest.

**Preliminaries.** The lower bound constructions below utilize only a single prompt $x$, and in the proofs (not the theorem statements), we drop the $x$ dependence for notational compactness. For example, $\pi(y) \equiv \pi(y \mid x)$ since $\mathcal{X} = \{x\}$, $\mathcal{C}^\pi \equiv \mathcal{C}^\pi(x)$, etc. Lastly, we formally define what we mean by "any selection algorithm" in the sample-and-evaluate framework.

**Definition F.2** (Inference-time selection algorithm $\mathcal{A}$). *Under the sample-and-evaluate framework (Definition 2.2), an inference-time selection algorithm $\mathcal{A}$ is any mapping from $\widehat{\mathcal{Y}}_N = (y_1, \ldots, y_N) \sim \pi_{\mathsf{ref}}(\cdot \mid x)$ and $\{\widehat{r}(x, y_i)\}_{i \in [N]}$ to a response $y \in \widehat{\mathcal{Y}}_N$; we define $\pi_{\mathcal{A}} : \mathcal{X} \to \Delta(\mathcal{Y})$ to be the law over responses that the algorithm induces.*

### F.2.1. LOWER BOUNDS UNDER $L_\infty$-COVERAGE

**Theorem F.3** (Regret lower bound for $\mathcal{C}_\infty^{\pi^\star}$). *For any $\varepsilon_{\mathsf{RM}}^2 \in [0, 1]$ and $C \geq 1$, there exists a comparator policy $\pi^\star$ over contexts $\mathcal{X} = \{x\}$ and responses $\mathcal{Y}$ for which the following statements hold.*

1. *There exists a problem instance $(\pi_{\mathsf{ref}}, r^\star, \widehat{r})$ with $\varepsilon_{\mathsf{RM}}^2(x) \leq \varepsilon_{\mathsf{RM}}$ and*

$$\mathcal{C}_\infty^{\pi^\star}(x) = \mathcal{C}^{\pi^\star}(x) = C$$

   *such that, for any $N < \frac{C}{2}$, any inference-time selection algorithm $\mathcal{A}$ (Definition F.2) has regret*

$$J(\pi^\star; x) - J(\pi_{\mathcal{A}}; x) > \min\left\{ 2\sqrt{\mathcal{C}_\infty^{\pi^\star}(x) \cdot \varepsilon_{\mathsf{RM}}^2(x)}, \ 1 \right\}.$$

2. *For any $N \geq 1$, there exists a problem instance $(\pi_{\mathsf{ref}}, r^\star, \widehat{r})$ with $\varepsilon_{\mathsf{RM}}^2(x) \leq \varepsilon_{\mathsf{RM}}$ and*

$$\mathcal{C}_\infty^{\pi^\star}(x) = \mathcal{C}^{\pi^\star}(x) = C$$

   *such that* BoN-Alignment *suffers regret*

$$J(\pi^\star; x) - J(\widehat{\pi}_{\mathsf{BoN}}; x) \geq c \cdot \min\left\{ \sqrt{N \cdot \varepsilon_{\mathsf{RM}}^2(x)}, \ 1 \right\},$$

   *where $c$ is a universal constant.*

**Proof of Theorem F.3.** Because we condition on a single prompt $x$, we omit $x$ dependencies for ease of presentation; in particular, we abbreviate $\mathcal{M}_\varepsilon^{\pi^\star} \equiv \mathcal{M}_{x,\varepsilon}^{\pi^\star}$, and $\mathcal{E}_M(\pi^\star, \pi_{\mathsf{ref}}) \equiv \mathcal{E}_M(\pi^\star(x), \pi_{\mathsf{ref}}(x))$.

Fix $N$ and $\varepsilon_{\mathsf{RM}}$ and $C$. The response space is $\mathcal{Y} = \{y_0, y^\star, y_{\mathsf{bad}}\}$, the comparator policy $\pi^\star(y) = \mathbb{I}[y = y^\star]$ plays $y^\star$ deterministically, and the reference policy is

$$\pi_{\mathsf{ref}}(y_0) = 1 - \tfrac{1}{2N} - \tfrac{1}{C}, \quad \pi_{\mathsf{ref}}(y^\star) = \tfrac{1}{C}, \quad \pi_{\mathsf{ref}}(y_{\mathsf{bad}}) = \tfrac{1}{2N},$$

for which we have $\mathcal{C}_\infty^{\pi^\star} = \mathcal{C}^{\pi^\star} = C$. Next, for some $\varepsilon \geq 0$ recall Definition D.2,

$$\mathcal{M}_\varepsilon^{\pi^\star} := \min\{M \mid \mathcal{E}_M(\pi^\star, \pi_{\mathsf{ref}}) \leq \varepsilon\}.$$

For our choice of policies we can compute $\mathcal{M}_\varepsilon^{\pi^\star}$ in closed form, since for any $M$,

$$\mathcal{E}_M(\pi^\star, \pi_{\mathsf{ref}}) = 1 - M \cdot \pi_{\mathsf{ref}}(y^\star) = 1 - \frac{M}{C}.$$

Then any $M \geq C \cdot (1 - \varepsilon)$ is sufficient to have $\mathcal{E}_M(\pi^\star, \pi_{\mathsf{ref}}) \leq \varepsilon$, therefore

$$\mathcal{M}_\varepsilon^{\pi^\star} = C \cdot (1 - \varepsilon). \tag{17}$$

**Part 1 (Small $N$).** Define the reward functions

$$
\begin{array}{lll}
r^\star(y_0) = 0 & r^\star(y^\star) = 1 & r^\star(y_{\mathsf{bad}}) = 0 \\
\widehat{r}(y_0) = 0 & \widehat{r}(y^\star) = 1 - \min\left\{\sqrt{C \cdot \varepsilon_{\mathsf{RM}}^2}, 1\right\} & \widehat{r}(y_{\mathsf{bad}}) = 0
\end{array}
$$

It is easy to see that

$$\mathbb{E}_{\pi_{\mathsf{ref}}}\left[(\widehat{r}(y) - r^\star(y))^2\right] \leq \frac{C \cdot \varepsilon_{\mathsf{RM}}^2}{C} = \varepsilon_{\mathsf{RM}}^2.$$

For some $\varepsilon$ that we will set shortly, when $N < \mathcal{M}_{x,\varepsilon}^{\pi^\star}$ the regret is lower bounded as

$$
\begin{aligned}
J(\pi^\star) - J(\widehat{\pi}_{\mathsf{BoN}}) &= \pi^\star(y^\star) - \widehat{\pi}_{\mathsf{BoN}}(y^\star) \\
&\geq 1 - \mathbb{P}\left(y^\star \in \widehat{\mathcal{Y}}_N\right) \\
&\geq 2\mathcal{E}_{\mathcal{M}_\varepsilon^{\pi^\star}}(\pi^\star, \pi_{\mathsf{ref}}).
\end{aligned}
$$

Then setting $\varepsilon = \min\left\{\sqrt{C \cdot \varepsilon_{\mathsf{RM}}^2}, \tfrac{1}{2}\right\}$ and using the closed form of $\mathcal{M}_{x,\varepsilon}^{\pi^\star}$ in Eq. (17), Lemma D.6 states that when

$$N < \mathcal{M}_\varepsilon^{\pi^\star} = C \cdot \left(1 - \min\left\{\sqrt{C \cdot \varepsilon_{\mathsf{RM}}^2}, \tfrac{1}{2}\right\}\right) = C \cdot \max\left\{1 - \sqrt{C \cdot \varepsilon_{\mathsf{RM}}^2}, \tfrac{1}{2}\right\},$$

we have $\mathcal{E}_{\mathcal{M}_\varepsilon^{\pi^\star}}(\pi^\star, \pi_{\mathsf{ref}}) > \varepsilon = \min\left\{\sqrt{C \cdot \varepsilon_{\mathsf{RM}}^2}, \tfrac{1}{2}\right\}$, and thus

$$J(\pi^\star) - J(\widehat{\pi}_{\mathsf{BoN}}) > \min\left\{2\sqrt{C \cdot \varepsilon_{\mathsf{RM}}^2}, 1\right\}.$$

**Part 2 (Large $N$).** Define the gap on $y_{\mathsf{bad}}$ to be $\Delta := \min\left\{1, \sqrt{N \cdot \varepsilon_{\mathsf{RM}}^2}\right\}$, and set the reward functions

$$
\begin{array}{lll}
r^\star(y_0) = 0 & r^\star(y^\star) = 1 & r^\star(y_{\mathsf{bad}}) = 1 - \Delta \\
\widehat{r}(y_0) = 0 & \widehat{r}(y^\star) = 1 - \min\left\{\sqrt{\tfrac{C}{2} \cdot \varepsilon_{\mathsf{RM}}^2}, 1\right\} & \widehat{r}(y_{\mathsf{bad}}) = 1
\end{array}
$$

and we can check that

$$\mathbb{E}_{\pi_{\mathsf{ref}}}\left[(\widehat{r}(y) - r^\star(y))^2\right] \leq \frac{C \cdot \varepsilon_{\mathsf{RM}}^2}{2C} + \frac{\Delta^2}{2N} = \frac{\varepsilon_{\mathsf{RM}}^2}{2} + \frac{\min\{1, N \cdot \varepsilon_{\mathsf{RM}}^2\}}{2N} \leq \varepsilon_{\mathsf{RM}}^2.$$

Note that for this construction, $\widehat{r}(y_{\mathsf{bad}})$ is the largest reward under $\widehat{r}$, so $\widehat{\pi}_{\mathsf{BoN}}$ will always play $y_{\mathsf{bad}}$ if $y_{\mathsf{bad}} \in \widehat{\mathcal{Y}}_N$, which is an event that occurs with at least constant probability under our choice for $\pi_{\mathsf{ref}}$

$$\mathbb{P}(y_{\mathsf{bad}} \in \widehat{\mathcal{Y}}_N) = 1 - \left(1 - \frac{1}{2N}\right)^N \geq 1 - e^{-\frac{1}{2}}.$$

Using this, we lower bound the regret as

$$
\begin{aligned}
J(\pi^\star) - J(\widehat{\pi}_{\mathsf{BoN}}) &= \mathbb{P}(y_{\mathsf{bad}} \in \widehat{\mathcal{Y}}_N) \cdot \mathbb{E}_{\widehat{\mathcal{Y}}_N}\Big[J(\pi^\star) - J(\widehat{\pi}_{\mathsf{BoN}}) \mid y_{\mathsf{bad}} \in \widehat{\mathcal{Y}}_N\Big] \\
&\quad + \mathbb{P}(y_{\mathsf{bad}} \notin \widehat{\mathcal{Y}}_N) \cdot \mathbb{E}_{\widehat{\mathcal{Y}}_N}\Big[J(\pi^\star) - J(\widehat{\pi}_{\mathsf{BoN}}) \mid y_{\mathsf{bad}} \notin \widehat{\mathcal{Y}}_N\Big] \\
&> \mathbb{P}(y_{\mathsf{bad}} \in \widehat{\mathcal{Y}}_N) \cdot \mathbb{E}_{\widehat{\mathcal{Y}}_N}\Big[J(\pi^\star) - J(\widehat{\pi}_{\mathsf{BoN}}) \mid y_{\mathsf{bad}} \in \widehat{\mathcal{Y}}_N\Big] \\
&= \mathbb{P}(y_{\mathsf{bad}} \in \widehat{\mathcal{Y}}_N) \cdot \Delta \\
&\geq c \cdot \min\left\{1, \sqrt{N \cdot \varepsilon_{\mathsf{RM}}^2}\right\},
\end{aligned}
$$

where in the first inequality we use the fact that $\pi^\star$ is optimal for $r^\star$, and in the last inequality we plug in the definition of $\Delta$. $\qquad\square$

### F.2.2. LOWER BOUNDS UNDER $L_1$-COVERAGE

**Theorem F.4** (Regret lower bound for $\mathcal{C}^{\pi^\star}$)**.** *For any $\varepsilon_{\mathsf{RM}} \in (0, 1/4]$ and $C \geq \varepsilon_{\mathsf{RM}}^{-1}$, there exists a comparator policy $\pi^\star$ over contexts $\mathcal{X} = \{x\}$ and response space $\mathcal{Y} = \mathbb{Z}^+$, and universal constants $c_1, c_2, c_3$ such that the following statements hold for any $p \in (0, 1/2]$.*

1. *For any $\varepsilon \in [\varepsilon_{\mathsf{RM}}, \frac{1}{4}]$, there exists a problem instance $(\pi_{\mathsf{ref}}, r^\star, \widehat{r})$ with $\varepsilon_{\mathsf{RM}}^2(x) \leq \varepsilon_{\mathsf{RM}}$ and*

$$
\begin{aligned}
\mathcal{C}^{\pi^\star}(x) &= O(\log C), \\
\mathcal{C}_\infty^{\pi^\star}(x) &= O(C),
\end{aligned}
$$

   *such that, for any $N < c_1 \cdot \left(\mathcal{C}^{\pi^\star}(x) \cdot \varepsilon^2\right)^{-p}$, any selection algorithm $\mathcal{A}$ (Definition F.2) suffers regret*

$$J(\pi^\star; x) - J(\pi_{\mathcal{A}}; x) > c_2 \cdot \left(\mathcal{C}^{\pi^\star}(x) \cdot \varepsilon^2\right)^p.$$

2. *For any $N \gtrsim 1$, there exists a problem instance $(\pi_{\mathsf{ref}}, r^\star, \widehat{r})$ with $\varepsilon_{\mathsf{RM}}^2(x) \leq \varepsilon_{\mathsf{RM}}$ and*

$$
\begin{aligned}
\mathcal{C}^{\pi^\star}(x) &= O(\log C) \\
\mathcal{C}_\infty^{\pi^\star}(x) &= O(C),
\end{aligned}
$$

   *such that the* BoN-Alignment *policy $\widehat{\pi}_{\mathsf{BoN}}$ has regret*

$$J(\pi^\star; x) - J(\widehat{\pi}_{\mathsf{BoN}}; x) > c_3 \cdot \sqrt{N \cdot \varepsilon_{\mathsf{RM}}^2(x)}.$$

**Proof of Theorem F.4.** As in the proof of Theorem F.3, $x$-dependencies are ommitted in the proof below, inclusive of complexity measures such as $\mathcal{E}_M(\pi^\star(x), \pi_{\mathsf{ref}}(x))$ and $\varepsilon_{\mathsf{RM}}(x)$, since there is only a single prompt.

**Part 1** ($N$ small). We prove the statement for $\mathcal{C}^{\pi^\star} \leq \varepsilon_{\mathsf{RM}}^{-2}$, otherwise $\sqrt{\mathcal{C}^{\pi^\star} \cdot \varepsilon_{\mathsf{RM}}^2} = \Omega(1)$.

For all $\varepsilon \in \left[\varepsilon_{\mathsf{RM}}, \frac{1}{4}\right]$, we will define $\pi_{\mathsf{ref}}$ and $\pi^\star$ to be the distributions from the construction in Lemma F.5 with $C$. These policies are defined as follows for all $i \in \mathcal{Y} = \{1, 2, \ldots\}$ (where we use $i$ instead of $y$ to index responses).

$$
\begin{aligned}
\pi_{\mathsf{ref}}(i) &= \frac{3}{4^i} \\
\pi^\star(i) &= \begin{cases} \frac{2^i}{3} \cdot \pi_{\mathsf{ref}}(i), & \text{if } i \leq I := \lceil \log C \rceil, \\ 2^I \cdot \pi_{\mathsf{ref}}(i), & \text{otherwise.} \end{cases}
\end{aligned}
$$

From the proof of Lemma F.5, we know that

$$\mathcal{C}^{\pi^\star} = O(\log C), \quad \text{and} \quad \mathcal{C}^{\pi^\star}_\infty = O(C).$$

Now fix a choice of $\varepsilon \in \left[\varepsilon_{\mathsf{RM}}, \frac{1}{4}\right]$. We will now define the reward functions for the construction. Let $k_\varepsilon \in \mathcal{Y}$ is an index that will be specified shortly, and, for $i \in \mathcal{Y}$, define $\Delta_\varepsilon(i) = 2^i \cdot \sqrt{\frac{\varepsilon^2_{\mathsf{RM}}}{4 \cdot k_\varepsilon}}$, and

$$r^\star(i) = \begin{cases} 0 & \text{if } i < k_\varepsilon, \\ \frac{1}{2} + \frac{\Delta_\varepsilon(k_\varepsilon)}{2} & \text{otherwise.} \end{cases} \qquad \widehat{r}(i) = \begin{cases} \Delta_\varepsilon(i) & \text{if } i < k_\varepsilon, \\ \frac{1}{2} - \frac{\Delta_\varepsilon(k_\varepsilon)}{2} & \text{otherwise.} \end{cases}$$

For this choice of $\widehat{r}$ and $r^\star$, we verify that the estimation error is upper bounded as $\varepsilon^2_{\mathsf{RM}}$,

$$\mathbb{E}_{\pi_{\mathsf{ref}}}\left[(\widehat{r}(i) - r^\star(i))^2\right] = 3\left(\sum_{i=1}^{k_\varepsilon} 4^{-i} \cdot \Delta^2_\varepsilon(i) + \Delta^2_\varepsilon(k_\varepsilon) \sum_{i=k_\varepsilon+1}^\infty 4^{-i}\right)$$

$$= 3 \cdot k_\varepsilon \cdot \frac{\varepsilon^2_{\mathsf{RM}}}{4 \cdot k_\varepsilon} + \left(\frac{4^{k_\varepsilon} \cdot \varepsilon^2_{\mathsf{RM}}}{4 \cdot k_\varepsilon}\right) \cdot 4^{-k_\varepsilon}$$

$$= \frac{\varepsilon^2_{\mathsf{RM}}}{4}\left(3 + \frac{1}{k_\varepsilon}\right)$$

$$\leq \varepsilon^2_{\mathsf{RM}}.$$

Next, we set

$$k_\varepsilon = \left\lfloor \log\left(\left(\mathcal{C}^{\pi^\star} \cdot \varepsilon^2\right)^{-p}\right)\right\rfloor.$$

We can check that $k_\varepsilon \leq I = \lceil \log C \rceil$, again using the precondition that $C \geq \frac{1}{\varepsilon_{\mathsf{RM}}} \geq \frac{1}{\varepsilon}$,

$$k_\varepsilon \leq \left\lfloor \log\left(\frac{1}{(\mathcal{C}^{\pi^\star} \cdot \varepsilon^2_{\mathsf{RM}})^p}\right)\right\rfloor \leq \left\lfloor \log\left(\frac{1}{\varepsilon_{\mathsf{RM}}}\right)\right\rfloor \leq \lfloor \log(C) \rfloor.$$

Then Lemma D.6 applied with $\varepsilon' = 2^{-k_\varepsilon} = c \cdot \left(\mathcal{C}^{\pi^\star} \cdot \varepsilon^2\right)^p$, where $c$ is an absolute constant, states that, if $N < c \cdot \left(\mathcal{C}^{\pi^\star} \cdot \varepsilon^2\right)^{-p}$, any selection algorithm (Definition F.2) has

$$D_{\mathsf{TV}}(\pi^\star, \pi_\mathcal{A}) \geq \mathcal{E}_{\frac{1}{\varepsilon'}}(\pi^\star, \pi_{\mathsf{ref}}) \gtrsim \left(\mathcal{C}^{\pi^\star} \cdot \varepsilon^2\right)^p. \tag{18}$$

We conclude by lower bounding the regret using Eq. (18). When $N < c \cdot \left(\mathcal{C}^{\pi^\star} \cdot \varepsilon^2\right)^{-p}$,

$$J(\pi^\star) - J(\pi_\mathcal{A}) = \sum_{i=1}^{k_\varepsilon} r^\star(i) \cdot (\pi^\star(i) - \pi_\mathcal{A}(i)) + r^\star(k_\varepsilon) \cdot \sum_{i=k_\varepsilon+1}^\infty (\pi^\star(i) - \pi_\mathcal{A}(i))$$

$$= r^\star(k_\varepsilon) \cdot \sum_{i=k_\varepsilon+1}^\infty (\pi^\star(i) - \pi_\mathcal{A}(i))$$

$$\geq \left(1 + \frac{\Delta_\varepsilon(k_\varepsilon)}{2}\right) \cdot \mathcal{E}_{\frac{1}{\varepsilon'}}(\pi^\star, \pi_{\mathsf{ref}})$$

$$> c_2 \cdot \left(\mathcal{C}^{\pi^\star} \cdot \varepsilon^2\right)^p,$$

where we have applied Eq. (18) in the last line, and in the first inequality we use the definition of $\mathcal{E}_M(\pi^\star, \pi_{\mathsf{ref}})$ with $M = \frac{1}{\varepsilon'} = 2^{k_\varepsilon}$, since $\{k_\varepsilon + 1, \ldots\} = \{i : \frac{\pi^\star(i)}{\pi_{\mathsf{ref}}(i)} \geq 2^{k_\varepsilon}\}$.

**Part 2 ($N$ large).** Fix $N \in \mathbb{Z}^+$. We prove the result for $N \lesssim \frac{1}{\varepsilon^2_{\mathsf{RM}}}$, otherwise the stated bound holds trivially. Let $k_N = \lfloor \log_4(N) \rfloor \leq I := \lceil \log C \rceil$, so that $\pi_{\mathsf{ref}}(k_N) \geq \frac{1}{N}$.

Next, let $\Delta_N(i) := 2^i \sqrt{\frac{\varepsilon_{\mathrm{RM}}^2}{8 \cdot k_N}}$, and define the reward functions

$$
r^\star(i) = \begin{cases} \frac{1}{2} + \frac{\Delta_N(i)}{2} & \text{if } i < k_N, \\ \frac{1}{2} - \frac{\sqrt{k_N} \cdot \Delta_N(k_N)}{2} & \text{if } i = k_N, \\ \frac{1}{2} + \frac{\sqrt{k_N} \cdot \Delta_N(k_N)}{2} & \text{otherwise.} \end{cases} \qquad \widehat{r}(i) = \begin{cases} \frac{1}{2} - \frac{\Delta_N(i)}{2} & \text{if } i < k_N, \\ \frac{1}{2} + \frac{\sqrt{k_N} \cdot \Delta_N(k_N)}{2} & \text{if } i = k_N, \\ \frac{1}{2} - \frac{\sqrt{k_N} \cdot \Delta_N(k_N)}{2} & \text{otherwise.} \end{cases}
$$

First, we check the reward ranges, and observe that $\sqrt{k_N} \cdot \Delta_N(k_N) = \sqrt{\frac{1}{8} \cdot N \cdot \varepsilon_{\mathrm{RM}}^2} \le 1$. For the reward model error, we have

$$
\begin{aligned}
\mathbb{E}_{\pi_{\mathrm{ref}}}\left[(\widehat{r}(i) - r^\star(i))^2\right] &= 3\left(\sum_{i=1}^{k_N-1} 4^{-i} \cdot \Delta_N^2(i) + k_N \cdot \Delta_{k_N}^2(k_N) \sum_{i=k_N}^{\infty} 4^{-i}\right) \\
&= \frac{3(k_N-1) \cdot \varepsilon_{\mathrm{RM}}^2}{8 k_N} + k_N \cdot \Delta_N^2(k_N) \cdot 4^{1-k_N} \\
&= \frac{\varepsilon_{\mathrm{RM}}^2}{8}\left(\frac{3(k_N-1)}{k_N} + 4\right) \\
&\le \varepsilon_{\mathrm{RM}}^2.
\end{aligned}
$$

Second, we show that $J(\pi^\star) - J(\pi_{\mathrm{ref}}) \ge 0$ as long as $k_N \ge 4$ by computing the policies in closed form. Recall that $\pi^\star(i) = 2^{-i}$ if $i \le I$, and $\pi^\star(i) = 3 \cdot 2^I \cdot 4^{-i}$ otherwise. Then by plugging in this expression and the definition of $r^\star$ above, we calculate its return as

$$
\begin{aligned}
J(\pi^\star) = &\sum_{i=1}^{k_N-1} 2^{-i} \cdot \left(\frac{1}{2} + \frac{\Delta_N(i)}{2}\right) + \frac{1}{2} \cdot \left(2^{-k_N} + \sum_{i=k_N+1}^{I} 2^{-i} + 3 \cdot 2^I \sum_{i=I+1}^{\infty} 4^{-i}\right) \\
&+ \frac{\sqrt{k_N} \cdot \Delta_N(k_N)}{2} \cdot \left(\sum_{i=k_N+1}^{I} 2^{-i} + 3 \cdot 2^I \sum_{i=I+1}^{\infty} 4^{-i} - 2^{-k_N}\right)
\end{aligned}
$$

Recall that

$$
1 = \sum_{i \in \mathcal{Y}} \pi^\star(y) = \sum_{i=1}^{I} 2^{-i} + 3 \cdot 2^I \sum_{i=I+1}^{\infty} 4^{-i}.
$$

Then grouping terms and substituting this identity,

$$
\begin{aligned}
J(\pi^\star) &= \frac{1}{2}\left(1 + \sum_{i=1}^{k_N-1} 2^{-i} \Delta_N(i) + \sqrt{k_N} \cdot \Delta_N(k_N) \cdot \left(1 - \sum_{i=1}^{k_N} 2^{-i} - 2^{-k_N}\right)\right) \\
&= \frac{1}{2}\left(1 + \sum_{i=1}^{k_N-1} 2^{-i} \cdot 2^i \sqrt{\frac{\varepsilon_{\mathrm{RM}}^2}{8 \cdot k_N}} + 2^{k_N} \sqrt{\frac{\varepsilon_{\mathrm{RM}}^2}{8}} \cdot \left(1 - \sum_{i=1}^{k_N} 2^{-i} - 2^{-k_N}\right)\right) \\
&= \frac{1}{2}\left(1 + (k_N-1)\sqrt{\frac{\varepsilon_{\mathrm{RM}}^2}{8 \cdot k_N}} + 2^{k_N} \sqrt{\frac{\varepsilon_{\mathrm{RM}}^2}{8}} \cdot \left(2^{-k_N} - 2^{-k_N}\right)\right) \\
&= \frac{1}{2} + (k_N-1)\sqrt{\frac{\varepsilon_{\mathrm{RM}}^2}{32 \cdot k_N}}.
\end{aligned}
$$

Also, for $\pi_{\text{ref}}$ we have

$$
\begin{aligned}
J(\pi_{\text{ref}}) &= 3\left( \sum_{i=1}^{k_N - 1} 4^{-i} \cdot \left( \frac{1}{2} + \frac{\Delta_N(i)}{2} \right) + \frac{1}{2} \cdot \left( 4^{-k_N} + \sum_{i=k_N+1}^{\infty} 4^{-i} \right) + \frac{\sqrt{k_N} \cdot \Delta_N(k_N)}{2} \cdot \left( \sum_{i=k_N+1}^{\infty} 4^{-i} - 4^{-k_N} \right) \right) \\
&= \frac{1}{2} + 3\sqrt{\frac{\varepsilon_{\text{RM}}^2}{32 k_N}} \cdot \sum_{i=1}^{k_N - 1} 2^{-i} + \frac{\sqrt{k_N} \cdot \Delta_N(k_N)}{2} \cdot \left( 4^{-k_N} - 4^{-k_N} \right) \\
&= \frac{1}{2} + 3\sqrt{\frac{\varepsilon_{\text{RM}}^2}{32 k_N}} \cdot \left( 1 - 2^{1-k_N} \right) \\
&\leq \frac{1}{2} + 3\sqrt{\frac{\varepsilon_{\text{RM}}^2}{32 k_N}}.
\end{aligned}
$$

Together, we obtain that

$$
J(\pi^\star) - J(\pi_{\text{ref}}) \geq (k_N - 4) \cdot \sqrt{\frac{\varepsilon_{\text{RM}}^2}{32 \cdot k_N}},
$$

which is nonnegative whenever $k_N = \lfloor \log_4 N \rfloor > 4$, or $N \geq 256$.

Lastly, we lower bound the regret by considering two cases.

- If $k_N \in \widehat{\mathcal{Y}}_N$, then BoN will choose $k_N$ since $k_N = \arg\max_i \widehat{r}(i)$, and

$$
\mathbb{E}\left[ J(\pi^\star) - J(\widehat{\pi}_{\text{BoN}}) \mid k_N \in \widehat{\mathcal{Y}}_N \right] \geq \frac{1}{2} - \left( \frac{1}{2} - \frac{\sqrt{k_N} \cdot \Delta_k}{2} \right) = \frac{\sqrt{k_N} \cdot \Delta_N(k_N)}{2} = \sqrt{\frac{1}{32} \cdot N \cdot \varepsilon_{\text{RM}}^2}.
$$

- If $k_N \notin \widehat{\mathcal{Y}}_N$, our design of $r^\star$ and $\widehat{r}$ ensures that BoN will always choose the response $y \in \widehat{\mathcal{Y}}_N$ with the smallest reward $r^\star$, while $\pi_{\text{ref}}$ is equivalent to choosing $y \in \widehat{\mathcal{Y}}_N$ uniformly, thus

$$
\mathbb{E}\left[ J(\pi_{\text{ref}}) - J(\widehat{\pi}_{\text{BoN}}) \mid k_N \notin \widehat{\mathcal{Y}}_N \right] \geq 0.
$$

Formally, combining these cases gives

$$
\begin{aligned}
J(\pi^\star) - J(\widehat{\pi}_{\text{BoN}}) &= \mathbb{P}(k_N \in \widehat{\mathcal{Y}}_N) \cdot \mathbb{E}\left[ J(\pi^\star) - J(\widehat{\pi}_{\text{BoN}}) \mid k_N \in \widehat{\mathcal{Y}}_N \right] + \mathbb{P}\left( k_N \notin \widehat{\mathcal{Y}}_N \right) \cdot \mathbb{E}\left[ J(\pi^\star) - J(\widehat{\pi}_{\text{BoN}}) \mid k_N \notin \widehat{\mathcal{Y}}_N \right] \\
&\geq \mathbb{P}(k_N \in \widehat{\mathcal{Y}}_N) \cdot \frac{\sqrt{k_N} \cdot \Delta_N(k_N)}{2} + \mathbb{P}\left( k_N \notin \widehat{\mathcal{Y}}_N \right) \cdot (J(\pi^\star) - J(\pi_{\text{ref}})) \\
&> \mathbb{P}(k_N \in \widehat{\mathcal{Y}}_N) \cdot \frac{\sqrt{k_N} \cdot \Delta_N(k_N)}{2} \\
&\geq \left( 1 - e^{-3} \right) \cdot \sqrt{\frac{1}{32} \cdot N \cdot \varepsilon_{\text{RM}}^2}
\end{aligned}
$$

where in the last inequality we use the fact that, since $\pi_{\text{ref}}(k_N) \geq \frac{3}{N}$,

$$
\mathbb{P}(k \in \widehat{\mathcal{Y}}_N) = 1 - \mathbb{P}(k \notin \widehat{\mathcal{Y}}_N) = 1 - \left( 1 - \frac{3}{N} \right)^N > 1 - e^{-3}.
$$

$\square$

### F.2.3. SUPPORTING LEMMAS

The following lemma identifies a pair of distributions $\pi$ and $\pi_{\mathsf{ref}}$ where the upper bound $\mathcal{M}_\varepsilon^\pi \leq \frac{\mathcal{C}^\pi}{\varepsilon}$ is tight up to logarithmic factors, and where it is information-theoretically hard to approximate $\pi$ using samples from $\pi_{\mathsf{ref}}$. In particular, the distributions exhibit $\mathcal{M}_\varepsilon^\pi = O(\varepsilon^{-1})$ and $\mathcal{C}^\pi = O(\log(\mathcal{C}_\infty^\pi))$.

**Lemma F.5.** *For any $\varepsilon \in (0, \frac{1}{4}]$ and $C \geq \frac{1}{2 \cdot \varepsilon}$, there exists a prompt space $\mathcal{X} = \{x\}$, response space $\mathcal{Y}$, and two distributions $\pi, \pi_{\mathsf{ref}} : \mathcal{X} \to \Delta(\mathcal{Y})$ with*

- $\mathcal{C}^\pi(x) = O(\log(C))$*; and*
- $\mathcal{C}_\infty^\pi(x) = O(C)$*;*

*such that if $N < \frac{1}{12 \cdot \varepsilon}$, any selection algorithm $\mathcal{A}$ (Definition F.2) has $D_{\mathsf{TV}}(\pi(x), \pi_{\mathcal{A}}(x)) > \varepsilon$.*

**Proof of Lemma F.5.** We omit all $x$ dependencies given that there is a single prompt, and all instances of log are base 2. The construction is as follows. The response space $\mathcal{Y} = \mathbb{N}$ is countably infinite and indexed by $i \in \{1, 2, 3, \ldots\}$. Let $I := \lceil \log(C) \rceil$ (the preconditions on $\varepsilon$ and $C$ are to ensure that $I \geq 1$), and define

$$\pi(i) = \begin{cases} \frac{2^{-i}}{Z_\pi} & \text{if } i \leq I, \\ \frac{2^I}{Z_\pi} \cdot \pi_{\mathsf{ref}}(i) & \text{if } i > I, \end{cases} \quad \text{and} \quad \pi_{\mathsf{ref}}(i) = \frac{4^{-i}}{Z_{\pi_{\mathsf{ref}}}},$$

where

$$Z_{\pi_{\mathsf{ref}}} := \sum_{i=1}^\infty 4^{-i} = \frac{1}{3}, \quad \text{and}$$

$$Z_\pi = \sum_{i=1}^I 2^{-i} + 2^I \cdot \sum_{i=I+1}^\infty \frac{4^{-i}}{Z_{\pi_{\mathsf{ref}}}} = (1 - 2^{-I}) + 2^I \cdot 4^{-I} = 1.$$

The coverage coefficient has $\mathcal{C}^\pi = O(\log C)$ since

$$\mathcal{C}^\pi = \sum_{i=1}^I \frac{\pi^2(i)}{\pi_{\mathsf{ref}}(i)} + \frac{4^I}{(Z_\pi)^2} \cdot \sum_{i=I+1}^\infty \pi_{\mathsf{ref}}(i) = \frac{Z_{\pi_{\mathsf{ref}}}}{(Z_\pi)^2} \cdot I + \frac{4^I \cdot Z_{\pi_{\mathsf{ref}}}}{(Z_\pi)^2} \cdot 4^{-I} = \frac{Z_{\pi_{\mathsf{ref}}}}{(Z_\pi)^2}(I + 1),$$

while $\mathcal{C}_\infty^\pi = O(C)$, since $\mathcal{C}_\infty^\pi = \frac{Z_{\pi_{\mathsf{ref}}}}{Z_\pi} \cdot 2^{\lceil \log C \rceil}$. Next, recall that for any $M \geq 1$,

$$\mathcal{E}_M(\pi, \pi_{\mathsf{ref}}) = \sum_{i=1}^\infty \pi_{\mathsf{ref}}(i) \cdot \left( \frac{\pi(i)}{\pi_{\mathsf{ref}}(i)} - M \right)_+ = \sum_{i=1}^\infty \frac{4^{-i}}{Z_{\pi_{\mathsf{ref}}}} \cdot \left( \frac{2^{(i \wedge I)} \cdot Z_{\pi_{\mathsf{ref}}}}{Z_\pi} - M \right)_+.$$

Note that $\frac{\pi(i)}{\pi_{\mathsf{ref}}(i)}$ increases with $i$. Motivated by this, we will set $M = M_k := \frac{2^k \cdot Z_{\pi_{\mathsf{ref}}}}{Z_\pi}$ for some index $k \leq I$ to be specified shortly, which effectively zeroes out all terms $i \leq k$ in the sum above:

$$\mathcal{E}_{M_k}(\pi, \pi_{\mathsf{ref}}) = \sum_{i=1}^\infty \frac{4^{-i}}{Z_{\pi_{\mathsf{ref}}}} \cdot \left( \frac{2^{(i \wedge I)} \cdot Z_{\pi_{\mathsf{ref}}}}{Z_\pi} - M_k \right)_+.$$

$$= \sum_{i=k+1}^\infty \frac{4^{-i}}{Z_{\pi_{\mathsf{ref}}}} \cdot \left( \frac{2^i \cdot Z_{\pi_{\mathsf{ref}}}}{Z_\pi} - M_k \right)$$

$$= \sum_{i=k+1}^\infty \frac{2^{-i}}{Z_\pi} - M_k \cdot \sum_{i=k}^\infty \frac{4^{-i}}{Z_{\pi_{\mathsf{ref}}}}$$

$$= 2^{-k} - M_k \cdot 4^{-k}$$

$$= 2^{-k} - \left( \frac{2^k \cdot Z_{\pi_{\mathsf{ref}}}}{Z_\pi} \right) \cdot 4^{-k}$$

$$= 2^{-k} \cdot \left( 1 - \frac{Z_{\pi_{\mathsf{ref}}}}{Z_\pi} \right).$$

Now let $Z = \frac{Z_{\pi_{\text{ref}}}}{Z_\pi} = \frac{1}{3}$ for short, and set $k = \lfloor \log(\frac{1}{2\varepsilon}) \rfloor$. We check that $k \leq \log(\frac{1}{2\varepsilon}) \leq \log(C) \leq I$, since $C \geq \frac{1}{2\varepsilon}$ by assumption. For this choice of $k$, we have

$$\mathcal{E}_{M_k}(\pi, \pi_{\text{ref}}) \geq 2^{-\log(\frac{1}{2\varepsilon})} \cdot (1 - Z) = \frac{4}{3} \cdot \varepsilon.$$

Recall that $\mathcal{M}_\varepsilon^\pi = \min\{M \mid \mathcal{E}_M(\pi, \pi_{\text{ref}}) \leq \varepsilon\}$. Since larger $M$ has smaller $\mathcal{E}_M(\pi, \pi_{\text{ref}})$, we conclude that

$$\mathcal{M}_\varepsilon^\pi \geq M_k \geq 2^{\log(\frac{1}{4\varepsilon})} \cdot Z = \frac{Z}{4\varepsilon} = \frac{1}{12 \cdot \varepsilon}.$$

The theorem statement then follows by applying Lemma D.6, which states that when $N < \frac{1}{12 \cdot \varepsilon}$, any selection strategy $\mathcal{A}$ must have $D_{\text{TV}}(\pi, \pi_{\mathcal{A}}) > \varepsilon$.

$\square$

### F.3. Proof of Theorem 3.1

**Proof of Theorem 3.1.** Recall from Definition D.1 that for a given prompt $x$ and target policy $\pi$,

$$\mathcal{E}_M(\pi(x), \pi_{\text{ref}}(x)) := \sum_{y \in \mathcal{Y}_M(x)} \pi(y \mid x) - M \cdot \pi_{\text{ref}}(y \mid x) = \mathbb{E}_{y \sim \pi_{\text{ref}}(\cdot \mid x)}\left[\left(\frac{\pi(y \mid x)}{\pi_{\text{ref}}(y \mid x)} - M\right)_+\right].$$

Proposition D.3 states that for any $M$ we can upper bound $\mathcal{E}_M(\pi^\star(x), \pi_{\text{ref}}(x)) \leq \frac{\mathcal{C}^{\pi^\star}(x)}{M}$, which when combined with Lemma F.1 results in

$$J(\pi^\star; x) - J(\widehat{\pi}_{\text{BoN}}; x)$$
$$\leq R_{\max} \cdot \left(\frac{\mathcal{C}^{\pi^\star}(x)}{M} + \exp\left(-\frac{N}{M} \cdot (1 - \mathcal{E}_M(\pi^\star(x), \pi_{\text{ref}}(x)))\right)\right) + 2 \cdot \sqrt{\mathcal{C}^{\pi^\star}(x) \cdot \varepsilon_{\text{RM}}^2(x)} + \frac{\varepsilon_{\text{RM}}(x)}{2} + \sqrt{N \cdot \varepsilon_{\text{RM}}^2(x)}$$

as long as $M$ is large enough such that $\mathcal{E}_M(\pi^\star(x), \pi_{\text{ref}}(x)) \leq \frac{\mathcal{C}^{\pi^\star}(x)}{M} \leq 1/2$. We will set $M = \frac{N}{\log(4R_{\max}^2/\varepsilon_{\text{RM}}^2(x))}$; we can check that as long as $N \geq 2 \cdot \mathcal{C}^{\pi^\star}(x) \cdot \log(4R_{\max}^2/\varepsilon_{\text{RM}}^2(x))$ then $\mathcal{E}_M(\pi^\star(x), \pi_{\text{ref}}(x)) \leq \frac{1}{2}$ since

$$\mathcal{E}_M(\pi^\star(x), \pi_{\text{ref}}(x)) \leq \frac{\mathcal{C}^{\pi^\star}(x)}{M} = \frac{\mathcal{C}^{\pi^\star}(x) \log(4R_{\max}^2/\varepsilon_{\text{RM}}^2(x))}{N} \leq \frac{1}{2}.$$

Then for any $N \geq 2 \cdot \mathcal{C}^{\pi^\star}(x) \cdot \log(4R_{\max}^2/\varepsilon_{\text{RM}}^2(x))$, we have

$$J(\pi^\star; x) - J(\widehat{\pi}_{\text{BoN}}; x) \leq R_{\max} \cdot \left(\frac{\mathcal{C}^{\pi^\star}(x)}{M} + \exp\left(-\frac{N}{2M}\right)\right) + 2 \cdot \sqrt{\mathcal{C}^{\pi^\star}(x) \cdot \varepsilon_{\text{RM}}^2(x)} + \frac{\varepsilon_{\text{RM}}(x)}{2} + \sqrt{N \cdot \varepsilon_{\text{RM}}^2(x)}$$

$$= R_{\max} \cdot \frac{\mathcal{C}^{\pi^\star}(x) \log\left(\frac{4R_{\max}^2}{\varepsilon_{\text{RM}}^2(x)}\right)}{N} + 2 \cdot \sqrt{\mathcal{C}^{\pi^\star}(x) \cdot \varepsilon_{\text{RM}}^2(x)} + \varepsilon_{\text{RM}}(x) + \sqrt{N \cdot \varepsilon_{\text{RM}}^2(x)},$$

which completes the first statement of the theorem. The second part of the theorem statement follows from by setting $N$ to minimize the RHS of the regret bound above, by balancing the two $N$-dependent terms. Namely, if

$$N \asymp \left(\frac{\mathcal{C}^{\pi^\star}(x) \cdot R_{\max} \cdot \log\left(\frac{4R_{\max}^2}{\varepsilon_{\text{RM}}^2(x)}\right)}{\varepsilon_{\text{RM}}(x)}\right)^{\frac{2}{3}}, \text{ then}$$

$$J(\pi^\star; x) - J(\widehat{\pi}_{\text{BoN}}; x) \lesssim \left(R_{\max} \cdot \mathcal{C}^{\pi^\star}(x) \cdot \varepsilon_{\text{RM}}^2(x) \cdot \log\left(\frac{R_{\max}}{\varepsilon_{\text{RM}}(x)}\right)\right)^{\frac{1}{3}}.$$

$\square$

### F.4. Proof of Theorem 3.2

**Proof of Theorem 3.2.** The first part of the theorem statement follows from Theorem F.3 with $C = 2$, which then states that

$$J(\pi^\star; x) - J(\widehat{\pi}_{\mathsf{BoN}}) \geq c \cdot \min\left\{ \sqrt{N \cdot \varepsilon_{\mathsf{RM}}^2(x)},\ 1 \right\}$$

as long as $N \geq 1$, where $c$ is a universal constant.

The second part of the theorem statement follows from invoking Theorem F.4 with $p = \frac{1}{3}$ and $C = \varepsilon_{\mathsf{RM}}(x)^{-1}$ which states that if $N = \widetilde{O}\left(\varepsilon_{\mathsf{RM}}^{-\frac{2}{3}}(x)\right)$ then

$$J(\pi^\star; x) - J(\widehat{\pi}_{\mathsf{BoN}}; x) > c_1 \cdot \left(\varepsilon_{\mathsf{RM}}^2(x) \cdot \log(\varepsilon_{\mathsf{RM}}(x))\right)^{\frac{1}{3}}$$

while if $N = \widetilde{\Omega}\left(\varepsilon_{\mathsf{RM}}^{-\frac{2}{3}}(x)\right)$ then

$$J(\pi^\star; x) - J(\widehat{\pi}_{\mathsf{BoN}}; x) > c_2 \cdot \sqrt{N \cdot \varepsilon_{\mathsf{RM}}^2(x)} \geq \widetilde{\Omega}\left(\varepsilon_{\mathsf{RM}}^{\frac{2}{3}}(x)\right).$$

$\square$

### F.5. Proof of Theorem 3.4

**Proof of Theorem 3.4.** When $M = \mathcal{C}_\infty^{\pi^\star}(x)$ we have $\mathcal{E}_M(\pi^\star(x), \pi_{\mathsf{ref}}(x)) = 0$, and using this choice of $M$ in Lemma F.1 gives

$$J(\pi^\star; x) - J(\widehat{\pi}_{\mathsf{BoN}}; x) \leq R_{\max} \cdot \exp\left(-\frac{N}{\mathcal{C}_\infty^{\pi^\star}(x)}\right) + 2 \cdot \sqrt{\mathcal{C}^{\pi^\star}(x) \cdot \varepsilon_{\mathsf{RM}}^2(x)} + \frac{\varepsilon_{\mathsf{RM}}(x)}{2} + \sqrt{N \cdot \varepsilon_{\mathsf{RM}}^2(x)},$$

which proves the first part of the theorem statement. For the second, we observe for any $N \geq \mathcal{C}_\infty^{\pi^\star}(x) \log(2R_{\max}/\varepsilon_{\mathsf{RM}}(x))$, we have

$$J(\pi^\star; x) - J(\widehat{\pi}_{\mathsf{BoN}}; x) \lesssim \varepsilon_{\mathsf{RM}}(x) + \sqrt{N \cdot \varepsilon_{\mathsf{RM}}^2(x)},$$

and if $N \asymp \mathcal{C}_\infty^{\pi^\star}(x) \cdot \log(R_{\max}/\varepsilon_{\mathsf{RM}}(x))$ additionally, then

$$J(\pi^\star; x) - J(\widehat{\pi}_{\mathsf{BoN}}; x) \lesssim \sqrt{\mathcal{C}_\infty^{\pi^\star}(x) \cdot \varepsilon_{\mathsf{RM}}^2(x) \cdot \log(R_{\max}/\varepsilon_{\mathsf{RM}}(x))}.$$

$\square$

### F.6. Additional Results

Here, as an additional result, we prove a regret guarantee for `BoN-Alignment` in terms of the information-theoretic quantities $\mathcal{M}_{x,\varepsilon}^\pi$ and $\mathcal{M}_\varepsilon^\pi$ (Definition D.2). Our main theorems, Theorem 3.1 and Theorem 3.4, can be viewed as more interpretable relaxations that isolate the dependencies on coverage coefficients and $N$.

**Theorem F.6** (BoN regret with $\mathcal{M}_\varepsilon^{\pi^\star}$)**.** *Given prompt $x$, for any $\varepsilon \in [0, \frac{1}{2}]$ and $N \in \mathbb{Z}$ and comparator policy $\pi^\star$, the BoN policy $\widehat{\pi}_{\mathsf{BoN}}$ satisfies*

$$J(\pi^\star; x) - J(\widehat{\pi}_{\mathsf{BoN}}; x) \leq R_{\max} \cdot \left(\varepsilon + \exp\left(-\frac{N}{2\mathcal{M}_{x,\varepsilon}^{\pi^\star}}\right)\right) + 2 \cdot \sqrt{\mathcal{C}^{\pi^\star}(x) \cdot \varepsilon_{\mathsf{RM}}^2(x)} + \frac{\varepsilon_{\mathsf{RM}}(x)}{2} + \sqrt{N \cdot \varepsilon_{\mathsf{RM}}^2(x)}.$$

*In particular, if $N = 2\mathcal{M}_{x,\varepsilon}^{\pi^\star} \cdot \log\left(\frac{\varepsilon_{\mathsf{RM}}(x)}{2R_{\max}}\right)$,*

$$J(\pi^\star; x) - J(\widehat{\pi}_{\mathsf{BoN}}; x) \lesssim R_{\max} \cdot \varepsilon + \sqrt{\mathcal{C}^{\pi^\star}(x) \cdot \varepsilon_{\mathsf{RM}}^2(x)} + \sqrt{\mathcal{M}_{x,\varepsilon}^{\pi^\star} \cdot \varepsilon_{\mathsf{RM}}^2(x) \cdot \log\left(\frac{\varepsilon_{\mathsf{RM}}(x)}{2R_{\max}}\right)}.$$

**Proof of Theorem F.6.** We will apply Lemma F.1 with $M = \mathcal{M}_{x,\varepsilon}^{\pi^\star}$ for $\varepsilon \in [0, \frac{1}{2}]$. Whenever $\varepsilon \leq \frac{1}{2}$, we have $\mathcal{E}_M(\pi^\star(x), \pi_{\mathsf{ref}}(x)) \leq \frac{1}{2}$ by definition, since

$$\mathcal{E}_{\mathcal{M}_{x,\varepsilon}^{\pi^\star}}(\pi^\star(x), \pi_{\mathsf{ref}}(x)) \leq \varepsilon \leq \frac{1}{2}.$$

As a result, $1 - \mathcal{E}_{\mathcal{M}_{x,\varepsilon}^{\pi^\star}}(\pi^\star(x), \pi_{\mathsf{ref}}(x)) \geq \frac{1}{2}$, and Lemma F.1 states that, for any $N$,

$$J(\pi^\star; x) - J(\widehat{\pi}_{\mathsf{BoN}}; x) \leq R_{\mathsf{max}} \cdot \left(\varepsilon + \exp\left(-\frac{N}{2\mathcal{M}_{x,\varepsilon}^{\pi^\star}}\right)\right) + 2 \cdot \sqrt{\mathcal{C}^{\pi^\star}(x) \cdot \varepsilon_{\mathsf{RM}}^2(x)} + \frac{\varepsilon_{\mathsf{RM}}(x)}{2} + \sqrt{N \cdot \varepsilon_{\mathsf{RM}}^2(x)}.$$

Setting $N = 2\mathcal{M}_{x,\varepsilon}^{\pi^\star} \log\left(\frac{\varepsilon_{\mathsf{RM}}(x)}{2R_{\mathsf{max}}}\right)$, we obtain

$$J(\pi^\star; x) - J(\widehat{\pi}_{\mathsf{BoN}}; x) \leq R_{\mathsf{max}} \cdot \varepsilon + 2 \cdot \sqrt{\mathcal{C}^{\pi^\star}(x) \cdot \varepsilon_{\mathsf{RM}}^2(x)} + \varepsilon_{\mathsf{RM}}(x) + \sqrt{2\mathcal{M}_{x,\varepsilon}^{\pi^\star} \cdot \varepsilon_{\mathsf{RM}}^2(x) \cdot \log\left(\frac{\varepsilon_{\mathsf{RM}}(x)}{2R_{\mathsf{max}}}\right)}$$

$$\lesssim R_{\mathsf{max}} \cdot \varepsilon + \sqrt{\mathcal{C}^{\pi^\star}(x) \cdot \varepsilon_{\mathsf{RM}}^2(x)} + \sqrt{\mathcal{M}_{x,\varepsilon}^{\pi^\star} \cdot \varepsilon_{\mathsf{RM}}^2(x) \cdot \log\left(\frac{\varepsilon_{\mathsf{RM}}(x)}{2R_{\mathsf{max}}}\right)}.$$

$\square$

# G. Proofs from Section 4

The proofs below are conditioned on a single prompt $x$, and as a result we omit $x$ dependencies to simplify presentation throughout. We refer to $x$-dependent quantities via their $x$-independent analogs, e.g., $J(\pi; x) \equiv J(\pi)$, $\lambda(x) \equiv \lambda$, etc.

## G.1. Proof of Theorem 4.1

**Proof of Theorem 4.1.** In the proof below, for a reward model $r$ we define the expected return under it to be $J_r(\pi; x) := \mathbb{E}_{y \sim \pi(x)}[r(x, y)]$, and using this definition we can also write $J(\pi; x) = J_{r^\star}(\pi; x)$. With prompt dependencies ommitted, we equivalently write $J_r(\pi; x) \equiv J_r(\pi)$ .

Given a reward function $\widehat{r}$, define for a normalization constant $\lambda$ the following functions:

$$\pi_\lambda(y) = \frac{\pi_{\mathsf{ref}}(y)\mathsf{relu}\big(\beta^{-1}(\widehat{r}(y) - \lambda)\big)}{\Phi(\lambda)}, \tag{19}$$

$$\Phi(\lambda) = \mathbb{E}_{\pi_{\mathsf{ref}}}\big[\mathsf{relu}\big(\beta^{-1}(\widehat{r}(y) - \lambda)\big)\big], \tag{20}$$

and

$$\widehat{\Phi}(\lambda) = \frac{1}{N}\sum_{i=1}^{N}\mathsf{relu}\big(\beta^{-1}(\widehat{r}(y_i) - \lambda)\big).$$

We analyze the regret using the following decomposition, where $\widehat{\lambda} \equiv \widehat{\lambda}(x)$ is the output of Eq. (9) in Algorithm 2.

$$J(\pi^\star) - J(\widehat{\pi}) = \underbrace{J(\pi^\star) - J_{\widehat{r}}(\pi^\star)}_{(\mathrm{T1})} + \underbrace{J_{\widehat{r}}(\pi^\star) - J_{\widehat{r}}(\pi_{\widehat{\lambda}})}_{(\mathrm{T2})} + \underbrace{J_{\widehat{r}}(\pi_{\widehat{\lambda}}) - J(\pi_{\widehat{\lambda}})}_{(\mathrm{T3})} + \underbrace{J(\pi_{\widehat{\lambda}}) - J(\widehat{\pi})}_{(\mathrm{T4})}.$$

For (T2), recall that $\widehat{\lambda}$ is computed using using Algorithm 3 in Eq. (9), which Lemma C.1 shows obtains $\widehat{\lambda}$ such that $\widehat{\Phi}(\lambda) = 1$. Further, the procedure in Algorithm 3 is equivalent to solving the optimization problem in Lemma G.2 with $\pi_{\mathsf{ref}} = \mathsf{unif}\big(\widehat{\mathcal{Y}}_N\big)$ and $\alpha = 1$, and, as a result, we have $\widehat{\lambda} \in [-\beta, R_{\mathsf{max}} - \beta]$. Now for a fixed $N$, the bound in Lemma G.3 states that with probability at least $1 - \delta$, for any $\delta \in (0, 1)$ we have

$$\frac{7}{9} + \frac{8}{9} \cdot \varepsilon_N \leq \Phi(\widehat{\lambda}) \leq \frac{9}{7} + \frac{8}{7} \cdot \varepsilon_N,$$

where $\varepsilon_N := 12\left(\frac{R_{\mathsf{max}} + \beta}{\beta}\right)\frac{\log\left(\frac{60R_{\mathsf{max}}}{\beta\delta}\right)}{N}$ is the expression in the RHS of the bound. Then to have $\Phi(\widehat{\lambda}) \in \left[\frac{1}{2}, \frac{3}{2}\right]$ with probability $\geq 1 - \delta$, it is sufficient to choose $N$ large enough such that $\varepsilon_N \leq \frac{1}{4}$, which is satisfied when

$$N \geq 48\left(\frac{R_{\mathsf{max}} + \beta}{\beta}\right)\log\left(\frac{60R_{\mathsf{max}}}{\beta\delta}\right).$$

Next, let $\mathcal{E} = \left\{\Phi(\widehat{\lambda}) \in \left[\frac{1}{2}, \frac{3}{2}\right]\right\}$ denote the aforementioned high-probability event. Using Lemma G.1, we may bound

$$\begin{aligned}
J_{\widehat{r}}(\pi^\star) - J_{\widehat{r}}(\pi_{\widehat{\lambda}}) &\leq \mathbb{P}(\mathcal{E}) \cdot \mathbb{E}\big[J_{\widehat{r}}(\pi^\star) - J_{\widehat{r}}(\pi_{\widehat{\lambda}}) \mid \mathcal{E}\big] + \mathbb{P}(\neg\mathcal{E}) \cdot \mathbb{E}\big[J_{\widehat{r}}(\pi^\star) - J_{\widehat{r}}(\pi_{\widehat{\lambda}}) \mid \neg\mathcal{E}\big] \\
&\leq \mathbb{E}\big[J_{\widehat{r}}(\pi^\star) - J_{\widehat{r}}(\pi_{\widehat{\lambda}}) \mid \mathcal{E}\big] + \delta \cdot R_{\mathsf{max}} \\
&\leq \frac{3\beta}{4} \cdot \mathcal{C}^{\pi^\star} - \frac{\beta}{4} \cdot \mathcal{C}^{\pi_{\widehat{\lambda}}} + \delta \cdot R_{\mathsf{max}},
\end{aligned}$$

By setting $\delta = \frac{\varepsilon_{\mathsf{RM}}}{R_{\mathsf{max}}}$, we obtain that

$$(\mathrm{T2}) = J_{\widehat{r}}(\pi^\star) - J_{\widehat{r}}(\pi_{\widehat{\lambda}}) \leq \frac{3\beta}{4} \cdot \mathcal{C}^{\pi^\star} - \frac{\beta}{4} \cdot \mathcal{C}^{\pi_{\widehat{\lambda}}} + \varepsilon_{\mathsf{RM}}$$

if $N = \Omega\left(\left(1 + \frac{R_{\mathsf{max}}}{\beta}\right)\log\left(\frac{R_{\mathsf{max}}}{\beta \cdot \varepsilon_{\mathsf{RM}}}\right)\right)$.

For (T4), from Lemma D.4 with $M = \frac{R_{\max}}{\beta}$, we have

$$D_{\mathsf{TV}}\left(\pi_{\widehat{\lambda}}, \widehat{\pi}\right) \leq \exp\left(-\frac{N \cdot \beta}{2R_{\max}}\right)$$

thus as long as $N \gtrsim \frac{R_{\max}}{\beta} \log\left(\frac{R_{\max}}{\varepsilon_{\mathsf{RM}}}\right)$ we have

$$D_{\mathsf{TV}}\left(\pi_{\widehat{\lambda}}, \widehat{\pi}\right) \leq \frac{\varepsilon_{\mathsf{RM}}}{R_{\max}}.$$

As a result,

$$(\mathrm{T4}) = J(\pi_{\widehat{\lambda}}) - J(\widehat{\pi}) \leq R_{\max} \cdot D_{\mathsf{TV}}\left(\pi_{\widehat{\lambda}}, \widehat{\pi}\right) \leq \varepsilon_{\mathsf{RM}}.$$

For (T1), we know from the standard Cauchy-Schwarz bound that

$$(\mathrm{T1}) = J(\pi^\star) - J_{\widehat{r}}(\pi^\star) = \mathbb{E}_{y \sim \pi^\star}[r^\star(y) - \widehat{r}(y)] \leq \sqrt{\mathcal{C}^{\pi^\star} \cdot \varepsilon_{\mathsf{RM}}^2}$$

Lastly, for (T3), Cauchy-Schwarz and the AM-GM inequality imply that

$$(\mathrm{T3}) = J_{\widehat{r}}(\pi_{\widehat{\lambda}}) - J(\pi_{\widehat{\lambda}}) = \mathbb{E}_{y \sim \pi_{\widehat{\lambda}}}[\widehat{r}(y) - r^\star(y)] \leq \sqrt{\mathcal{C}^{\pi_{\widehat{\lambda}}} \cdot \varepsilon_{\mathsf{RM}}^2} \leq \frac{\beta}{4} \cdot \mathcal{C}^{\pi_{\widehat{\lambda}}} + \frac{\varepsilon_{\mathsf{RM}}^2}{\beta}$$

Putting things together, as long as $N \gtrsim \widetilde{\Omega}\left(\left(1 + \frac{R_{\max}}{\beta}\right) \log\left(\frac{R_{\max}}{\beta \cdot \varepsilon_{\mathsf{RM}}}\right)\right)$ we have

$$J(\pi^\star) - J(\widehat{\pi}) \leq \sqrt{\mathcal{C}^{\pi^\star} \cdot \varepsilon_{\mathsf{RM}}^2} + \frac{3\beta}{4} \cdot \mathcal{C}^{\pi^\star} + \frac{\varepsilon_{\mathsf{RM}}^2}{\beta} + 2\varepsilon_{\mathsf{RM}}.$$

We set $\beta = \sqrt{\frac{\varepsilon_{\mathsf{RM}}^2}{\mathcal{C}^{\pi^\star}}}$ to balance the terms for the final result. $\qquad\square$

### G.1.1. SUPPORTING LEMMAS
Throughout this section, we consider a fix prompt $x \in \mathcal{X}$ and omit dependence on $x$ as above.

**Lemma G.1.** *Suppose we have $\lambda$ such that $\Phi(\lambda) = \alpha$ for some $\alpha > 0$ (cf. Eq. (20)). Then for $\pi_\lambda$ defined in Eq. (19), it holds for any policy $\pi$ that*

$$J_{\widehat{r}}(\pi) - J_{\widehat{r}}(\pi_\lambda) \leq \frac{\alpha\beta}{2} \cdot \mathcal{C}^\pi - \frac{\alpha\beta}{2} \cdot \mathcal{C}^{\pi_\lambda}.$$

*In particular, if $\alpha \in \left[\frac{1}{2}, \frac{3}{2}\right]$, then*

$$J_{\widehat{r}}(\pi) - J_{\widehat{r}}(\pi_\lambda) \leq \frac{3\beta}{4} \cdot \mathcal{C}^\pi - \frac{\beta}{4} \cdot \mathcal{C}^{\pi_\lambda}$$

**Proof of Lemma G.1.** Let $\alpha = \Phi(\lambda)$ and define $\pi_\lambda' = \mathrm{relu}\left(\beta^{-1}(\widehat{r}(y) - \lambda)\right) = \alpha \cdot \pi_\lambda$, so that $\pi_\lambda' \in \Delta_\alpha(\mathcal{Y})$ (cf. Eq. (21)). Further, for the comparator policy $\pi$, define $\pi' = \alpha \cdot \pi \in \Delta_\alpha(\mathcal{Y})$. We know from Lemma G.2 that

$$J_{\widehat{r}}(\pi') - J_{\widehat{r}}(\pi_\lambda') \leq \frac{\beta}{2} \cdot \mathcal{C}^{\pi'} - \frac{\beta}{2}\mathcal{C}^\pi$$

It can be observed that $\mathcal{C}^{\pi'} = \alpha^2 \cdot \mathcal{C}^\pi$ and $\mathcal{C}^{\pi_\lambda'} = \alpha^2 \cdot \mathcal{C}^{\pi_\lambda}$, as well as $\alpha(J_{\widehat{r}}(\pi) - J_{\widehat{r}}(\pi_\lambda)) = J_{\widehat{r}}(\pi') - J_{\widehat{r}}(\pi_\lambda')$, therefore

$$J_{\widehat{r}}(\pi) - J(\pi_\lambda) \leq \frac{\alpha\beta}{2} \cdot \mathcal{C}^\pi - \frac{\alpha\beta}{2}\mathcal{C}^{\pi_\lambda}.$$

For the second statement, for any $\alpha \in \left[\frac{1}{2}, \frac{3}{2}\right]$ we have

$$J_{\widehat{r}}(\pi) - J(\pi_\lambda) \leq \frac{\frac{3}{2} \cdot \beta}{2} \cdot \mathcal{C}^\pi - \frac{\frac{1}{2} \cdot \beta}{2}\mathcal{C}^{\pi_\lambda} = \frac{3\beta}{4} \cdot \mathcal{C}^{\pi^\star} - \frac{\beta}{4} \cdot \mathcal{C}^\pi$$

$\qquad\square$

**Lemma G.2.** *Let a reward function $r$ and parameter $\alpha > 0$ be given, and define*

$$\Delta_\alpha(\mathcal{Y}) = \left\{ \pi \in \mathbb{R}_+^{\mathcal{Y}} \mid \sum_{y \in \mathcal{Y}} \pi(y) = \alpha \right\}. \tag{21}$$

*Then there exists a choice of $\lambda \in [J(\pi_{\text{ref}}) - \alpha\beta, \max_{y \in \mathcal{Y}} r(y) - \alpha\beta]$ such that $\sum_{y \in \mathcal{Y}} \pi(y) = \alpha$. Furthermore, given any $\lambda$ such that $\sum_{y \in \mathcal{Y}} \pi(y) = \alpha$,*

$$\pi(y) = \pi_{\text{ref}}(y) \cdot \text{relu}\left(\beta^{-1}(r(y) - \lambda)\right)$$

*is the optimal solution to*

$$\pi = \arg\max_{\pi \in \Delta_\alpha(\mathcal{Y})} \left[ J(\pi) - \frac{\beta}{2} \mathcal{C}^\pi \right]. \tag{22}$$

**Proof of Lemma G.2.** First we rewrite the objective in Eq. (22) in primal form,

$$\pi = \arg\min_{\pi \in \mathbb{R}^{\mathcal{Y}}} \left\{ -J(\pi) + \frac{\beta}{2} \mathcal{C}^\pi \right\}.$$
$$\text{s.t.} \sum_{y \in \mathcal{Y}} \pi(y) = \alpha$$
$$- \pi(y) \leq 0, \ \forall y \in \mathcal{Y}$$

and we can verify that Slater's condition holds, because the objective is convex, since $J(\pi)$ is affine and $\mathcal{C}^\pi$ is (strongly) convex, and there exists at least one strictly feasible point, an example being the function $\pi'$ that sets $\pi'(y) = \frac{\alpha}{|\mathcal{Y}|}$ for all $y \in \mathcal{Y}$.

Under strong duality, the KKT conditions are both necessary and sufficient for optimality; further, the objective has a unique minimum due to strong convexity, and therefore, to prove the theorem statement, it is sufficient to show that the proposed $\pi$ satisfies the KKT conditions.

For primal variable $\pi$ and dual variables $(\nu, \lambda)$, the Lagrangian is given by

$$\min_{\pi \in \mathbb{R}^{\mathcal{Y}}} \max_{\lambda \in \mathbb{R}, \nu \in \mathbb{R}_+^{\mathcal{Y}}} \mathcal{L}(\pi, \nu, \lambda) = -\sum_{y \in \mathcal{Y}} \pi(y) r(y) + \frac{\beta}{2} \mathcal{C}^\pi + \lambda \left( \sum_{y \in \mathcal{Y}} \pi(y) - \alpha \right) - \sum_{y \in \mathcal{Y}} \nu(y) \pi(y)$$

and under strong duality, we know that the optimal primal and dual variables $(\pi, \nu, \lambda)$ satisfy

- $\pi \geq 0$
- $\sum_y \pi(y) = \alpha$
- $\nu \geq 0$
- $\pi(y) \cdot \nu(y) = 0$ for all $y$ (complementary slackness).
- $\nabla_\pi \mathcal{L}(\pi, (\nu, \lambda)) = 0$ (first-order condition).

From the first-order condition $\nabla_\pi \mathcal{L}(\pi, \nu, \lambda) = 0$, we know that $\pi$ satisfies for all $y \in \mathcal{Y}$:

$$r(y) - \beta \frac{\pi(y)}{\pi_{\text{ref}}(y)} - \lambda + \nu(y) = 0,$$

or after rearranging,

$$\pi(y) = \pi_{\text{ref}}(y) \cdot \beta^{-1}(r(y) - \lambda + \nu(y)). \tag{23}$$

Now consider a fixed $y \in \mathcal{Y}$. We consider three cases:

- If $r(y) - \lambda < 0$, then we must have $\nu(y) \geq -(r(y) - \lambda) > 0$ to satisfy $\pi(y) \geq 0$ by Eq. (23). By complementary slackness, this implies that $\pi(y) = 0$.

- If $r(y) - \lambda = 0$, then Eq. (23) gives $\pi(y) = \pi_{\text{ref}}(y)\beta^{-1}\nu(y)$, which implies $\pi(y) = 0$ by complementary slackness.

- if $r(y) - \lambda > 0$, then $r(y) - \lambda + \nu(y) > 0$ (since $\nu(y) \geq 0$), which in turn gives $\pi(y) > 0$ by Eq. (23). This implies that $\nu(y) = 0$ by complementary slackness.

Combining these cases, we conclude that

$$\pi(y) = \begin{cases} 0, & r(y) - \lambda \leq 0, \\ \pi_{\text{ref}}(y)\beta^{-1}(r(y) - \lambda) & r(y) - \lambda > 0. \end{cases}$$

This is equivalent to $\pi(y) = \pi_{\text{ref}}(y) \cdot \text{relu}(\beta^{-1}(r(y) - \lambda))$, which is also optimal under strong duality. Finally, the condition $\sum_y \pi(y) = \alpha$ implies that $\lambda$ must be chosen such $\pi \in \Delta_\alpha(\mathcal{Y})$ normalizes to $\alpha$. $\qquad \square$

**Lemma G.3.** *Recall* $\Phi(\lambda) = \mathbb{E}_{y \sim \pi_{\text{ref}}}\big[\text{relu}\big(\beta^{-1}(r(y) - \lambda)\big)\big]$, *and given $N$ samples from $y_1, \ldots, y_N \sim \pi_{\text{ref}}$, define*

$$\widehat{\Phi}(\lambda) = \frac{1}{N}\sum_{i=1}^{N}\text{relu}\big(\beta^{-1}(r(y_i) - \lambda)\big).$$

*Fix any $\gamma \in \mathbb{R}_+$. With probability at least $1 - \delta$, for all $\lambda \in [-\gamma, R_{\max} - \gamma]$, we have*

$$\max\left\{\frac{7}{8}\Phi(\lambda) - \widehat{\Phi}(\lambda), \widehat{\Phi}(\lambda) - \frac{9}{8}\Phi(\lambda)\right\} \leq \frac{1}{8} + 12\left(\frac{R_{\max} + \gamma}{\beta}\right)\frac{\log\left(\frac{60R_{\max}}{\beta\delta}\right)}{N}.$$

**Proof of Lemma G.3.** We start with Bernstein's inequality, which states that for a bounded random variable $X \in [a, b]$, with probability at least $1 - \delta'$, the empirical mean $\widehat{\mathbb{E}}[X] = \frac{1}{N}\sum_{i=1}^{N}X_i$ satisfies

$$\left|\widehat{\mathbb{E}}[X] - \mathbb{E}[X]\right| \leq 2\sqrt{\frac{\mathbb{V}[X]\log(\frac{2}{\delta})}{N}} + \frac{4(b-a)\log(\frac{2}{\delta})}{N},$$

where $\mathbb{V}[X]$ is the variance of $X$. Now fix $\lambda$, and let the random variable be $X = \text{relu}(r(y) - \lambda)$. Since $\lambda \in [-\gamma, R_{\max} - \gamma]$, the random variable $X \in \left[0, \frac{R_{\max} + \gamma}{\beta}\right]$, and henceforth we refer to the range as $R = \frac{R_{\max} + \gamma}{\beta}$. We can bound the variance of this random variable as

$$\mathbb{V}[X] \leq \mathbb{E}[X^2] \leq R \cdot \mathbb{E}[X] = R \cdot \Phi(\lambda).$$

Then with probability at least $1 - \delta'$, we have

$$\begin{aligned} \left|\Phi(\lambda) - \widehat{\Phi}(\lambda)\right| &\leq 2\sqrt{\frac{R\Phi(\lambda)\log(\frac{2}{\delta'})}{N}} + \frac{4R\log(\frac{2}{\delta'})}{N} \\ &\leq \frac{\Phi(\lambda)}{8} + \frac{8R\log(\frac{2}{\delta'})}{N} + \frac{4R\log(\frac{2}{\delta'})}{N} \\ &= \frac{\Phi(\lambda)}{8} + \frac{12R\log(\frac{2}{\delta'})}{N}, \end{aligned}$$

where we use the AM-GM inequality in the second inequality. This then implies that

$$\begin{aligned} \Phi(\lambda) - \widehat{\Phi}(\lambda) &\leq \frac{\Phi(\lambda)}{8} + \frac{12R\log(\frac{2}{\delta'})}{N}, \quad \text{and} \\ \widehat{\Phi}(\lambda) - \Phi(\lambda) &\leq \frac{\Phi(\lambda)}{8} + \frac{12R\log(\frac{2}{\delta'})}{N}, \end{aligned}$$

which after rearranging and plugging in the expression for $R$, results in

$$\frac{7}{8}\Phi(\lambda) - \widehat{\Phi}(\lambda) \le 12\left(\frac{R_{\text{max}} + \gamma}{\beta}\right)\frac{\log\left(\frac{2}{\delta'}\right)}{N}, \quad \text{and} \tag{24}$$

$$\widehat{\Phi}(\lambda) - \frac{9}{8}\Phi(\lambda) \le 12\left(\frac{R_{\text{max}} + \gamma}{\beta}\right)\frac{\log\left(\frac{2}{\delta'}\right)}{N},$$

which proves that the desired inequality holds for any $\lambda$ fixed a-priori.

We now convert this guarantee into a uniform-in-$\lambda$ bound. Consider the interval $[-\gamma, R_{\text{max}} - \gamma]$, and, for some fixed $\varepsilon$, let $\Lambda_\varepsilon = \left\{-\gamma + \varepsilon \cdot i : i = 1, \ldots, \lceil\frac{R_{\text{max}}}{\varepsilon}\rceil\right\}$, which has cardinality $|\Lambda_\varepsilon| \le \frac{2R_{\text{max}}}{\varepsilon}$. Then for any $\lambda \in [-\gamma, R_{\text{max}} - \gamma]$, we can always find $\lambda' \in \Lambda_\varepsilon$ such that $|\lambda - \lambda'| \le \varepsilon$, for which

$$\left|\text{relu}\left(\beta^{-1}(r(y) - \lambda)\right) - \text{relu}\left(\beta^{-1}(r(y) - \lambda')\right)\right| \le \beta^{-1} \cdot |\lambda - \lambda'| \le \beta^{-1} \cdot \varepsilon.$$

Applying Eq. (24) and taking a union bound, with probability at least $1 - \delta$ we have that for all $\lambda' \in \Lambda_\varepsilon$,

$$\frac{7}{8}\Phi(\lambda') - \widehat{\Phi}(\lambda') \le 12\left(\frac{R_{\text{max}} + \gamma}{\beta}\right)\frac{\log\left(\frac{4R_{\text{max}}}{\varepsilon\delta}\right)}{N}$$

$$\widehat{\Phi}(\lambda') - \frac{9}{8}\Phi(\lambda') \le 12\left(\frac{R_{\text{max}} + \gamma}{\beta}\right)\frac{\log\left(\frac{4R_{\text{max}}}{\varepsilon\delta}\right)}{N},$$

thus for all $\lambda \in [-\gamma, R_{\text{max}} - \gamma]$,

$$\frac{7}{8}\Phi(\lambda) - \widehat{\Phi}(\lambda) \le \frac{15}{8}\frac{\varepsilon}{\beta} + 12\left(\frac{R_{\text{max}} + \gamma}{\beta}\right)\frac{\log\left(\frac{4R_{\text{max}}}{\varepsilon\delta}\right)}{N}$$

$$\widehat{\Phi}(\lambda) - \frac{9}{8}\Phi(\lambda) \le \frac{15}{8}\frac{\varepsilon}{\beta} + 12\left(\frac{R_{\text{max}} + \gamma}{\beta}\right)\frac{\log\left(\frac{4R_{\text{max}}}{\varepsilon\delta}\right)}{N},$$

and choosing $\varepsilon = \frac{\beta}{15}$ results in

$$\frac{7}{8}\Phi(\lambda) - \widehat{\Phi}(\lambda) \le \frac{1}{8} + 12\left(\frac{R_{\text{max}} + \gamma}{\beta}\right)\frac{\log\left(\frac{60R_{\text{max}}}{\beta\delta}\right)}{N}$$

$$\widehat{\Phi}(\lambda) - \frac{9}{8}\Phi(\lambda) \le \frac{1}{8} + 12\left(\frac{R_{\text{max}} + \gamma}{\beta}\right)\frac{\log\left(\frac{60R_{\text{max}}}{\beta\delta}\right)}{N}$$

which when combined proves the lemma statement. $\qquad\square$

### G.2. Proof of Theorem 4.2

**Proof of Theorem 4.2.** Fix $N \lesssim \frac{1}{\varepsilon_{\text{RM}}}$. We apply the first part of Theorem F.4 with $\varepsilon_{\text{RM}} = \varepsilon_{\text{RM}}$, $p = \frac{1}{2}$, $C = O(\varepsilon_{\text{RM}}^{-1})$, and $\varepsilon = \frac{1}{c_1 \cdot N \cdot \sqrt{2C^{\pi^\star}(x)}}$. Note that the construction has $\mathcal{C}^{\pi^\star}(x) = O(\log C)$. With this set of parameters, any algorithm $\mathcal{A}$ using $N'$ such that

$$N' < c_1 \cdot \left(\mathcal{C}^{\pi^\star}(x) \cdot \varepsilon^2\right)^{-\frac{1}{2}} = c_1 \cdot \left(\frac{1}{2c_1^2 N^2}\right)^{-\frac{1}{2}} = 2N$$

must suffer regret at least

$$J(\pi^\star) - J(\widehat{\pi}_{\mathcal{A}}) > c_2 \cdot \left(\mathcal{C}^{\pi^\star}(x) \cdot \varepsilon^2\right)^{\frac{1}{2}}$$

$$= c_2 \cdot \left(\frac{\mathcal{C}^{\pi^\star}(x)}{2c_1^2 \cdot \mathcal{C}^{\pi^\star}(x) \cdot N^2}\right)^{\frac{1}{2}}$$

$$= \frac{c_2}{2c_1} \cdot \frac{1}{N},$$

which is therefore a lower bound that applies to $N' = N$.

$\square$

