# OpenReview forum: "Is Best-of-N the Best of Them? Coverage, Scaling, and Optimality in Inference-Time Alignment"
_ICML.cc/2025/Conference — ICML 2025 poster_

### Official Review · Reviewer_rHW3 · 2025-03-08

**Overall Recommendation:** 3

**Summary:**

This paper investigates inference-time alignment in language models and demonstrates that naively scaling the Best-of-N heuristic leads to reward hacking, causing performance degradation beyond a certain computational threshold. The authors introduce a new algorithm, InferenceTimePessimism, which leverages χ²-regularization and rejection sampling to mitigate reward hacking effectively. The authors provide theoretical guarantees showing InferenceTimePessimism achieves optimal regret.

**Claims And Evidence:**

The paper's claims are supported by theoretical proofs and experimental results. The paper clearly shows that the fundamental limitations of Best-of-N alignment through explicit regret analysis. In addition, it also show the robustness and optimal regret guarantees of InferenceTimePessimism.

**Essential References Not Discussed:**

Not sure about the essential references not discussed.

**Experimental Designs Or Analyses:**

The experimental designs were found to be sound. Experiments effectively demonstrate the robustness of InferenceTimePessimism in mitigating reward hacking and improving performance across multiple reward models and tasks.

**Methods And Evaluation Criteria:**

The methods and evaluation criteria selected are appropriate and justified. The use of standard benchmark datasets such as GSM8K, MMLU, and MATH is suitable to validate alignment effectiveness and reward model quality.

**Other Comments Or Suggestions:**

N/A

**Other Strengths And Weaknesses:**

- Novel theoretical insights significantly advancing understanding of inference-time alignment.
- Clearly articulated and robust theoretical framework that provides meaningful guidance for practical algorithm design.
- Empirical validation effectively demonstrates theoretical predictions, enhancing practical relevance.

**Questions For Authors:**

Have you tested or analyzed the performance and robustness of InferenceTimePessimism with significantly noisier or weaker reward models? If so, how sensitive is the algorithm to degradation in reward model accuracy? Similarly, have you experimented with or considered extending InferenceTimePessimism to more open-ended, subjective, or complex tasks, and what challenges might arise in these settings?

**Relation To Broader Scientific Literature:**

Not sure about the relation To broader scientific literature.

**Theoretical Claims:**

The correctness of the theoretical claims was found to be solid. The proofs of lower bounds establishing the necessity of coverage and the limitations of the Best-of-N approach are rigorous and correct.

---

### Official Review · Reviewer_WiRT · 2025-03-11

**Overall Recommendation:** 4

**Summary:**

This paper examines inference-time alignment, where additional computation at generation time is used to improve language model outputs. Specifically, the authors focus on the widely used best-of-$n$ approach, which generates multiple responses and selects the one with the highest reward according to a (possibly imperfect) reward model.
The key contribution is a theoretical framework that explains when and why best-of-$n$ works, and more importantly, when it fails. The authors show that while best-of-$n$ can be effective with proper tuning, it inevitably suffers from "reward hacking" when scaled too aggressively. This happens because imperfections in the reward model become amplified when $n$ is large.
To address this, they introduce InferenceTimePessimism, an algorithm that deliberately uses inference-time computation to quantify and account for uncertainty in the reward model. Their theoretical analysis proves this approach is optimal in terms of regret and maintains performance even with increased computation.
The authors validate their theoretical findings with experiments on benchmarks using various language and reward models, demonstrating that their algorithm can mitigate reward overoptimization.

**Claims And Evidence:**

Yes

**Essential References Not Discussed:**

No

**Experimental Designs Or Analyses:**

No

**Methods And Evaluation Criteria:**

Yes

**Other Comments Or Suggestions:**

No

**Other Strengths And Weaknesses:**

### Strengths

- **Novel Theoretical Framework:** The paper is the first to build the theoretical framework of best-of-$n$ with ideas from offline RL, with the core notion of coverage.
- **Theoretical Guarantees:** The authors first give comprehensive results (both upper and lower bounds) of best-of-$n$. The results can be stronger when a uniform converage is guaranteed. Then they prove their algorithm, InferenceTimePessimism, is regret-optimal within their framework, which is a significant theoretical contribution.
- **Practical Algorithm:** The authors give the practical implementation for InferenceTimePessimism and perform extensive experimental validation on multiple tasks, reward models, and language models.

### Weaknesses

- **Lacking some cases in theory:** For Theorem 3.1, the authors didn't discuss the case where $\varepsilon_{RM} (x) = 0$. If we directly apply the results, then the regret is unbounded, which seems counter-intuitive. For Theorem 4.1, the authors didn't give a dependency on $N$, which makes it hard to compare with Theorem 3.1.

**Questions For Authors:**

Regarding the weakness, what is the bound for best-of-$n$ when $\varepsilon_{RM} (x) = 0$? What is the dependency on $N$ for InferenceTimePessimism?

**Relation To Broader Scientific Literature:**

TBA

**Theoretical Claims:**

No

---

### Official Review · Reviewer_KBSW · 2025-03-13

**Overall Recommendation:** 4

**Summary:**

The paper first theoretically shows the overoptimization problem is inevitable with the well-known best-of-N algorithm, especially when N increases. Then they propose Inference-Time Pessimism Algorithm, and show that the proposed algorithm resolves the overoptimization problem and also achieves optimal regret rate in terms of the reward approximation error. The paper also provides experiments to validate their theoretical findings.

**Claims And Evidence:**

**Claims with evidences**
- On the Best-of-N (BoN) algorithm, the paper shows that (i) it causes overoptimization in the worst cases when N increases (Theorem 3.2), and (ii) the regret achieved by BoN is also suboptimal in terms of $\epsilon_{RM}$, the approximation error of an estimated reward. (Theorem 3.1, 3.2)
- To resolve these issues, the paper proposes Inference-Time Pessimism Algorithm (Algorithm 1), which is a sampling procedure from $\chi^2$-regularized reward maximizing distribution, by rejection sampling with N samples from the reference model. The proposed algorithm does not cause overoptimization by its nature, and also achieves optimal regret bound in terms of $\epsilon_{RM}$. (Theorem 4.1, 4.2)
- Experimental results:
  - Figure 1 validates that the proposed algorithm with tuned hyperparameter $\beta$, the strength of $\chi^2$ regularization, successfully avoids the overoptimization issue caused in BoN when N increases.
  - Figure 2 investigates the behavior of the proposed algorithm with varying $\beta$ compared to the BoN algorithm, with fixed $N=2^{13}$. However, this is an extreme setting and also unfair for BoN.

Overall, the claims are theoretically well-supported, and experiments validate the benefits of the proposed algorithms as predicted from their theories. However, I still have some concerns especially about the experimental evidences:
- The paper claims that the proposed algorithm is superior to BoN because (i) it avoids overoptimization and (ii) it achieves the improved regret rate. However, experimental results only verify the first benefit (i), and it is unclear how the second benefit (ii) appears in the real-world experiments.
- The performance of the proposed algorithm seems to depend on the choice of the hyperparameter of regularization strength $\beta$, but the behavior with changing $\beta$ is only shown with the extreme setting of $N=2^{13}$.

**Essential References Not Discussed:**

Yang et al. [1] also investigates the theoretical properties of the BoN algorithm in relation to KL-regularized reward-maximizing distribution, but not cited in this paper.

[1] Yang et al., "Asymptotics of Language Model Alignment" (ISIT'24)

**Experimental Designs Or Analyses:**

The experimental designs and analyses seem sound.

**Methods And Evaluation Criteria:**

The proposed method makes sense and is well-motivated. The experimental settings follow the standard ones.

**Other Comments Or Suggestions:**

N/A

**Other Strengths And Weaknesses:**

See Claims and Evidences.

**Questions For Authors:**

See Claims and Evidences.

**Relation To Broader Scientific Literature:**

The proposed algorithm successfully resolves the overoptimization problem caused with the BoN algorithm, which is a widely used method for inference-time alignment. In addition to analyses of the proposed algorithm, the theoretical results on BoN themselves are also novel and insightful.

**Theoretical Claims:**

I read some proofs in Appendix to understand the theoretical results, but did not check all of them.

---

### Official Review · Reviewer_jQEp · 2025-03-15

**Overall Recommendation:** 3

**Summary:**

The paper analyzes the Best-of-N (BoN) algorithm for selecting among language model generations and introduces InferenceTimePessimism, a new algorithm that mitigates reward hacking. The authors formalize inference-time alignment as improving a pre-trained policy’s responses using an imperfect reward model. They show that BoN can achieve optimal performance under strict coverage but suffers from reward hacking when N is large. InferenceTimePessimism is proven to be optimal and scaling-monotonic, with empirical validation across tasks like GSM8K, MMLU, and MATH.

**Claims And Evidence:**

The claims are well-supported by theoretical proofs and empirical results. The authors demonstrate BoN's limitations and prove InferenceTimePessimism's optimality and robustness through experiments on multiple tasks and models.

**Essential References Not Discussed:**

The paper could reference more recent work on preference-based learning and scaling laws for language models to provide additional context.

**Experimental Designs Or Analyses:**

The experiments are well-designed, covering multiple tasks and models. The results show InferenceTimePessimism avoids reward hacking and outperforms BoN in many cases.

**Methods And Evaluation Criteria:**

The methods are appropriate, focusing on inference-time alignment and evaluated using standard benchmarks (e.g., GSM8K, MMLU). The criteria, including accuracy and estimated reward, effectively measure performance.

**Other Comments Or Suggestions:**

Please see above

**Other Strengths And Weaknesses:**

Strengths:

1. Addresses a timely problem with theoretical insights and practical algorithms.

2. InferenceTimePessimism is a novel contribution with rigorous proofs and empirical validation.

3. The proposed methods work reasonably well.

Weaknesses:

1. Limited guidance on tuning the regularization parameter $\beta$

2. Experiments focus on mathematical tasks; more diverse tasks (e.g., dialogue) would strengthen generalizability.

Q1. How should practitioners choose $\beta$ in InferenceTimePessimism?

Q2. Have you considered evaluating on more diverse tasks like open-ended dialogue?

Q3. What are the computational costs of InferenceTimePessimism compared to BoN?

**Questions For Authors:**

Please see above

**Relation To Broader Scientific Literature:**

The paper connects well to prior work on reward overoptimization and offline RL, extending ideas like pessimism to inference-time alignment.

**Theoretical Claims:**

The theoretical claims are supported by rigorous proofs. The authors prove InferenceTimePessimism achieves optimal regret and is scaling-monotonic, with detailed proofs in the supplementary material.

---

### Decision · Program_Chairs · 2025-05-01

**Decision:**

Accept (poster)

**Comment:**

This paper provides a theoretical analysis of Best-of-N (BoN) sampling for inference-time alignment, formalizing the problem and highlighting BoN's key limitations regarding reward hacking and suboptimality. The introduction and analysis of their pessimistic algorithm is a good contribution, and includes regret guarantees and scaling-monotonicity, which are mostly supported by empirical results. It is still unclear how to optimally tune the hyperparameter of the proposed approach without an oracle. Also, an analysis of the computational overhead compared to standard BoN is missing. While the work is self-contained and focuses correctly on inference-time mechanisms, a direct comparison to inference-aware training methods (like those in the IAFT paper) may add additional context.